# Engineering and characterization of gymnosperm sapwood toward enabling the design of water filtration devices

Krithika Ramchander [1✉], Megha Hegde[2], Anish Paul Antony [2], Luda Wang[1,3,4], Kendra Leith[2], Amy Smith[2] & Rohit Karnik [1✉]

Naturally-occurring membranes in the xylem tissue of gymnosperm sapwood enable its use as an abundantly-available material to construct filters, with potential to facilitate access to safe drinking water in resource-constrained settings. However, the material's behavior as a filter is poorly understood, and challenges such as short shelf life have not been addressed. Here, we characterize the operational attributes of xylem filters and show that the material exhibits a highly non-linear dependence of flow resistance on thickness upon drying, and a tendency for self-blocking. We develop guidelines for the design and fabrication of xylem filters, demonstrate gravity-operated filters with shelf life >2 years, and show that the filters can provide >3 log removal of *E. coli*, MS-2 phage, and rotavirus from synthetic test waters and coliform bacteria from contaminated spring, tap, and ground waters. Through interviews and workshops in India, we use a user-centric approach to design a prototype filtration device with daily- to weekly-replaceable xylem filters, and uncover indicators of social acceptance of xylem as a natural water filter. Our work enhances the understanding of xylem as a filtration material, and opens opportunities for engineering a diverse range of low-cost, biodegradable xylem-based filtration products on a global scale.

[1] Department of Mechanical Engineering, Massachusetts Institute of Technology, Cambridge, MA, USA. [2] D-Lab, Massachusetts Institute of Technology, Cambridge, MA, USA. [3] Present address: Institute of Microelectronics, School of Electronics Engineering and Computer Science, Peking University, Beijing, China. [4] Present address: Academy for Advanced Interdisciplinary Studies, Peking University, Beijing, China. ✉email: krithika.ramchander@gmail.com; karnik@mit.edu

Diarrheal diseases caused by microbial contamination of water and poor sanitation are a global problem. In 2019, diarrheal diseases accounted for 1.5 million deaths per year, primarily in resource-limited settings amongst children under the age of five[1]. Majority of the deaths (57.8%) are caused by bacterial pathogens, while water-borne viruses and protozoa account for 33.8% and 8.3% of the fatalities, respectively[2]. Household water treatment (HWT) methods like chlorination, solar disinfection, and filtration can significantly reduce the risk of diarrheal diseases[3,4]. However, the adoption of these methods in resource-constrained settings is often hindered by their limited availability in remote locations, incompatibility with local socio-cultural practices, high cost, or lack of suitable financing schemes[3–5]. In addition, the common perception that water that appears clear is safe for drinking, and the difficulty in appreciating the link between diarrheal diseases and poor water quality, also impede uptake[3–5]. Novel water treatment technologies that are inexpensive, readily available, socially acceptable, and effective against water-borne pathogens have the potential to address these challenges and improve access to safe drinking water.

Gymnosperm (non-flowering plants like conifers) wood, a common material that is widely available and traded across the globe[6], presents the intriguing possibility of creating inexpensive, sustainable, and socially acceptable filters to address this challenge[7–11]. The gymnosperm sapwood consists largely of xylem tissue that conducts sap, with longitudinally-oriented conduits called tracheids up to 10-mm long that are interconnected by 'pit membranes' with pore size ranging from 100 to 500 nm (Fig. 1a and b)[12]. Fluid flowing through a transverse section of a branch that is thicker than a single tracheid must therefore pass through the pit membranes, which can act as physical sieves that trap particulate contaminants present in water[7] (Fig. 1c). Compared to most angiosperms (flowering plants), the short length of tracheids and their high proportion in the cross-section makes gymnosperm sapwood better-suited to creating compact filters.

The unique structure of gymnosperm xylem gives rise to two interesting questions: (a) is the xylem a suitable material for water filtration, and (b) if so, how can it be engineered to create practically useful water filters. Previous studies have reported that pit membranes in pine xylem can filter bacteria from deionized water and incorporation of silver nanoparticles in xylem can enhance removal of bacteria[7,10]. However, several other material characteristics of xylem that are critical for practical water filtration applications, such as its structural stability over the course of its shelf- and operational life, susceptibility to different foulants present in water, and mechanisms of fouling, remain to be explored. While the hydraulic properties of xylem have been well-characterized in the context of sap transport in plants[13–15], xylem's functional attributes as a water filter, such as flow rate, filtration capacity, and variation in flow rate over time, particularly with contaminated water as the fluid medium and in the absence of active transport mechanisms that regulate flow in plants, are currently not well understood. A known challenge with xylem filters is that their permeance (defined as flow rate per unit area per unit pressure difference) drops by a factor of ~100 upon drying, which limits their usability in dry state[7]. Wet filters have reasonable permeance, but have limited shelf life due to their propensity for degradation and are heavy to transport. Thus, identifying the underlying mechanism that leads to this behavior and developing methods for preserving xylem in dry state are critical for their supply and distribution, particularly to remote, low-resource settings where they are most needed. In addition, simple and inexpensive methods for filter design and manufacture that help tailor xylem's functional attributes to suit practical needs are required to facilitate technology translation.

Here, we investigate the material attributes of xylem to reveal a unique nonlinear dependence of permeance on filter thickness, intrinsic tendency for 'self-blocking', and susceptibility to organic and dust foulants in water. By studying the filtration capacity, permeance, and its variation over the operational lifetime with different water qualities, we characterize the performance of these filters and evaluate their suitability for practical water filtration applications. Literature reports and our field trips to India revealed that, to be useful in households in resource-limited settings, xylem filters should (a) process at least 8 L of water to meet the daily drinking water requirement (see Supplementary Note 1), (b) have flow rates of at least 1 L/h, (c) effectively remove contaminants[16], (d) function reliably with contaminated water, (e) operate under gravity with heads <1 m to minimize operation costs and space requirements, and (f) be easy to access and use[16] (Supplementary Note 1). By combining our insights on material behavior with an understanding of how the filter's geometry (thickness, area) affects its performance, we develop a simple, inexpensive manufacturing method that can be performed in resource-limited settings to transform gymnosperm xylem into a dry-preservable, biodegradable, lightweight filter that meets the aforementioned metrics. Further, through field studies in India, we demonstrate the practical utility of this technology by manufacturing filters locally, validating their performance with natural water sources, and presenting evidence for positive user reception toward xylem filters. To illustrate potential for translation into a practically useful product, we develop a functional device prototype using a user-centered design approach.

Our work enhances the understanding of xylem as a water filtration material and presents the engineering tools necessary for creating a diverse range of xylem-based filtration products. The ability to create filters from different gymnosperms, widespread availability of gymnosperm xylem[17] (Fig. 1d), low cost, natural appeal, ease of transport and distribution, and the traditional comfort associated with wood, could help xylem filters lower the barriers of access, affordability, and social acceptance, and thereby facilitate access to safe drinking water.

## Results

**Nonlinear dependence of permeance on filter thickness**. The blockage of xylem filters upon drying is related to the physiological function of pit membranes that have evolved to protect the plant against cavitation (i.e., nucleation of vapor bubbles) that could severely disrupt sap flow[12]. In gymnosperms, surface tension forces of a receding liquid meniscus (corresponding to an advancing vapor bubble) pull the pit membrane toward an aperture in the cell wall; water-mediated adhesive forces cause the pit membrane to seal against the cell wall, thereby isolating any cavitated conduits[18,19]. While the exact mechanism underlying this phenomenon, referred to as 'pit aspiration', remains to be elucidated, it relies on the presence of water to mediate adhesion[19]. Similar to cavitation, drying induces the formation of liquid–vapor interfaces in the xylem, which triggers pit aspiration and reduces the permeance (with prior work reporting 100× drop in flow rate for 1-inch-thick filters[7]).

To retain some permeance in dried filters, we examined the effect of reducing filter thickness. Traditionally, Darcy's law, which is commonly applied to porous media and predicts a linearly inverse relation between thickness and permeance (i.e., permeability, defined as permeance normalized by thickness, is constant), has been used to model the permeance of xylem[20,21]. Since filters have to be thicker than the conduit length to ensure contaminant removal and gymnosperm conduits are typically <0.22-inch long[14], we expected that reducing filter thickness from 1 to 0.25 inch would lower flow resistance without compromising

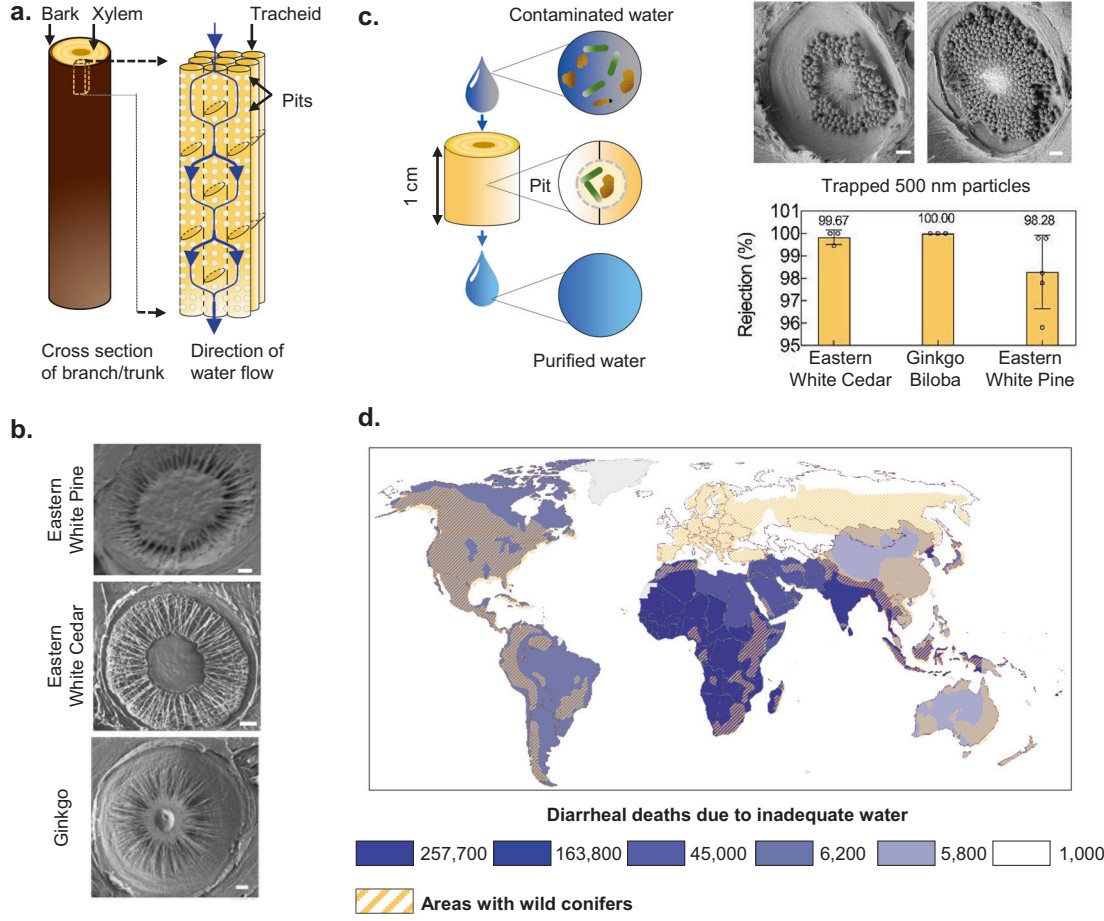

**Fig. 1 Gymnosperm xylem water filter. a** Illustration of gymnosperm xylem structure. **b** Scanning Electron Microscopy (SEM) images of pit membranes in different gymnosperms. Scale bar, 1 μm. **c** Schematic depiction and SEM images of filtration of 500-nm particles by xylem pit membranes in a section of a branch (ginkgo, 1-cm diameter, 0.25-inch thickness). Scale bar, 1 μm. The bar graph shows rejection of 500-nm particles by fresh xylem filters made from different tree species (1-cm diameter, 0.25-inch thickness). Mean ± s.d.; $n = 3$ different filters; see "Methods" for details. **d** The global distribution of wild conifers overlaid on the number of diarrheal deaths caused by inadequate (contaminated) water in 2016 within each colored region illustrates the potential for scaling and impact of xylem-based filtration devices. The numbers in the legend represent the total diarrheal deaths for the regions shaded in the same color (world map by Diana Beltekien adapted from https://ourworldindata.org/world-region-map-definitions under Creative Commons BY license; see "Methods" for figure construction details and references).

the rejection ability. Experiments with 1-μm microspheres (used as proxy for *E. coli*, justification provided in "Methods") confirmed that the rejection ability of 0.25-inch-thick filters made from Eastern white pine (*Pinus strobus*) was comparable to 0.50- and 1-inch-thick filters (Fig. 2a). However, in stark contrast to fresh filters where Darcy's law was followed, the inverse dependence of permeance on thickness in dried filters was highly nonlinear; permeability dropped abruptly on increasing filter thickness beyond 0.25 inch (Fig. 2b).

To explain this observation, we note that the permeability of dried xylem filters is a function of not only the flow resistance of tracheids and pit membranes[13,14], but also the tracheid interconnectivity[22]. The length scale over which tracheids maintain connectivity depends on the degree to which the pit membranes get blocked during drying, and corresponds to cluster size in percolation theory[22–24]. Filters much thicker than this length scale will be impermeable to flow, while those that are thinner, will have non-zero permeance (Fig. 2c). When filter thickness is comparable to this length scale, a highly nonlinear dependence of permeance on thickness that deviates strongly from Darcy's law, is expected. The experimental results suggest that the length scale of connectivity in dried filters was ~0.25 inch, which also implies that any 0.25-inch section of a

longer, impermeable filter should have non-zero permeance. This hypothesis was confirmed through experiments where 1.5-inch dried filters made from Eastern white pine were completely blocked, but 0.25-inch sections cut from the same blocked filters were permeable to flow (Fig. 2d).

The non-zero permeance of 0.25-inch sections in dried filters indicates that some pit membranes remain open (unaspirated) even after drying, i.e., the probability of pit aspiration blocking off a tracheid-tracheid interconnection upon drying is <1. We built a probabilistic model based on percolation theory[23] to capture the flow characteristics of a dried filter and better understand the dependence of permeance on filter thickness. We modeled the xylem as a node-edge network, where the tracheids and pits correspond to the nodes and edges, respectively (Fig. 2c). The model associated a probability $p$ for an edge being broken[23], which in the case of a xylem filter represents the likelihood of connectivity between two tracheids being broken by pit aspiration. Simulations of this percolation model in a simplified, 2-D xylem network using MATLAB corroborated experimental observations; for a given pit aspiration probability, the connectivity (and thus the permeance) dropped to zero beyond a critical filter thickness (Supplementary Fig. 1a). Further, the model suggested that the converse should also be true, i.e., for a

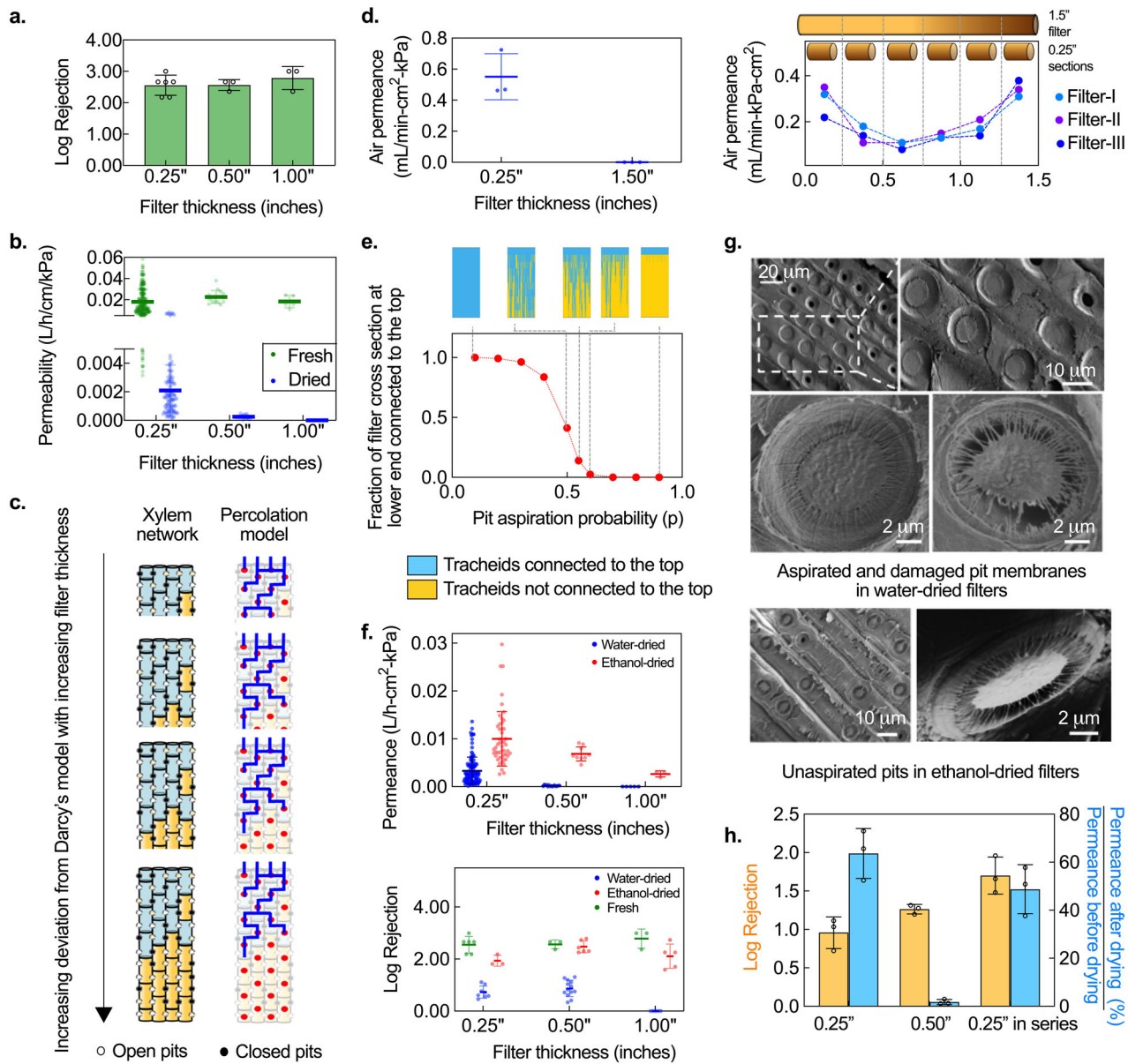

**Fig. 2 Thickness-dependent permeation and dry preservation of xylem filters (1-cm-diameter filters made from Eastern white pine; rejection of 1-μm microspheres).** **a** Rejection of 0.25-inch-thick fresh filters is comparable to thicker ones. Mean ± s.d. and individual data points are shown; *n* = 5, 3, 3 different filters with 0.25-, 0.50-, and 1-inch thickness. **b** Permeability (permeance normalized by thickness) is constant with filter thickness in fresh filters, but drops abruptly with increasing thickness in the case of dried filters. Mean ± s.d. and individual data points are shown; *n* = (280,118), (15,15), and (5,5) different filters (numbers denote fresh and water-dried, respectively) with 0.25-, 0.50-, and 1-inch thickness, respectively. **c** Schematic of the physical structure of xylem and the proposed percolation-based model (tracheids depicted as red dots) for filters of different thickness, illustrating percolation-governed length dependence. Black dots represent blocked tracheid connections (pit aspiration probability, *p* = 0.35 for all filters). Blue shading and lines represent tracheids and flow pathways that are connected to the top surface; there is no flow pathway from the top to bottom surface for the two thickest filters. **d** Air permeance of water-dried filters. 0.25-inch-thick water-dried filters are permeable to air whereas 1.5-inch-thick filters are not but 0.25-inch sections cut from 1.5-inch-thick blocked filters are permeable, consistent with percolation-governed permeance. Left: Individual data points and mean ± s.d. are shown; *n* = 3 different filters. Right: Each data point corresponds to a single measurement on a 0.25-inch-thick section. **e** Simulation results show that interconnectivity of tracheids to the top face decreases with pit aspiration probability. Inset depicts interconnected (blue) and isolated (yellow) tracheids at different pit aspiration probabilities. **f** Ethanol-drying reduces pit aspiration and improves permeance and rejection over water-drying in a thickness-dependent manner. Individual data points and mean ± s.d. are shown. Top: *n* = (118, 41), (15, 9), and (5,3) different filters (numbers denote water-dried and ethanol-dried, respectively) with 0.25-, 0.50-, and 1.00-inch thickness, respectively. Bottom: *n* = (8, 3, 6), (13, 6, 3), and (5, 5, 3) different filters (numbers denote water-dried, ethanol-dried, and fresh filters, respectively) with 0.25-, 0.50-, and 1.00-inch thickness, respectively. **g** SEM images reveal aspirated or damaged pit membranes in water-dried filters, but not in ethanol-dried filters. **h** Two 0.25-inch-thick water-dried filters stacked in series perform significantly better than 0.25-inch and 0.50-inch-thick water-dried filters. Mean ± s.d; *n* = 3 different filters.

given filter thickness, there exists a critical probability $p = p_c$, at which there is transition from zero to non-zero permeance (Fig. 2e, see Supplementary Note 2 for model details).

**Dry preservation of xylem filters.** The knowledge of how xylem permeance is affected by thickness and pit aspiration probability ($p$) suggests two methods that could be used for preserving permeance in dried filters. First, permeance in dried filters may be retained by restricting their thickness to below a certain threshold value but above the tracheid length (~0.25 inches for Eastern white pine). Second, methods could be implemented to mitigate pit aspiration and thereby retain permeance in thicker filters.

Previous studies have shown that pit aspiration can be reduced by replacing the sap in the xylem with non-aqueous solvents, like alcohols, as it precludes water-mediated adhesion between the pit membrane and the cell wall during drying[18,19,25]. To evaluate whether treatment with non-aqueous solvents can improve permeance, we compared the permeance of filters (made from Eastern white pine) that were dried after flushing with ethanol ('ethanol-dried') to those that were flushed with water before drying ('water-dried'). Ethanol-dried filters exhibited higher permeance than their water-dried counterparts (Fig. 2f); the effect was more pronounced for thicker filters (0.5- and 1.0-inch) where water-dried filters were almost completely blocked whereas ethanol-dried filters retained permeance. Similar effects on permeance were observed on treating filters with other alcohols like isopropanol (see Supplementary Fig. 1b–f and Supplementary Note 3 for more details on solvent-based preservation). When benchmarked against commercial microfiltration membranes with similar pore size, the permeance of 0.25-inch ethanol-dried filters was comparable. The permeance range for commercial membranes is 0.002–0.05 L/h.cm$^2$.kPa[26–30] while 95% of the ethanol-dried filters from among 47 filters made from different Eastern white pine trees in Cambridge, MA, tested over a 2-year period consistently had permeance >0.005 L/h.cm$^2$.kPa (Supplementary Fig. 1b).

The rejection performance of ethanol-dried filters with 1-µm microspheres was significantly better than water-dried filters ($p < 0.001$ for 0.25-, 0.50-, and 1-inch filters, respectively) and comparable to fresh filters ($p = 0.02$, 0.59, and 0.08 for 0.25-, 0.50-, and 1-inch filters, respectively) (Fig. 2f). Scanning electron microscopy (SEM) confirmed that the pit membranes in ethanol-dried filters were unaspirated and intact, whereas those in water-dried filters were aspirated or damaged, consistent with their low permeance and rejection (Fig. 2g)[25]. Nevertheless, the rejection performance of water-dried filters could be improved by stacking multiple filters in series (Fig. 2h). Two 0.25-inch water-dried filters in series had better rejection than a single 0.25-inch filter ($1.70 \pm 0.24$ log versus $0.95 \pm 0.16$ log ($p = 0.015$)), with the log rejection being additive, and higher rejection and permeance recovery than a 0.50-inch filter ($1.70 \pm 0.24$ log versus $1.26 \pm 0.06$ log ($p = 0.04$), and $48.8 \pm 4.9\%$ versus $1.8 \pm 1.0\%$ ($p = 0.0013$), respectively). Stacking could be also used in conjunction with solvent treatment to further improve the rejection performance of filters, which offers opportunities for tailoring the rejection capability of xylem to suit different applications.

Both methods of dry preservation, thickness control and non-aqueous solvent treatment, offer different advantages and disadvantages with respect to rejection performance, ease of implementation, and reliability. Thickness control could be particularly useful where access to solvents is difficult or expensive, whereas solvent treatment is likely to be more robust but relatively expensive to perform, although the cost may be reduced by solvent recovery and reuse. The solvent used for dry-preservation must be certified as food-grade and the level of residual solvent in dried filters should be maintained within the permissible limits for human consumption as prescribed by food safety standards[31].

**Self-blocking and its control.** In membrane-based filters, fouling due to contaminants in the feed water determines the filter's volumetric capacity, i.e., the total amount of water that can be processed before the filter needs to be replaced[32]. Surprisingly, we observed a decrease in the flow rate of xylem filters that eventually led to blockage after a certain period of time even when filtering uncontaminated, deionized (DI) water (Fig. 3a). After ruling out several potential mechanisms that could be responsible for this behavior (see Supplementary Note 4 and Supplementary Fig. 2a and b), we observed that filters soaked in DI water (without flow) over similar time durations were not blocked, indicating that fluid flow played an important role in the underlying mechanism leading to blockage (Supplementary Fig. 2c and d). Furthermore, SEM imaging revealed an apparent deposition of material on the pit membranes of the blocked filters (Fig. 3b; compare this to pit membranes in unblocked ethanol-dried filters shown in Fig. 2g). Deposition of material even with DI water indicated that the material must originate from the filter itself. Xylem is composed of cellulose and hemicellulose fibers and hydrophobic lignin polymers, of which hemicellulose fibers are highly amorphous and relatively easily soluble in water[33]. We therefore hypothesized that the dissolution of hemicellulose fibers in DI water and their convective re-deposition on the pit membranes gives rise to self-blocking of xylem filters. Analysis of the water filtered through the xylem filters under atomic force microscopy (AFM) revealed the presence of dissolved solids (Fig. 3b) and further Fourier transform infrared spectroscopy (FTIR) measurements confirmed the presence of hemicellulose, validating our hypothesis (Fig. 3c)[34].

Self-blocking of xylem imposes an intrinsic limit on filter life and its volumetric capacity. However, it could also safeguard users against the risk of using a filter degraded by prolonged exposure to contaminated water or trapped microbes and signal the need for filter replacement. The ability to regulate self-blocking is therefore important, as it can help balance performance and safety. Broadly, self-blocking may be regulated by fixing the molecules within the xylem (which could also reduce degradation), or prior removal of the material responsible for the behavior. Effect on structural integrity of pit membrane (critical for rejection performance) and ease of implementation in low-resource settings are considerations that govern the choice of such methods.

We leveraged the solubility of hemicellulose in water to develop a simple process for mitigating self-blocking by soaking the filters in hot water to remove hemicellulose. We identified optimal temperature and duration of soaking to improve volumetric capacity without compromising structural integrity of the pit membranes; soaking the filters in hot water at 60–65 °C and atmospheric pressure for 1 h before ethanol-drying doubled the capacity while maintaining its ability for filtration (Fig. 3d, e, Supplementary Fig. 2e). In practice, the volumetric capacity of filters will also be limited by the fouling due to external water contaminants. Consequently, the necessity for measures to minimize self-fouling will be low if external contaminant load is high, and hot water soaking may not be needed. It is to be noted that this soaking process is different from industrial hydrolysis of hemicellulose that is typically performed at high temperature and pressure for extraction of chemical derivatives such as sugars[35].

Eastern white pine filters fabricated using hot water soaking and ethanol-drying could maintain permeance >0.01 L/h.cm$^2$.kPa

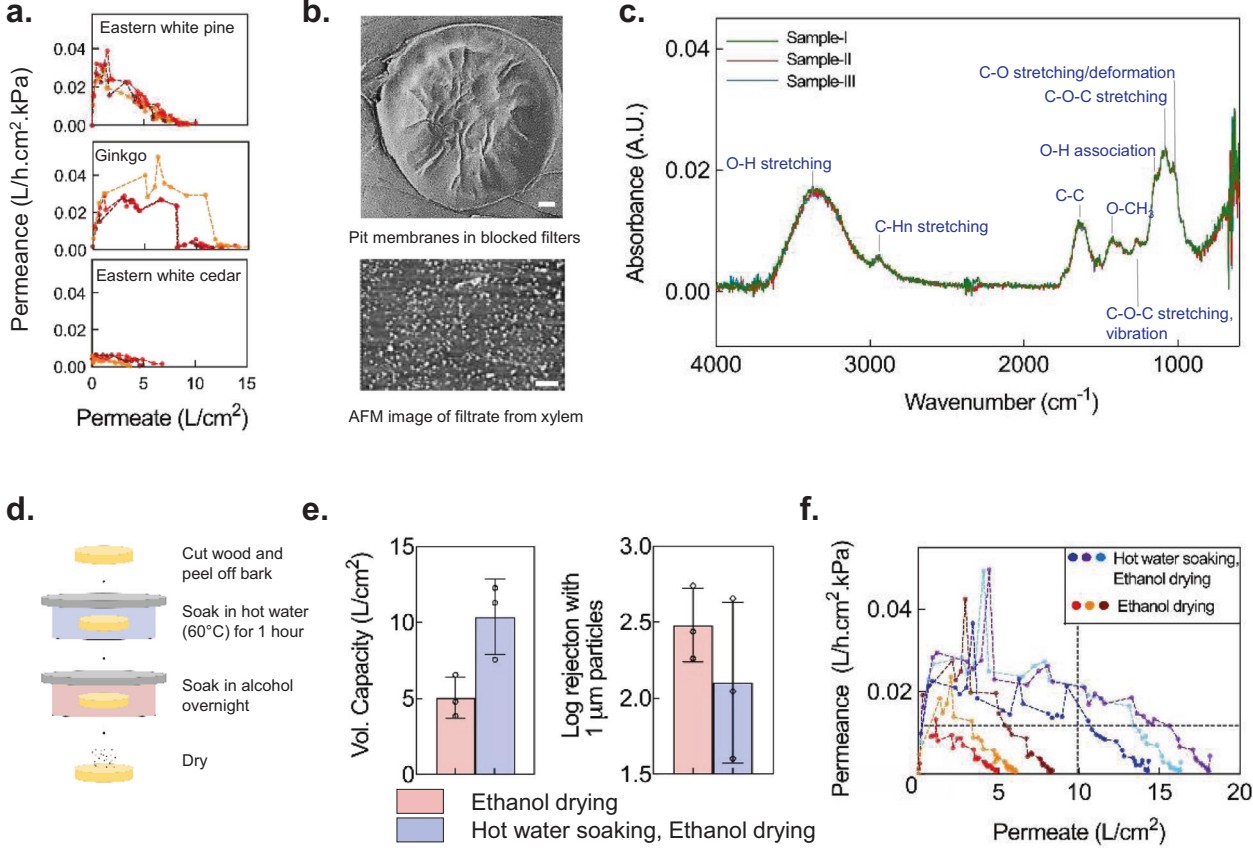

**Fig. 3 Self-fouling behavior. a** Permeance of 0.25-inch-thick ethanol-dried filters made from different gymnosperm species decreases with permeate volume when filtering deionized water; $n = 3$ different filters, denoted by different colors. **b** Microfibrils are covered by deposited material in pit membranes of blocked filters (SEM image, top) and filtrate dried on a surface contains particulates (AFM image, bottom), suggesting dissolution and deposition of organic material within the filter. Scale bars, 2 μm. **c** FTIR spectra of different samples of filtered water indicate that hemicellulose leaches out of xylem filters. Modes corresponding to FTIR peaks are specified. A.U. stands for Absorbance Unit. **d–f** Hot-water soaking improves volumetric capacity (vol. capacity) and retains rejection (0.375-inch-thick filters). Mean ± s.d. are indicated in (**e**); $n = 3$ different filters. Different colors denote different filters in (**f**)). Data were obtained with 1-cm diameter Eastern white pine filters operated under 1-m gravity head. The horizontal dashed line denotes the permeance (0.01 L/h.cm$^2$.kPa) corresponding to the target flow rate of 1 L/h with a 10-cm$^2$ filter area and 1-m gravity head, whereas the vertical dashed line corresponds to a volumetric capacity of 100 L, which is achieved by the hot water soaked and ethanol-dried filters while maintaining the target permeance.

while filtering at least 11 L/cm$^2$ of DI water (Fig. 3f). Thus, in the absence of fouling due to constituents in the feed water, filters with 10-cm$^2$ area (3.6-cm diameter) would achieve flow rates >1 L/h and volumetric capacity of ~100 L under gravity-driven operation with 1 m head (see Supplementary Fig. 2f for variation in permeance with permeate filtered for intermittent and continuous operation and Supplementary Fig. 2g for scaling of flow rate with filter area). However, we observed that the rejection performance of 0.25-inch-thick filters was sensitive to the variability in filter thickness, which is expected if the filter thickness approaches the length of the xylem conduits (tracheids) in Eastern white pine[36]. To circumvent this issue, the filter thickness was increased to 0.375 inches for all filters in subsequent studies.

**Effect of water quality.** Constituents in water such as humic acids or colloids typically cause fouling of membrane filters, reducing the flow rate with time. Understanding how such constituents affect the flow rate and volumetric filtration capacity of xylem filters is therefore essential to better inform how xylem filters would perform in practical settings. The World Health Organization (WHO) prescribes two kinds of synthetic test waters to evaluate the performance of household water treatment

technologies[37]: a general test water (GTW) representing high-quality groundwater or rainwater, and a challenge test water (CTW) with aggressive water specifications to represent turbid surface water (see Supplementary Fig. 3a for composition of GTW and CTW). With GTW, the volumetric capacity and peak permeance (highest permeance over the course of operation) of xylem filters (fabricated by hot-water soaking and ethanol-drying) were sufficient to meet the target metrics (flow rate >1 L/h and volumetric capacity >8 L). However, filter performance varied significantly with water quality; both peak permeance and capacity with CTW (0.022 ± 0.020 L/h.cm$^2$.kPa and 6.07 ± 4.40 L/cm$^2$, respectively) were an order of magnitude lower than those with GTW (0.002 ± 0.001L/h.cm$^2$.kPa and 0.58 ± 0.47 L/cm$^2$, respectively; Fig. 4a).

The deterioration in performance with CTW could be attributed to one or more of the water quality parameters that differ between CTW and GTW, which are (a) higher turbidity, (b) higher concentration of organics, and (c) the larger size of organic contaminants in CTW. To identify the key foulants that cause deterioration, we measured filtration capacity of xylem while selectively adding different constituents at varying concentrations. Xylem filters were most susceptible to fouling by humic acids (present in decomposed organic matter) followed by particulates (dust) (Fig. 4b and c). By contrast, tannic acid did

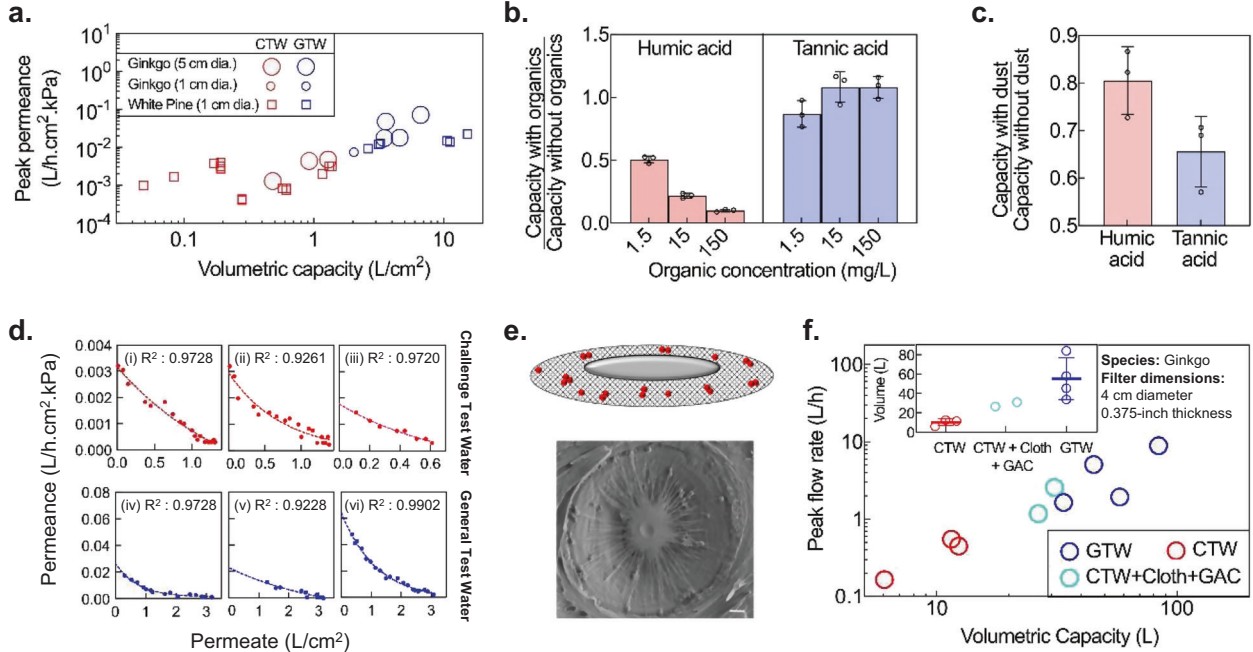

**Fig. 4 Effect of water quality on filter performance. a** Peak permeance and volumetric capacity normalized by area for general test water (GTW) and challenge test water (CTW). Each data point represents a different filter (one measurement per filter) with 0.375-inch thickness operated under 1-m gravity head. **b**, **c** Filter capacity is most susceptible to humic acid, followed by dust and tannic acid (1-cm diameter, 0.375-inch-thick Eastern white pine filters operated under 1-m gravity head; mean ± s.d.; $n = 3$ different filters) (see "Methods" for experiment details). In **b**, either humic acid or tannic acid is added to water. In **c**, water contains 70 mg/L dust or no dust in either 15 mg/L humic acid or tannic acid. **d** Decrease in filter permeance with CTW (red) and GTW (blue) is well-fitted by the intermediate fouling model (dashed lines). Each graph represents measurements on a single filter under 1-m gravitational head. (i)–(v): Eastern white pine, 1-cm diameter, 0.375-inch thickness. (vi): Ginkgo, 4-cm diameter, 0.375-inch thickness. **e** Schematic illustrating the deposition of foulant particles (red) on the pit membrane in the intermediate fouling model, which is consistent with foulant deposition observed by SEM in a partially fouled ginkgo filter. Scale bar, 1 μm. **f** Pre-treatment with cloth and granular activated carbon (GAC) improves the peak flow rates and volumetric capacity of ginkgo filters with GTW and CTW at 1-m gravity head. Inset shows mean ± s.d. of the volumetric capacity; $n = 3$, 2, and 4 different filters with CTW, CTW + Cloth + GAC, and GTW, respectively.

not impact filter capacity significantly, demonstrating that the filters have a low susceptibility for fouling with small, homogeneous organic molecules.

Fouling is a well-researched topic in membrane filtration and several fouling models have been developed to understand the nature of interaction between the foulants and membrane surface and aid membrane design, operation, and fouling control[38]. Based on the goodness of fit of different models to experimental data with GTW and CTW, we identified that the fouling behavior in xylem filters is best explained by the 'intermediate blocking' model. This model has commonly been used to represent the fouling of polymeric micro/ultrafiltration membranes by biological and organic contaminants[39–42] (Fig. 4d, see Supplementary Note 5 and Supplementary Fig. 3b,c for detailed comparison with other fouling models). In this model, foulant particles deposit randomly on the pit membranes and result in exponential decrease in permeance. SEM images of partially fouled filters were in agreement with this fouling mechanism (Fig. 4e). The fouling model helps predict the change in filter permeance with time for a given contaminant load; consequently, it can be used for estimating volumetric capacity, filter lifetime, and replacement frequency for different water qualities.

Knowledge that humic acid and dust particles adversely impact filter performance offers the possibility of mitigating their impact through approaches ranging from pre-treatment of water to chemical modification of xylem. To keep filter manufacturing simple and inexpensive, and accommodate variations in contaminant type and load, we explored pre-treatment methods that can be easily integrated in-line with xylem filters when the water

quality is poor. Specifically, we investigated cloth pre-filtration and granular activated carbon (GAC) adsorption to reduce the load of dust and humic acid, respectively[43]. Both these methods have been commonly used for household water treatment, but have limited efficacy in removing bacterial or viral pathogens from water[44,45]. After studying the adsorption kinetics of various types of GACs, we designed a GAC column to reduce humic acid content in CTW by 95% (see Supplementary Note 6, Supplementary Fig. 3d for GAC column design). When used in conjunction with cloth pre-filtration, the GAC column improved the performance of xylem filter with CTW significantly (Fig. 4f); on average, capacity and flow rates increased by a factor of ~3× and 5×, respectively.

In practice, pre-filtration is not essential for the operation of the filter; it is an option which, in conjunction with water quality, determines the flow characteristics. The decision whether to incorporate pre-treatment and the choice of pre-treatment would be governed by the tradeoff between the added convenience of longer filter lifetime or lower filter replacement frequency, cost, and the complication of an added replaceable component, plus the need to remove any chemical contaminants that may be present in the water (cost estimates for xylem filters and GAC provided in Supplementary Note 7). The replacement frequency of the cloth or the GAC module would vary depending on the type of cloth/GAC used, configuration of GAC module, and water quality. While the cloth pre-filter could be washed or replaced once it is dirty, the GAC might need replacement once every few months (1.5–6 months; see Supplementary Note 7 for estimates on GAC replacement frequency). The reduced lifetime or slower

flow rates even with newly-replaced xylem filters could be used as an indicator for pre-filtration module replacement.

In summary, these studies demonstrate that xylem filters offer promise for practical translation. Filters made from *Ginkgo biloba* (ginkgo) with an area of 13 cm$^2$ (4-cm diameter) using the fabrication protocol shown in Fig. 3d, operated under a 1-m gravity head could (a) process ~55 ± 21 L of GTW without pre-filtration and 28 ± 3 L of CTW with GAC and cloth pre-filtration, which is more than sufficient to meet the daily drinking water requirement of a household, (b) yield peak flow rates of 1.5–9 L/h depending on water quality (see Supplementary Fig. 3e–g for variation of flow rates over filter lifetime), (c) reject 99.76 ± 0.25% of 1-µm particles. Further, these filters had a shelf life of at least 1 year (the permeance of filters stored for 1 year was 0.0074 ± 0.0003 L/h.cm$^2$.kPa and the rejection of 1-µm microspheres was 99.92 ± 0.05%) and could be transported easily due to their lightweight (~7–8 g).

**Microbiological performance**. To assess the potential health impact of xylem filters and their effectiveness in reducing the risk of diarrheal diseases, we tested the filters' ability to remove *E. coli*, MS-2 phage, and rotavirus (the single largest causal organism of diarrhea[2]) from water. Xylem filters (4-cm diameter, 0.375-inch thickness, stored for 2 years, no pre-filtration) made from ginkgo were operated under a 1.2-m gravity head with GTW containing WHO-prescribed concentrations of *E. coli* (≥10$^6$ CFU/mL) and MS-2 phage (≥10$^5$ PFU/mL)[37] and NSF-prescribed concentrations of rotavirus (≥10$^4$ PFU/mL)[46]. *E. coli* and MS-2 phage were dosed simultaneously in the same test solution while rotavirus removal was tested separately. The bacteria and virus removal was tested at the start of filter operation and when permeance declined to 75, 50, and 25% of the initial value. After the first sampling point at the start of filter operation, dust was added to the test solution to accelerate clogging[47,48] (refer to "Methods" for further details on test procedure). The filters showed >4-log removal of rotavirus and >3-log removal of *E. coli* and MS-2 phage (Fig. 5a, data provided in Supplementary Table 1). With such rejection performance, xylem filters would fall under the 'comprehensive protection (high pathogen removal)' category (★★) as per the WHO scheme for classifying water treatment technologies (Fig. 5b)[48]. Since the virus particles are smaller than the expected pore size of the filters (MS-2 phage and rotavirus are 24[48] and 70 nm[49] in diameter, respectively, while the pore size is 100–500 nm[12]), the results suggest that the mechanism of virus removal is likely to be adsorption-driven. Virions can adsorb on cellulose-based materials[50], with cellulose nitrate reported to remove virions that are much smaller than the filter pore size[51]. We hypothesize that the relatively slow flow rate and the large thickness of xylem filters could facilitate adsorption and removal of viruses.

**Technology translation**. Motivated by encouraging results from the lab-based studies, we conducted field studies to assess the ability of xylem filters to function with natural water and facilitate access to safe drinking water in resource-constrained settings. We focused on India, which has the highest water-borne illness mortality rate in the world with more than 160 million people lacking access to safe and reliable water[2,52]. In particular, we targeted low-income communities in urban slums (Delhi and Bengaluru) and rural villages (Uttarakhand). We assessed filter performance in the field, developed a functional prototype device through user-centric design, and examined aspects of social acceptance and user preferences to gauge the potential of xylem filters to lower existing barriers for HWT adoption.

Xylem filters made from gingko trees in US and those manufactured in India with indigenous *Pinus roxburghii* (chir pine) using local resources for all fabrication steps such as cutting, hot water soaking, and dry preservation, were tested with water from natural springs, municipal taps, and tubewells (groundwater) (which were the primary sources of drinking water in Uttarakhand, Delhi, and Bengaluru, respectively; see Supplementary Table 2 for water quality information). Xylem filters with 4-cm diameter mounted by simply clamping the xylem filters in a tube (Fig. 5c) and operated under 1-m gravity head yielded peak flow rates exceeding 1 L/h and filtration capacities exceeding 10 L in most cases, with either cloth pre-filtration or cloth and GAC pre-filtration (Fig. 5d–h, see Supplementary Fig. 4a-d for variation of flow rates over filter lifetime). With cloth pre-filtration, xylem filter capacity ranged from ~40 L with groundwater to 12–30 L with turbid tap water. The benefits of adding a GAC pre-filtration module varied with water quality; GAC improved filter capacity from 38 to 102 L with groundwater and yielded a capacity of ~30 L with spring water, but did not improve xylem filter performance with tap water. No total or fecal coliform bacteria were detected in the filtrate for 5 out of the 6 xylem filters tested (3 filters operated with GAC and 3 filters operated without GAC) (Fig. 5h, Supplementary Fig. 4e–g). These results confirmed that xylem filters could remove coliform bacteria and function in realistic settings with replacement on a daily to weekly basis depending on the operating conditions.

Analogous to other membrane filters, xylem filters have to be housed in a device for HWT. A wide range of device configurations could be designed to suit different use cases, water quality, resources available, and user preferences. As an illustrative example, we built a functional, first-generation device prototype based on the feedback gathered through 600 semi-structured interviews and surveys, 53 focus group discussions, and 2 hands-on co-design workshops with over 1000 target users (see "Methods" for user study protocol). Product attributes desired by users revealed through these efforts included: (a) ease of operation (filling and extracting water from the device, replacement of filter cartridge, etc.), (b) low cost, and (c) aesthetic appeal. Combining these inputs with existing guidelines for fabricating household water filters[16], we developed a device consisting of a top container for feed water, a screw-on holder that houses a xylem filter and allows for easy replacement, and a bottom receptacle with a dispenser to collect filtered water and a cover to minimize the risk of recontamination. The device height was optimized such that users could fill the water in the top receptacle conveniently and the water head was sufficient to yield adequate flow rates (Fig. 6a–c, Supplementary Fig. 5a-f, see Supplementary Note 1 for other device configurations). Some challenges encountered during device design included obtaining a leak-proof seal between the holder and xylem filter due to irregularities on the wood surface, and preventing air entrapment in the tubing and filter holder that could obstruct flow. These challenges were overcome by cutting the wood at high speeds using a cold saw to obtain a smooth surface finish, using O-rings with appropriate compliance to conform to the wood surface, proper sizing of tubing and connectors to avoid bubble entrapment, and providing a vent in the holder for releasing any trapped air. This device showed 99.76 ± 0.41% rejection of 1-µm particles in lab studies and filtered 10 L of tap water in Delhi at flow rates >1 L/h (Fig. 6c). Flow rates can be improved further by clamping the filter on the side instead of the face, which prevents complete utilization of the peripheral filter cross-sectional area containing the xylem (the effective filtration area in the device was 7 cm$^2$; see Supplementary Fig. 5d-f for alternate holder configurations; see Supplementary Note 8 for details on resources required for manufacturing filters and filtration devices).

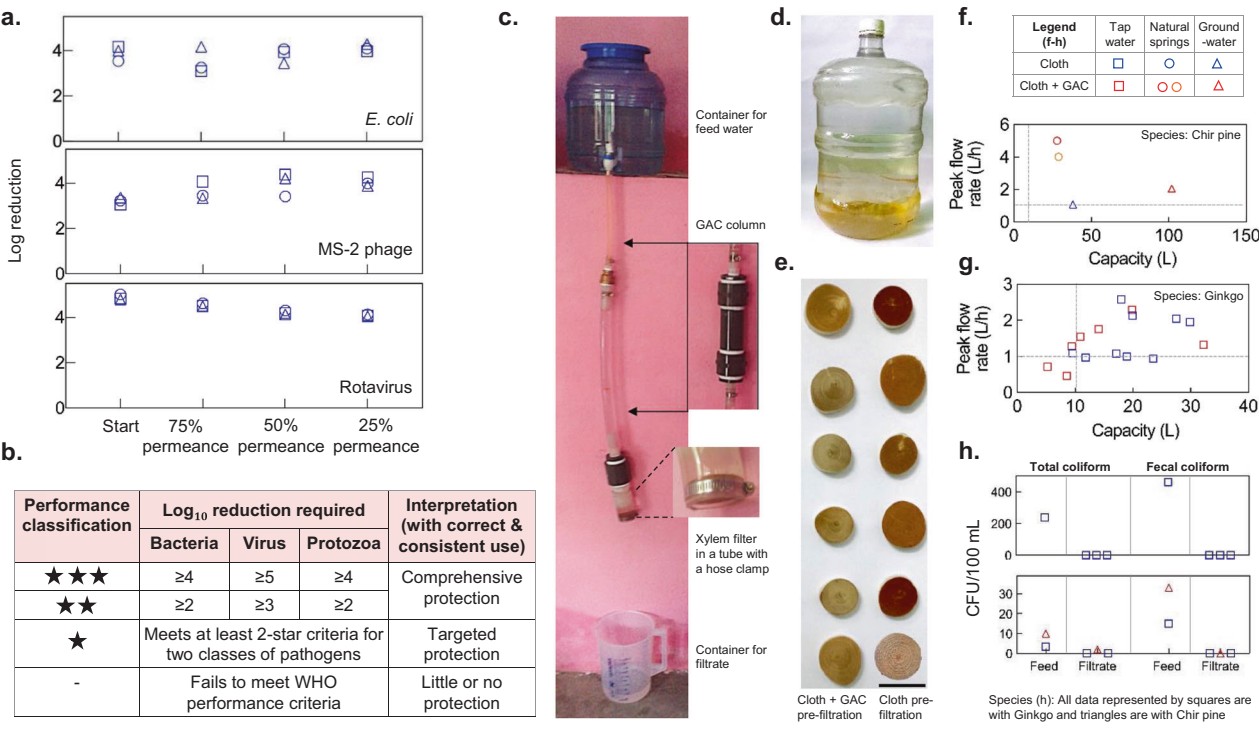

**Fig. 5 Microbiological and field performance. a** Microbial removal performance of xylem filters (ginkgo, 4-cm diameter, 0.375-inch thickness, no pre-filtration) when operated under 1.2-m gravity head with general test water containing *E. coli* (≥10^6 CFU/mL) and MS-2 phage (≥10^5 PFU/mL) dosed simultaneously, or rotavirus (≥10^5 PFU/mL). Rejection was measured at the start of filter operation and when permeance dropped to 75, 50, and 25% of initial permeance (see "Methods" for further details). Different symbols indicate different filters. **b** World Health Organization (WHO) scheme for classification of household water treatment technologies. **c** Field setup for testing filter performance. **d** Tap water sample (New Delhi, India) used for testing. **e** Xylem filters used with granular activated carbon (GAC) show reduced deposits after filtration with tap water. Scale bar, 4 cm. **f–h** Chir pine and ginkgo filters show peak flow rates and capacity exceeding 1 L/h and 10 L, respectively, indicated by dashed lines in (**f**) and (**g**), and absence of coliform in the filtered water. Legend is shown at the top. In **h**, chir pine and ginkgo filters were used for tap water and groundwater studies, respectively. All data are obtained with 4-cm diameter, 0.375-inch-thick filters operated under 1-m head (see "Methods" for details on field testing of filters). Each data point represents a different filter (one measurement per filter).

Although performance is important to the success of any technology, its adoption by the target population is critical to achieve impact. In addition to lack of awareness, the adoption of water treatment technologies in resource-constrained settings faces three main barriers: access, affordability, and acceptance. Xylem filters could lower these barriers in the following ways. First, the abundant availability of gymnosperms, local access to other raw materials required for filter fabrication (primarily alcohol for dry preservation), simplicity of the manufacturing process, and open access to this technology (filter fabrication has not been patented to facilitate adoption) could provide an opportunity for local manufacture of these filters, making them more accessible to local communities (see Supplementary Note 8 for a list of equipment that can be used in low-resource settings). Due to their low weight (<10 g) and volume (~13 cm^3), xylem filters could be shipped easily (even by post to remote locations) and stocked in local shops to facilitate access. Second, our preliminary cost estimation studies suggest that, in comparison to conventional filter cartridges that cost USD 5–10 and require replacement every 3–6 months, xylem filters could cost USD 0.06–0.10 and require replacement every 1–5 days with a comparable cost per liter water filtered (see Supplementary Note 7 for cost estimation details and Supplementary Fig. 6 for cost

comparison with currently available commercial filters in India). Such amortization of filter replacement costs could significantly lower the barrier to affordability for low-income households, where 'pay-as-you-go' or 'buying less, but more often' model is preferred over longer-term filter replacements[53]. In Indian urban slums where reverse-osmosis (RO) filtered water cans costing USD 0.28–0.56 per 20 L or government-operated water booths which provide RO water at USD 0.06–0.10 per 20 L are the only options, xylem filters could provide a viable HWT alternative for those who cannot afford or access these options easily. They could also facilitate sustained usage of HWT amongst those who are unable to purchase cans or fetch water from the booths on a regular basis to realize effective health outcomes. Furthermore, it could offer the opportunity to involve local suppliers, e.g., in small stores, in the daily distribution of filters. Finally, the traditional comfort associated with using wood for fuels and utensils could facilitate the social acceptance of xylem filters. During our field studies, 40% of the 300 survey respondents cited natural appeal and simplicity as the primary attributes they like about xylem filters, suggesting positive prospects for user reception (Supplementary Note 1).

The need for frequent filter replacement creates the risk of lack of user compliance and disruption of sustained usage, which

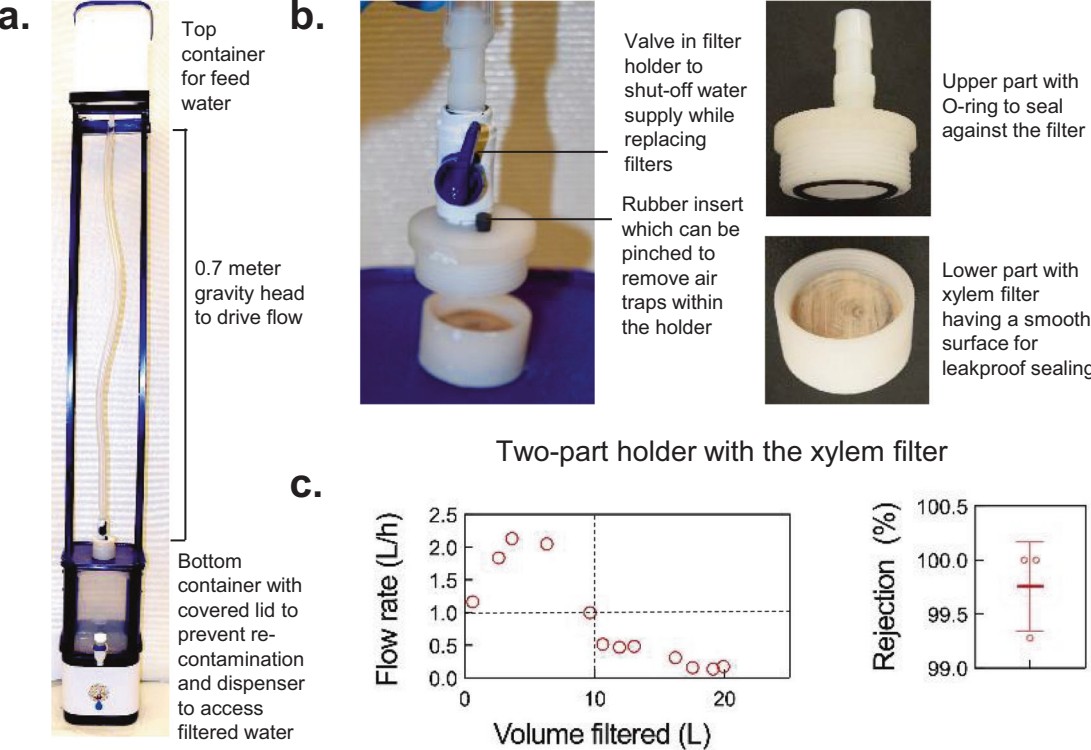

**Fig. 6 Xylem filter device prototype. a, b** Device prototype and its components. **c** Device flow rate with tap water in New Delhi and rejection of 1-μm particles measured in the laboratory are consistent with prior characterization of xylem filters. Filters were made from ginkgo of 4-cm diameter and 0.375-inch thickness. For the rejection performance, mean±s.d and individual data points are shown; $n = 3$ different filters.

could be exacerbated if pre-filtration units that require replacement at a different frequency than the xylem filters are used. Technological and product design improvements (such as engineering filters with longer lifetime, designing holders for easy filter replacement) or supply-related solutions (such as filters being available at stores that provide groceries or other regularly-purchased items) could help mitigate the risk of disruption of sustained usage, whereas the decrease in flow rate of filters with time mitigates the risk of lack of user compliance in replacing the filters. These efforts would nevertheless have to be complemented with behavior change interventions. The motivators for behavior change and thus, the nature of these interventions, would depend on the local social, cultural and environmental characteristics, such as water quality (whether water has visible coloration, odor, or poor taste), perceived health risk associated with water quality, education level of the population, etc.[3,54–59]. The influence of these characteristics on HWT adoption has been extensively studied in literature and this knowledge, in addition to field studies with user groups, could be leveraged to facilitate sustained adoption of xylem filters (see Supplementary Note 9 for factors that affect behavior change and sustained adoption of HWT and examples of behavior change interventions for xylem filters)[54–60].

## Discussion

This work provides new insights into gymnosperm xylem from the perspective of its use as a material for water filtration—namely, the interplay between permeance and rejection governed by percolation, the intriguing 'self-blocking' behavior arising from dissolution and re-deposition of hemicellulose, and elevated propensity for fouling in the presence of large organic molecules and dust. We leveraged these insights to develop engineering methods for preserving xylem filters in dry state, mitigating self-

blocking, and obtaining practically useful performance with different water qualities (see Table 1 for a summary of the effect of the key design, manufacturing, and operating parameters on filter performance). To demonstrate potential for practical utility and translation, filters were fabricated using locally available gymnosperms in India and filter performance was validated with natural water sources used for drinking. As an example of how xylem filters could be incorporated in filtration devices, a gravity-operated, functional device prototype for household drinking water treatment was developed using user-centered design approaches. Finally, evidence gathered from user research and preliminary cost estimation studies was used to show how xylem filters could potentially reduce the barriers of access, affordability, and social acceptance to serve as an attractive HWT option for low-income communities that are at the highest risk of water-borne diseases.

With >3-log removal for bacteria and virus (>4-log removal for rotavirus), xylem filters can provide 'comprehensive protection' against water-borne pathogens as per WHO's performance criteria for household water treatment technologies[48] and have potential for reducing the health burden of water-borne diseases (Fig. 7a). The contaminant removal ability of xylem filters could be improved further due to fouling (Supplementary Note 10), by stacking filters or using other approaches. Use of silver nano-particles in xylem to enhance the removal of bacteria and methylene blue (a commonly used industrial dye that causes water pollution) using xylem[10] and chemical modification of xylem surface for copper adsorption has been reported[9]. Incorporating suitable sorbents such as zeolites or ferrous oxide into pre-filtration modules could enhance removal of viruses, arsenic, or other pollutants[61,62]. More generally, xylem filters could be used synergistically with other water treatment methods, e.g., in conjunction with chlorine to remove protozoan cysts that are

**Table 1 Effect of filter geometry, fabrication process, and operating conditions/parameters on xylem filter performance (flow rate, volumetric capacity (or filter lifetime), and rejection).**

| Performance metrics | Xylem filter geometry | | Fabrication methods | | | Operating conditions/parameters | |
| --- | --- | --- | --- | --- | --- | --- | --- |
| | | | Dry preservation | | | | |
| | Sapwood area (A) | Thickness (t) | Alcohol treatment[a] | Stacking multiple filters in series[a] | Hot-water treatment[b] | With cloth or GAC pre-filtration | Pressure used for driving flow (P) |
| Flow rate (Q) | $Q \propto A$ | $Q \propto 1/t$, if tracheids are well-connected, else decreases more rapidly with thickness. | Increases | $Q \propto 1/n$, where n is the number of filters | No significant effect. | Generally increases; depends on water quality and pre-filtration process. | $Q \propto P$ |
| Volumetric capacity (V) | $V \propto A$ is expected for the same driving pressures | Unknown | Increases | Unknown; may be comparable to a single filter. | Increases if self-blocking is more dominant than fouling due to contaminants in water, else significant effect is not expected. | Generally increases; depends on water quality and pre-filtration process. | Unknown; depends on how pressure affects fouling. |
| Rejection: Filtration-based | No effect | Depends on particle size. Typically increases rapidly till thickness exceeds tracheid length; may not increase further if sapwood has resin canals, else theoretically expected to increase with length. | Increases | Increases; theoretically log-additive, if rejection of each filter is identical. | Depends on temperature and duration of treatment; may decrease rejection. Soaking at high temperatures or for long durations compromises rejection performance. | Depends on pre-filtration method used; GAC can augment performance by removal of chemical contaminants. | No significant effect expected. |
| Rejection: Adsorption-based (viruses) | No significant effect expected if permeate flux is constant | Expected to increase for the same driving pressure. $\tau \propto 1/t^2$, where $\tau$ is contact time. | Unknown | Expected to increase; theoretically log-additive, if rejection of each filter is identical. | Unknown | | Expected to decrease. $\tau \propto 1/P$, where $\tau$ is contact time. |

[a]Benchmarked against water-dried filters.
[b]Hot-water treated, alcohol-dried filters compared to filters that are just alcohol-dried.

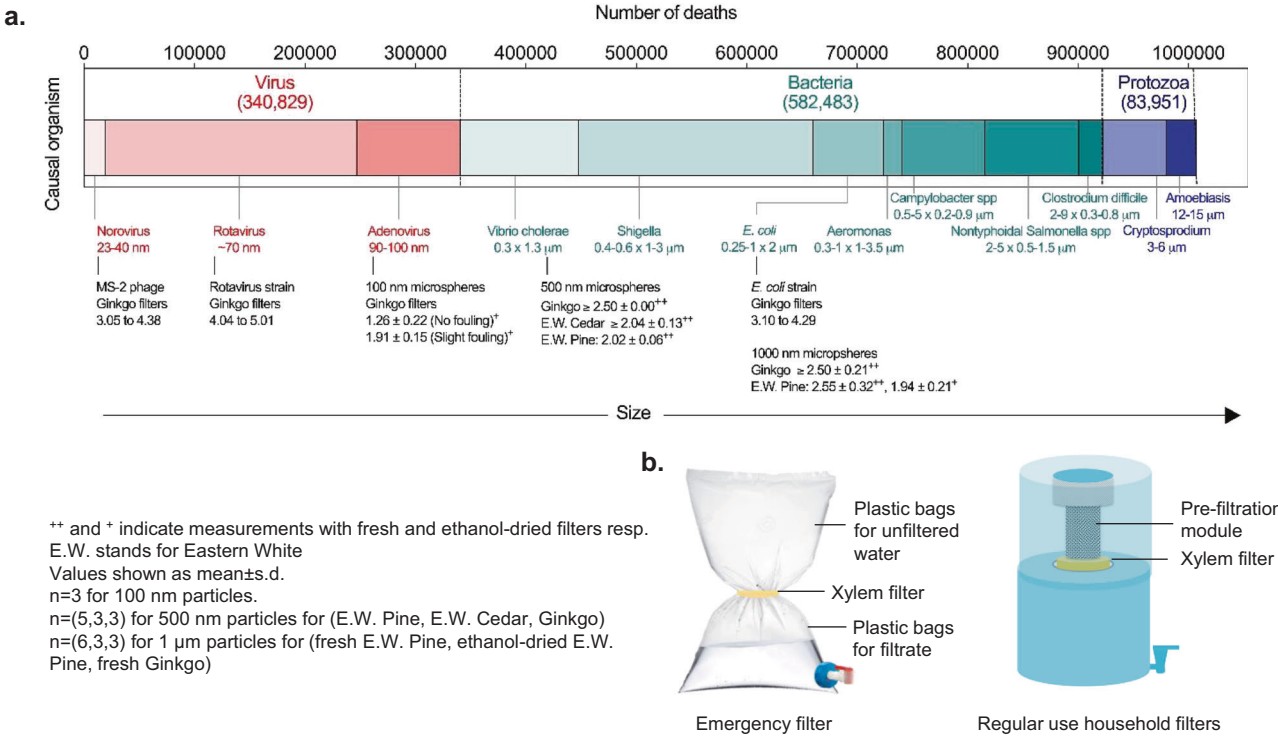

**Fig. 7 Potential health impact and applications of xylem filters. a** Comparison of diarrheal mortality burden due to different water-borne pathogens[2] arranged by size to the measured log rejection of particles of different size by single 0.375-inch-thick xylem filters (except for rejection experiments with 500-nm microspheres, where filter thickness was 0.25 inch) made from different tree species. Each *n* corresponds to a different filter. **b** Illustrations of possible configurations of xylem-based filtration devices.

relatively chlorine-resistant. Beyond gymnosperms, plants like primitive angiosperms that have short xylem conduits and nanoscale pit membrane pores and are not prone to aspiration[63], or ubiquitous bamboo nodes[64], could be explored for filtration applications.

Some limitations of xylem filters for household water treatment currently are (a) the higher replacement frequency of xylem in comparison to current filtration technologies could deter sustained usage, (b) heterogeneity in xylem structure within and across gymnosperm species could create variability in filter performance, and (c) a possibility that degradation of filters under certain conditions could cause their rejection performance to deteriorate and prevent them from getting blocked posing a health risk to users (timescales for degradation would depend on the feed water quality, type of wood used for manufacturing the filter, and local environmental factors and could be of the order of 1–2 weeks[65,66] or less since bacteria are directly seeded on pit membranes during filtration). Mitigation of these risks will require: (a) pilot studies of filter performance under different operational conditions (and measures such as increasing filter thickness or providing guidance as to when xylem filters are suitable for use if filter degradation is found to be an issue), (b) standardization of sources of wood and development of methods to test and assure quality control[67], (c) safety assessment of xylem filters for human use (e.g., assessing the filtrate for presence of organic compounds leached from xylem and evaluating their safety for human consumption, noting that cellulose, hemi-cellulose, lignin, and pectin occur naturally in many foods, and that plant material contributes to organic content in drinking water), (d) designing filter holders for maximal ease of replacement along with ready local availability of filters, and (e) behavior change interventions to facilitate sustained usage (see Supplementary Note 9 and 11 for more information on these topics). To

facilitate uptake and further advances in xylem filter technology, we have not patented this technology and have created design guide for fabricating xylem filters (Supplementary Note 11) and compendium enlisting the geographic availability, structural characteristics, and degradation properties of different gymnosperms (Supplementary Data 1). It is important to use non-toxic sapwood for making xylem filters (see Supplementary Note 11 for toxicity-related information).

Xylem filters present a HWT solution with unique characteristics (see Table 2 for comparison with other low-cost water treatment technologies). The availability of wood is not a bottleneck for scaling and large-scale dissemination—the sapwood used for making these filters is a low-grade by-product of the timber industry; we estimate that ~0.01% of the timber used in softwood global trade could be used to make a billion filters annually[68]. The compactness, lightweight, and long shelf life of xylem filters could enable easy shipping to remote locations and bulk storage for prolonged use. Due to the worldwide availability of gymnosperms and simplicity of the filter fabrication process, xylem filters offer potential for local manufacture of a wide variety of water treatment and other filtration products, ranging from compact filter pouches for use in emergencies to household filtration devices (Fig. 7b, see Supplementary Note 12 for further details on HWT for emergency use). This opens up the possibility for involvement of micro-entrepreneurs at a global scale[69], implementation of innovative business models, and engagement of local communities in filter distribution[70,71] and manufacture to raise awareness about importance of safe drinking water and encourage adoption (see Supplementary Note 13 for further details on potential avenues for engaging local communities and micro-enterprises). In addition, the characterization, modeling, and engineering of xylem filters is not limited to household water filters and has the potential to be applied to different filtration needs, such as for disaster and

**Table 2 Comparison of xylem filters with other low-cost water treatment technologies.**

| Technology | Level of protection | Advantages | Limitations |
|---|---|---|---|
| Chlorination[72] | Targeted (bacteria and viruses only) | • Portable, lightweight<br>• Simple to use, no maintenance | • Less effective in inorganic-rich or turbid water<br>• Unpleasant taste and odor<br>• Dosage depends on water quality<br>• Harmful organic by-products |
| Solar disinfection[72] | Targeted to comprehensive (bacteria, virus, protozoa) | • Little/no maintenance<br>• Minimal chance of recontamination<br>• Simple to use | • Poor efficacy with turbid water<br>• Dependent on weather<br>• Volume to treat depends on availability of intact container<br>• Long treatment time |
| Ceramic filters[72] | Targeted (bacteria and protozoa only) | • Local production<br>• Minimal chance of recontamination | • Performance heterogeneity due to manufacturing variability and susceptibility to cracks<br>• Requires regular cleaning<br>• Difficult to transport<br>• Low flow rate: 1–3 L/h |
| Flocculation-disinfection[72,73] | Comprehensive (bacteria, virus and protozoa) | • Residual protection against recontamination<br>• Easy to transport (typically available as sachets)<br>• Reduction of some heavy metals and particle-associated pesticides | • Need for multiple steps and additional user support<br>• Can have a negative effect on odor and taste<br>• Requires reliable supply chain |
| Xylem filters | Comprehensive (>3-log removal of bacteria and virus) | • Biodegradable<br>• Lightweight<br>• Minimal chance of recontamination<br>• Local production | • Frequency of replacement is relatively high (maintenance concerns)<br>• Needs pre-filtration or large filter sizes with water having high turbidity or organic content<br>• Low flow rate: 1–3 L/h |

emergency use (Fig. 7b), microfiltration for assessment of water quality, and as an alternative to synthetic microfiltration membranes for some applications.

## Methods

**Cutting of xylem filters from branches.** Straight sections of branches of appropriate diameters were excised from trees (Eastern white pine, Eastern white cedar, ginkgo or chir pine) using a hand pruner and immediately immersed in water to prevent drying (in winter, foliate branches were excised and stored indoors at room temperature for 2–3 days with one end immersed in water to restore xylem activity and improve permeance). Branches without leaves or sections near branch attachments were avoided due to their low hydraulic conductivity[74,75]. The natural direction of water flow in the branch was marked with a sharpie and the same flow direction was maintained in experiments. The branches were subsequently cut into smaller sections with the desired filter thickness using a band saw. The outer bark and the cambium were peeled off by hand to obtain a fresh xylem filter. The filters were stored in deionized (DI) water for up to 8 h before experiments or subsequent processing steps.

**Water flow rate and volumetric capacity measurements.** Xylem filters were mounted in PVC (polyvinyl chloride) tubes with 0.375-, 1.5-, and 1.875-inch inner diameters (McMaster-Carr, part numbers 5233K63, 5233K77, and 53945K22) for 1-, 4-, and 5-cm-diameter filters, respectively. The tubing was connected to a nitrogen tank and regulator set at 10 psi (69 kPa) pressure, or to a water-filled tank providing a 1-m head (specified in figure captions). For nitrogen gas-driven measurement of permeance of ~1-cm-diameter filters, the tube was filled with 5 mL of DI water and the time required for the filter to process the water was used to obtain the flow rate. In gravity-driven experiments to measure permeance and capacity of 4–5-cm-diameter filters, the volume of fluid processed by the filters was monitored with time by collecting it in a cylinder with 20 mL graduations to obtain flow rates. For experiments with DI water containing $Ca^{2+}$ and $K^+$ ions, calcium chloride dehydrate (ACS reagent grade, CAS number 10035-04-8, Sigma-Aldrich) and potassium chloride (ACS reagent grade, CAS number 7447-40-71, Sigma-Aldrich) were dissolved in DI water at the desired concentrations. For experiments involving intermittent filter operation, filters made from ginkgo (4-cm diameter, 0.375-inch thickness) were operated for a duration of 7 days while processing 5 L of DI water per day. The filters were mounted in the device (shown in Supplementary Fig. 5a) and operated under a gravitational pressure head of 1 m with 5 L of DI water added into the device every day. The filters stopped producing water after the 5 L water was filtered, till 5 L of water was again added on the following day. The time required to filter the 5 L of water was measured and used to calculate the average permeance to filter 5 L of water on a given day. Permeance was calculated by normalizing the flow rate by the filter cross-section area and the driving pressure. The permeate processed by the filter per unit area was calculated by normalizing the volume of the fluid processed by the filter by the filter cross-section area.

**SEM imaging.** Xylem structure was visualized using scanning electron microscopy (SEM, Zeiss Merlin High Resolution). Samples were prepared by cutting thin slices of dried filters in the direction of the flow (longitudinally) using a razor blade. No coating was applied on the samples. The slices were imaged under an extra high tension (EHT) voltage of 0.7–1.2 kV and a probe current of 75–90 pA. The depth-of-resolution mode was used for acquiring images. The images in Fig. 1c were obtained from ethanol-dried xylem filters made from ginkgo (1-cm diameter and 0.25-inch thickness) after filtration of ~5 mL DI water spiked with 500-nm carboxylate-modified latex microspheres with yellow-green fluorescence (Thermo Fischer Scientific) at a concentration of $10^9$ particles/mL.

**Rejection of fluorescent microspheres.** Carboxylate-modified latex microspheres with yellow-green fluorescence (Thermo Fischer Scientific) with diameters of 100 nm, 500 nm, and 1 μm were used. The microspheres were sonicated for 1 min using a VWR Ultrasonic Mixer at room temperature (25 °C), suspended in DI water at $10^6$ mL$^{-1}$ concentration, and vortexed for 1–2 min to obtain a homogenous feed solution. The feed solution was filtered through xylem filters mounted in PVC tubing at 10 psi pressure as described in Methods sub-section "Water flow rate and volumetric capacity measurements". The filtrate was collected in clean glass vials; a feed volume 3–4× of the filter volume was used.

10 μL of the feed or filtrate solutions were introduced into a hemacytometer (Neubauer-modified C-Chip, InCyto) and imaged with a Nikon TE2000U epifluorescence microscope using a ×20 objective for the 1-μm microspheres or ×40 objective for the 500- and 100-nm microspheres. Due to the limited depth of focus of the ×40 objective, images were acquired (by Andor iXon camera) with the focus at the mid-plane of the C-chip, and the particle count in this plane was used to estimate the concentration of the 100-nm microspheres. Particle counts over three draws of the feed/filtrate solution were averaged to calculate rejection as follows, where $c_{feed}$ and $c_{filtrate}$ are microsphere counts in the feed and filtrate solutions, respectively.

$$\text{Rejection} = \frac{(C_{feed} - C_{filtrate})}{C_{feed}} \tag{1}$$

$$\text{Log rejection} = -\log_{10}(1 - \text{Rejection}) \tag{2}$$

Rejection tests with the xylem filter device were conducted by mounting the xylem filter in holder (Fig. 6b), connecting the holder to PVC tubing, filling the tubing and the holder with a fluospheres solution (concentration of $\sim10^6\,\mathrm{mL}^{-1}$) of volume $3-4\times$ that of the filter and pressurizing the fluid through the holder at 10 psi gas pressure. The filtrate was collected in a glass beaker. The particle count in the feed and the filtrate were evaluated as described in Methods sub-section "Water flow rate and volumetric capacity measurements".

As the mechanism of bacterial removal in xylem filters is physical sieving[7], we expected the contaminant size to be the key determinant of the filtration characteristics and used 1-µm latex microspheres as surrogates for *E. coli*. Latex microspheres have been used as proxies to study the transport of *E. coli* in porous media due to the similarity in size and zeta potential[76,77].

**Treatment of xylem filters.** For hot-water treatment, filters were soaked in covered glass containers filled with deionized water maintained at 60 °C using a temperature-controlled water bath for 1 h (tap water and a simple stove were used for filters fabricated in India). A wood to water volume ratio of 1:20 was used. Subsequently, filters were immersed in room-temperature DI water 4–5 min before subsequent processing. 200-proof ethanol (CAS number 64-17-5, Koptec) or iso-propyl alcohol (ACS reagent grade, CAS number 67-63-0, Macron Chemicals) were used for alcohol treatment; ethanol with 99.9% purity (CAS number 64-17-5, Merck) was used for field studies in India. Ethanol–water mixtures of different concentrations were prepared by adding DI water to 200-proof ethanol (CAS number 64-17-5, Koptec). Xylem filters (freshly cut or hot-water soaked) of 1-cm and 4–5-cm diameter were treated by flushing and soaking, respectively (filter to ethanol/isopropyl alcohol volume ratio of 1:6–8). For flushing, 10 psi gas pressure was used to drive the desired volume of ethanol/isopropyl alcohol loaded in plastic tubing in which xylem filters were mounted, similar to permeance measurements described in Methods sub-section "Water flow rate and volumetric capacity measurements". For soaking, filters were placed edge-on in appropriately sized glass vessels and completely immersed in ethanol for 24–48 h. Most of the filters (~90%) were soaked for more than 24 h, in which case the ethanol in the vessel was typically replaced with fresh ethanol after 24 h. Filters manufactured in the field were treated with ethanol only by soaking.

**Drying of xylem filters.** In laboratory experiments, all filters (soaked in water or in ethanol/isopropyl alcohol) were dried by placing edge-on on aluminum foil in an oven (VWR 1410 vacuum oven) at atmospheric pressure and 45 °C. To ensure complete drying, the filter weights were monitored (using Mettler Toledo AL104 scale) till they stabilized. Water-dried filters with 1-cm diameter and 0.25-, 0.5-, and 1.5-inch thickness required around 10, 25, and 50 h for complete drying. Ethanol treatment resulted in faster drying (~6–8 h for 1-cm diameter, 0.25-inch-thick filters). In field studies, filters were dried at 45 °C and atmospheric pressure for equivalent durations (at least 48 h) using an oven (Bajaj Majesty 1603 TSS Oven Toaster Grill) but weights were not monitored due to lack of access to appropriate instruments.

**Air flow measurements.** Air flow rates were measured using the setup illustrated in Supplementary Fig. 1d (inset). Xylem filters with 1-cm diameter were mounted in a PVC tube (0.375-inch inner diameter, McMaster-Carr part number 5233K63) secured with a hose clamp and supplied with nitrogen at 1 psi using a nitrogen cylinder and pressure regulator. The gas flowing through the filter was collected in an inverted 250-mL beaker fully immersed in water, and the time required for the gas to displace water from the inverted beaker was measured to obtain the flow rate (and permeance).

**Rewetting of water-dried filters with ethanol.** To obtain data shown in Supplementary Fig. 1e, freshly cut xylem filters with 1-cm diameter, 0.25-inch thickness were mounted in a PVC tube (0.375-inch inner diameter, part number 5233K63, McMaster-Carr), secured using a hose clamp and flushed with 5 mL of DI water at 10 psi nitrogen gas pressure. Flow rates were recorded and filters were dried in an oven at 45 °C and atmospheric pressure. Dried filters were flushed with 5 mL of 200-proof ethanol (CAS number 64-17-5, Koptec) and re-dried in the oven at 45 °C and atmospheric pressure. The flow rates of the dried filters were measured with DI water as described in Methods sub-section "Water flow rate and volumetric capacity measurements". The ratio of flow rate after drying to that before was used to calculate permeance recovery.

**Atomic force microscopy (AFM).** DI water was filtered through Eastern white pine filter (1-cm diameter, 0.375-inch thickness) at a gas pressure of 10 psi using the method described in Methods sub-section "Water flow rate and volumetric capacity measurements". The filtrate was collected in a clean glass vial, dispersed and dried on a substrate, and analyzed under an Asylum-2 MFP-3D Coax AFM.

**FTIR.** Eastern white pine xylem filter (1-cm diameter, 0.375-inch thickness) was placed in a glass vial filled with 20 mL deionized water maintained at 60 °C for 1 h. The filter was removed and the residual water was evaporated until the volume

reduced to 5 mL to concentrate any extracts from the filter. The concentrate was analyzed using a Perkin Elmer FTIR spectrometer.

**Preparation of synthetic test waters.** GTW and CTW were prepared by adding the WHO-prescribed dosage of sea salts (Sigma-Aldrich, product number S9883), sodium bicarbonate (BioXtra, 99.5–100.5%, CAS number 144-55-8 procured from Sigma-Aldrich), and tannic acid (ACS reagent grade, CAS number 1401-55-4 obtained from Sigma-Aldrich) or humic acid (50–60%, CAS number 68131-04-4 procured from Alfa Aesar) in DI water (see Supplementary Fig. 3a for dosages)[37]. To achieve turbidity of 40 NTU in CTW, Arizona Test Dust (ISO 12103-1, A2 fine test dust obtained from Powder Technologies Inc.) was added to DI water at a concentration of 70 mg/L based on calibration reported in literature[78].

**Susceptibility of xylem filter to different contaminants.** To determine the susceptibility of volumetric capacity to different organics, DI water having same alkalinity, salinity, and turbidity as CTW (1.5 g/L sea salts, 100–120 mg/L sodium bicarbonate, 70 mg/L Arizona test dust) and varying dosages (0, 1.5, 15, or 150 mg/L) of tannic or humic acid was used. To evaluate the effect of dust on capacity, DI water with the same alkalinity and salinity as CTW (1.5 g/L sea salts and 100–120 mg/L sodium bicarbonate), 15 mg/L of humic or tannic acid, and varying dosages of dust were used (0 and 70 mg/L). All measurements were performed with Eastern white pine filters (1-cm diameter and 0.375-inch thickness) operated under a 1-m gravity head. Volumetric capacity was measured as per the protocol specified in Methods sub-section "Water flow rate and volumetric capacity measurements".

**Cloth pre-filtration.** A sediment pre-filter manufactured by Hindustan Unilever Limited (PureIt microfiber mesh) was used for removing visible dust particles. Feed water was first passed through the pre-filter, collected in a separated beaker, and subsequently poured into the container connected to the xylem filter for further filtration.

**Activated carbon pre-filtration.** Granular Activated Carbon (GAC, Activated Carbon Corporation) of $12 \times 40$ grain was sieved in a fume hood to obtain $30 \times 40$ grain size. Prior to usage, the GAC was soaked in tap water overnight and rinsed thoroughly under running tap water to remove dust that otherwise clogged the filters. The GAC column was fabricated by assembling off-the shelf components. The GAC was packed in a 5-cm diameter, 15-cm-long pipe threaded at both ends (MPT (male pipe thread) × MPT, product number 4677T45, procured from McMaster-Carr). A circular mesh with 5-cm diameter was cut from the PureIt microfiber mesh manufactured by Hindustan Unilever Limited and fixed to the lower end of the pipe using epoxy (3 M Scotch-Weld Epoxy Adhesive DP100 Plus) to keep the GAC granules in place. The top end of the pipe was attached to 0.375-inch diameter PVC tubing (McMaster-Carr, part numbers 5233K63) that was connected to the feed water container through a set of connectors, which consisted of a 2-inch coupling (FPT (female pipe thread) × FPT, part number 9499, Metropolitan Pipe and Supply), a 2-inch × 0.75-inch reducer (MPT × FPT, part number 9530, Metropolitan Pipe and Supply), and a 0.75-inch × 0.375-inch reducer (McMaster-Carr, product number 5372K154). The lower end of the pipe was threaded onto a cap (McMaster-Carr, part number 4880K807) and connected to a standard port valve (McMaster-Carr, part number 45975K32) to regulate the flow rate (and thus the GAC contact time). The valve was connected to the xylem filter. During experiments, care was taken to avoid formation of air bubbles. If formed, air bubbles were removed by squeezing the tubing. The humic acid concentration in the feed and the filtrate was measured by UV–Vis spectrometry (Agilent Cary 60). The absorbance was averaged over 300–500 nm wavelengths and absorbance at 700 nm was subtracted. Calibration curves were generated using different concentrations of humic acid in DI water (curves were linear). Dust was not used in experiments involving measurement of humic acid concentration due to interference with the UV–Vis measurement. DI water with 1.5 g/L of sea salts, 120 mg/L of sodium bicarbonate, and ~10 mg/L of humic acid was used as the test solution for comparing the performance of different GACs (Supplementary Fig. 3d).

**Testing xylem filters in field studies.** Experiments in India were conducted with locally fabricated filters made from chir pine (*Pinus roxburghii*), as well as ginkgo (*Ginkgo biloba*) and Eastern white pine (*Pinus strobus*) filters fabricated in Massachusetts. Filters made in Massachusetts were transported to India by air in zip-locked pouches containing desiccant (Silica gel manufactured by Dry Packs, serial number 1203-61) to prevent condensation of moisture and blocking. Water was collected from different water sources in clean 20 L plastic cans. Before starting the experiment, the tube, holder, and device container surfaces were wiped with cotton soaked in isopropyl alcohol to prevent contamination of the feed water. The feed was introduced after 5–10 min to allow sufficient time for isopropyl alcohol evaporation. For microbiological testing, the feed and filtrate were collected in glass and plastic bottles that were previously disinfected in boiling hot water for 15–20 min. The filtrate for microbiological testing was collected during the filtration process as follows. Prior to collecting the filtrate for microbiological analysis, the bottom surface of the xylem filter, the hose clamp, and lower end of the tubing were wiped with cotton soaked in isopropyl alcohol (while filtration continued). Collection of the filtered sample was started after 10 min to allow sufficient time for any residual

isopropyl alcohol to be flushed or evaporated. The microbiological tests were conducted by certified third-party labs (Uttarakhand Jal Sansthan for all the tests conducted in Uttarakhand (Supplementary Fig. 4e–g) and Delhi Analytical Research Laboratory for tests conducted in Delhi (Fig. 5h)). The methods used for sampling and microbiological examination conformed to Indian Standards, IS:1622 (1981) for data in Fig. 5h and American Public Health Association (APHA) 22nd edition for data in Supplementary Fig. 4e–g.

**Construction of xylem filters from trunks.** Eastern white pine (*Pinus strobus*) trunks for constructing xylem filters were procured from a local sawmill in Essex, MA. Due to logistical issues, logs were used ~2 weeks after the tree was felled (partial drying of the wood within this duration could have compromised rejection). The bark was removed using a band saw. $10 \times 4\text{-cm}^2$ sections were cut from the sapwood (lighter in color than the heartwood) to obtain xylem filters. Hot-water soaking and ethanol treatment were performed as described in Methods subsection "Treatment of xylem filters". Rejection performance of these filters was tested as described in Methods sub-section "Rejection of fluorescent microspheres" on cylindrical 1-cm diameter filters cored out from the dried filters using a hammer-driven small hole punch with 0.375-inch hole diameter (McMaster-Carr, part number 3424A25). To avoid leaks, the xylem filter was sealed in the tubing using epoxy.

**Construction of Fig. 1d.** The number of diarrheal deaths for the year 2016 was taken as is from the Global Health Observatory data repository managed by the World Health Organization (WHO)[79]. The total deaths corresponding to each of the six WHO world regions (African region, region of the Americas, South-East Asia region, European region, Eastern Mediterranean region, and Western Pacific region)[80] were calculated by adding the deaths associated with the countries in that region. The number of diarrheal deaths was rounded to the nearest hundreds for ease of reading. The numbers were color-coded on the world map depicting the WHO world regions using Adobe Photoshop (the world map was obtained from an open access website, 'Our World in Data' (URL: https://ourworldindata.org/world-region-map-definitions, Product name: 'World Map Region Definition' > 'World Health Organization', Author: Diana Beltekien) and re-colored using Adobe Photoshop). The global distribution of wild conifers reported in literature[17] was re-sketched over this map in Microsoft Power Point (Version 16.37).

**Statistical information and reproducibility.** All graphs were plotted using GraphPad Prism (Version 8.4.3). The error bars in all figures represent positive and negative standard deviations from the mean and calculated using in-built functions in GraphPad Prism v8. The *n* values for the data are included in the figure captions. The *p*-values for Fig. 2f and 2h were determined by performing two-sample, two-tailed *t*-tests on the data using Microsoft Excel (Version 16.37). The sample sets were considered unpaired with equal variance. Homogeneity in variance was determined using Levene's test. For micrographs shown in Figs. 1b, 2g, 3b, and 4e, experiments were repeated independently 3, 5, 3, and 3 times, respectively, with similar results.

**Field studies.** Information on user needs and preferences for water filtration devices and feedback on xylem filtration device prototypes was gathered through 600 individual semi-structured interviews, 53 focus group discussions, and 2 hands-on co-design workshops with over 1000 potential users. The research sample for the field studies included low-income rural households in the mountainous state of Uttarakhand, India, and the urban slum households from Bengaluru and Delhi and other stakeholders in the water filter supply chain, including, filter vendors, manufacturers, NGO staff, and local health officials. The 'National Ethical Guidelines for Biomedical and Health Research Involving Human Participants' guidelines published by the Indian Council of Medical Research (ICMR) were followed while conducting user research in India. These guidelines suggest that social and behavioral research for health applications should be approved by the ethics committee for the researchers' institution, which in our case would be the Massachusetts Institute of Technology Committee on the Use of Human Subjects (MIT COUHES). All human-subjects research procedures were approved by MIT COUHES under protocol 1612798762. Informed consent of the participants was obtained prior to data collection.

Different sampling strategies were followed for different data collection methods. Potential segments with potentially different needs were stratified based on a variety of factors including the geography, types of water sources, proximity to town, and heterogeneity of the population in terms of water-related practices. Villages were selected based on these criteria with the assistance of local partner NGOs. For semi-structured interviews, simple random sampling was used within the selected villages. In general, 20–30 subjects were interviewed for each segment[81]. For focus group discussions and design workshops, a combination of non-probability sampling techniques such as convenience sampling and snowball sampling techniques were used. For key informant interviews, a purposeful sampling strategy was used to select participants. Data saturation was also considered in the studies. The homogeneity of the population with regards to water usage practices was assessed with the help of local experts and data was considered saturated when no new information or themes were observed. Pen and paper were

used to record the data. Photos were taken and voice recorders were also used with participant consent. A translator was present during data collection. Researchers were not blind to experimental condition and study hypothesis. No data points were omitted while performing data analysis.

**Microbiological performance as per WHO protocol.** The microbiological performance of xylem filters was tested by a third-party laboratory (Quality Filter Testing (QFT) Laboratory LLC, 1041 Glassboro Road Suite E-4, Williamstown, New Jersey 08094, USA; ISO 17025 laboratory certified by the International Association of Plumbing and Mechanical Officials (IAPMO) to test water filters as per National Sanitation Foundation (NSF)/American National Standards Institute (ANSI) standards). Xylem filters (4-cm diameter, 0.375-inch thickness) made from ginkgo (*Ginkgo biloba*) wood using hot water treatment and ethanol-based dry preservation at Massachusetts Institute of Technology (MIT), USA, and stored for 2 years were mounted in 1.5-inch diameter PVC (polyvinyl chloride) tubes (McMaster-Carr, part number 5233K77) and secured in place using hose clamps, and were shipped to the third-party laboratory. In the third-party laboratory, the tubing was connected to a water-filled tank providing a 1.2-m head. *E. coli* (ATCC 11229) and MS-2 phage (ATCC-15597-B1, with host organism *E. coli* ATCC-15597) were dosed in GTW at concentrations $\geq 10^6$ CFU/mL and $\geq 10^5$ PFU/mL, respectively, as per the 'WHO International Scheme to Evaluate Household Water Treatment Technologies'[48]. For rotavirus removal, rotavirus strain SA-11 (ATCC-VR-899) was spiked in GTW at concentration $\geq 10^4$ PFU/mL (as specified by NSF Protocol P231 for Microbiological Water Purifiers)[46]. The removal of bacteria and virus was tested at the start of filter operation and when permeance declined to 75, 50, and 25% of the initial value. Flow rates were measured by monitoring the volume of fluid processed by the filters over time. After the first sample at the start of filter operation, dust was added to achieve accelerated clogging of the filter (dust concentration raised turbidity to 120 NTU and 130 NTU for *E. coli*/MS-2 phage and rotavirus, respectively, which is above what is required for CTW). Prior to collecting the filtrate for microbiological analysis, the bottom surface of the xylem filter, the hose clamp, and lower end of the tubing were wiped with cotton soaked in ethanol (while filtration continued). Collection of the filtered sample was started after 10 min to allow sufficient time for any residual ethanol to be flushed or evaporated. *E. coli* and MS-2 phage were assayed using Standard Method 9222 and 9224 published by American Public Health Association (APHA) for the Examination of Water and Wastewater. The log-reduction values were determined by measuring the bacteria/virus count in the feed solution ($C_{feed}$), and filtrate ($C_{filtrate}$) using the following equation:

$$\text{log removal} = -\log_{10} \frac{C_{filtrate}}{C_{feed}} \tag{3}$$

This method was used for experiments with results shown in Fig. 5a.

**Reporting summary.** Further information on research design is available in the Nature Research Reporting Summary linked to this article.

## Data availability
All data that support the findings of this study are available in the main text, figures, and Supplementary information. Field study results are presented in aggregate to protect privacy of survey respondents. Data for number of diarrheal deaths shown in Fig. 1d was obtained from the Global Health Observatory data repository managed by the World Health Organization (URL (accessed on 14 June 2019): https://apps.who.int/gho/data/view.main.INADEQUATEWATERv?lang=en, Dataset title: 'Number of diarrhea deaths from inadequate water (2016)' within the section 'Burden of disease from inadequate water in low- and middle-income countries'.

## Code availability
Codes used in the percolation model in this study are available from the corresponding author on reasonable request.

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

## Acknowledgements

The authors are grateful for support from the Abdul Latif Jameel Water and Food Systems Lab through the J-WAFS Solutions Program sponsored by Community Jameel. We also thank the MIT Tata Center for Technology and Design, Rasikabhai L. Meswani Fellowship, and J-WAFS Grant for Water and Food Projects in India for funding this research. We thank Himmotthan Society, People's Science Institute, Shramyog, and Pan Himalayan Grassroots in Uttarakhand, Essmart in Bengaluru, Surinder Nagar, R. Akhilesh Kumar, and translators for facilitating the field studies in India. We thank MIT Facilities, particularly, Daniel Caterino, Sogna Scott, and Todd Gillan for providing access to wood samples. This work made use of the MRSEC Shared Experimental Facilities at MIT, supported by the National Science Foundation under award number DMR-1419807 and was in particular, aided by Research Specialist, Patrick Boisvert. We would also like to thank Prof. Daniel Frey and Dr. Michael Bono from Department of Mechanical Engineering at MIT, Dr. Eric Verploegen from D-Lab, Dr. Chintan Vaishnav from MIT Sloan School of Management, Dr. James K Wheeler from UCSC, and Prof. Jonathan Schilling from the University of Minnesota for their insightful suggestions. We also thank the many other people who contributed to various aspects of the work on xylem filters.

## Author contributions

K.R. performed lab and field experiments, data analysis, simulations for percolation theory, and cost estimation. L.W. performed FTIR and AFM measurements. K.R. and R.K. conceived and designed the experiments. R.K. supervised the studies. M.H. and K.L. designed user studies, with inputs from all authors. M.H., K.R., and A.P.A. performed user studies and analyzed data from user studies. K.L. and A.S. supervised user studies. K.R. and A.P.A. designed and fabricated filter devices. K.R. and R.K. wrote the paper with inputs from all authors.

## Competing interests

The authors declare no competing (financial/non-financial) interests. The authors have not filed patent applications on xylem filters and the technology and filter designs are open-source. R.K. is involved with a start-up company related to water. R.K. currently does not currently have financial interest in the company; the company does not have any rights to xylem filters and is not working on xylem filters, although it is possible that this may change in the future.
