## [Peer Review File · Nature Communications]

REVIEWER COMMENTS

Reviewer #1 (Remarks to the Author):

Dear Authors

Overall, this is an interesting manuscript providing some scientific data and local information from India about a proposed new household filter - the xylem filters.

The authors should consider:

- Reviewing the manuscript and changing language such as "The xylem filters showed excellent removal of total and fecal coliform, both with and without GAC" in a more scientifically rigorous and unbiased manner.
- Referencing the extensive literature on adoption of household water filtration; and discussing the extensive behavior change that would be needed to work with users to use a filter that first requires pre-treatment with GAC or cloth, then requires daily-to-weekly changing of a wood filter, and only produces ~1 Liter/hour.
- Describing with the WHO methods for testing the efficacy of household water filters (which are published) were not used, and if microspheres are appropriate for reporting data. In particular, viral results are presented in Table 2, but it is not clear where those results came from?
- Discussing whether 20 Liters per household per day per filter is an appropriate long-term target for water use.
- Discussing if local ethics approval was obtained in India before the human subjects work was conducted.
- Clearly linking the methods described to scientific results, and unbiased discussion.

Reviewer #2 (Remarks to the Author):

This paper reported on the evaluation of xylem filters for water treatment applications. The paper develops guidelines for use of the filters in practical applications and evaluates the filters in a treatment system that results from participatory co-design. Overall, I judge the manuscript to be well-written and the results are interesting. They take items that are very fundamental and elucidate their influence on the end practical implications. However, there are some items which need to be addressed before this work would be considered for publication in a journal like Nature Communications.

1. Justify the usage of 1 μm microspheres as a proxy for *E. coli*. Justify that the surface properties are the same.
2. One of the big challenge the authors state is that the filters can degrade when dry. They conduct a treatment to minimize this factor for new filters. However, in many developing world situations, it would not be expected that the filters will be used continuously. What is this implication on the permanence of the filters?
3. Please confirm that the type of ethanol being used is safe for food use.
4. You note that the cost could be broken into small amounts and filters replaced every day (or few days). In this point, you compare with other filters. You should note here that the xylem filters may end up being more expensive per unit water produced, as shown in supplementary figure 6.
5. You note a few days replacement time. I know you noted that this could be good from a user perspective since the cost can be spread out. However, this type of operation can often lead to equipment falling into disrepair. I was unable to find many academic studies showing these results, however regular maintenance of simple things, like water containers, is even shown to be infrequently done. Please review and discuss potential downsides of needing to replace the filters frequently. <https://link.springer.com/article/10.1186/s12199-019-0799-3>
6. You note the frequency of maintenance noted for the filters, but the frequency of maintenance for the cloth or GAC filter are not noted.
7. Please be sure you discuss any further issues associated with the proposed technology.
8. You discuss the local manufacture of the filters, but do not tie this back to any of the issues you encountered in your prototyping. Do you feel any of these will be barriers towards local manufacturing of the filters.
9. In figure 1d, you give very high accuracy on the number of diarrheal deaths. I doubt you can know the breakdown with this level of accuracy.
10. In figure 2h, something is fishy with your rejection numbers. Shouldn't these be in the high 90s? Please review and revise accordingly.
11. Figure 3, there are hash lines in multiple figures and it's not clear what they represent.
12. Figure 4b, why does the capacity not decrease with tannic acid. If you aren't going to explain, then why is it there? Does it add some insight?
13. Figure 4a and 4f, why is there a reason that the axes are different here? Aren't they essentially a scaled version of each other. Also, why is it 5cm one spot and 4cm another spot.
14. You discuss on how 0.25" is the right filter thickness, but then your prototype uses 0.375" thick filters. Please explain and justify why this thickness should be used?

Reviewer #3 (Remarks to the Author):

The authors have presented a very thorough characterization of gymnosperm xylem based filtration system and its engineering use in limited resource communities through successful field trials in India. Some of the key fluid transport issues have been addressed, particularly the "self-blocking" characteristics of these filters and an innovative solution is proposed to overcome it. It has been

demonstrated that the proposed system can work well with different types of contaminated water. Also, utilization of a pre-filtration stage is interesting. The manuscript is very well-written, easy to understand for general audience and allowing an "open source" design architecture for the filter system would ensure significant uptake among limited resource communities globally. The manuscript is strongly recommended for publication.

Response to reviewer comments

We thank the reviewers for their careful review of the manuscript and for providing constructive feedback. In particular, we are grateful to the many suggestions of Reviewers 1 & 2, and to Reviewer 3 for positive assessment and strongly recommending publication of the manuscript. We have performed new experiments to address reviewer concerns and have revised the manuscript with these new data, which we believe adequately addresses the reviewer comments. Below, we present a point-by-point response to reviewer comments.

Original reviewer comments are *italicized*, and our response is presented in normal font with revisions to the manuscript shown in blue. Unmodified text from the main manuscript is shown as normal text. Text deletions have been shown as ~~strikethrough text~~. Text additions have been *italicized*. A separate review-only file containing highlighted changes made to the manuscript and SI is included for the reviewer's reference.

Reviewer 1

Overall, this is an interesting manuscript providing some scientific data and local information from India about a proposed new household filter - the xylem filters.

We thank the reviewer for their time in reviewing the manuscript and for their constructive comments. We would like to emphasize that, in addition to design of a xylem-based household water filter, a key contribution of the present work is to provide understanding of the properties of xylem as a material for filtration and the underlying engineering principles to enable rational design of different kinds of filters and filtration devices.

RI.1. Reviewing the manuscript and changing language such as "The xylem filters showed excellent removal of total and fecal coliform, both with and without GAC" in a more scientifically rigorous and unbiased manner.

We thank the reviewer for the comment. We have carefully reviewed and revised the manuscript in several places to present the results more rigorously and objectively by providing quantitative information and statistical significance wherever possible. The changes include the following:

Text on Page 5:

The rejection performance of ethanol-dried filters with 1 μm microspheres was significantly better than water-dried filters ($p = 2.7 \times 10^{-5}$, 3.0×10^{-9} , and 1.4×10^{-6} for 0.25-, 0.50-, and 1-inch filters respectively) and comparable to fresh filters ($p = 0.02$, 0.59 , and 0.08 for 0.25-, 0.50-, and 1-inch filters respectively) (Fig. 2f).

Two 0.25-inch water-dried filters in series had better rejection than a single 0.25-inch filter ($1.70 \pm 0.24 \log$ versus $0.95 \pm 0.16 \log$ ($p = 0.015$)), with the log rejection being additive, and higher rejection and permeance recovery than a 0.50-inch filter ($1.70 \pm 0.24 \log$ versus $1.26 \pm 0.06 \log$ ($p = 0.04$), and $48.8 \pm 4.9\%$ versus $1.8 \pm 1.0\%$ ($p = 0.0013$), respectively).

Text on Page 7:

However, filter performance varied significantly with water quality and both permeance and capacity were compromised with CTW; both peak permeance and capacity with CTW (0.022 ± 0.020 L/h-cm²-kPa and 6.07 ± 4.40 L/cm² respectively) were an order of magnitude lower than those with GTW (0.002 ± 0.001 L/h-cm²-kPa and 0.58 ± 0.47 L/cm² respectively; Fig. 4a).

Text on Page 10: ~~The xylem filters showed excellent removal of total and fecal coliform, both with and without GAC. No total or fecal coliform bacteria were detected in the filtrate for 5 out of the 6 xylem filters tested (3 filters operated with GAC and 3 filters operated without GAC).~~

Methods used for calculating the p-value have been described in methods section:

Text on Page 30 ‘M18. Statistical Information’:

The p-values for Fig. 2f and Fig. 2h were determined by performing two-sample t-tests on the data using Microsoft Excel. The sample sets were considered unpaired with equal variance. Homogeneity in variance was determined using Levene’s test.

Further, we have discussed the risks associated with the technology and comparatively high filter replacement frequency in greater detail in the revised manuscript (please refer to responses to Comment R1.2 and Comment R1.6).

In case there are still any instances where the reviewer feels that statements are not sufficiently objective, we request the reviewer to kindly identify them out so that we may address them.

***R1.2.** Referencing the extensive literature on adoption of household water filtration; and discussing the extensive behavior change that would be needed to work with users to use a filter that first requires pre-treatment with GAC or cloth, then requires daily-to-weekly changing of a wood filter, and only produces ~1 Liter/hour.*

We thank the reviewer for the helpful comment. We appreciate the reviewer’s comment and agree with the reviewer that a discussion of the suggested points and discussion of adoption and behavior change is important. Investigating the kind of behavior change that may be needed was one of the goals of our study, and we have addressed it as described below.

We would also like to clarify that pre-treatment is not essential for operation of the filter; it is an option that, in conjunction with water quality, determines the flow characteristics. It is also not a separate process, but can be included in-line as is typical in many commercially available filters. Furthermore, replacement of filters is much simpler and can be accomplished within a minute without tools, compared to maintenance activities such as cleaning of filters that requires much longer and requires space and other tools. As such, our work would enable the design of filter devices with different characteristics and the reported filter device is intended as an example and a starting point for various future designs. In addition, the characterization, modeling, and engineering of xylem filters is not limited to household water filters and has potential to be applied to different filtration uses beyond household water filters such as disaster and emergency use,

microfiltration for assessment of water quality, and as an alternative for synthetic microfiltration membranes in some applications.

The adoption of existing HWT methods has been impeded by barriers associated with affordability, access, acceptance, and awareness. Although xylem filters require frequent replacement, xylem filters could also lower these barriers and serve as a viable HWT alternative for the bottom-of-the-pyramid due to the following reasons (also mentioned on Page 11, second paragraph, ‘Technology Translation’ section): a) Our preliminary cost estimation suggest that, compared to conventional filter cartridges that cost USD 5-10 and require replacement every 3-6 months, xylem filters could cost USD 0.06–0.10 and require replacement every 1–5 days. Such amortization of filter replacement costs could significantly lower the barrier to affordability for low-income households, where ‘pay-as-you-go’ or ‘buying less, but more often’ model is preferred over longer-term filter replacements¹. In Indian urban slums where Reverse Osmosis (RO) filtered water cans costing USD 0.28-0.56 per 20 L or government-operated water booths which provide RO water at USD 0.06-0.10 per 20 L are the only alternative, xylem filters could provide a viable HWT alternative for those who cannot afford or access these options easily. They could also facilitate sustained usage of HWT amongst those who are unable to purchase cans or fetch water from the booths on a regular basis to realize effective health outcomes; b) The ubiquity of wood, open-source nature of the technology, and simplicity of the manufacturing process can enable production of xylem filters at a global scale to facilitate access, c) Due to their low weight (<10 g) and volume (~13 cm³) and long shelf-life (> 1 year), xylem filters could be shipped easily to remote locations and adequately stocked to serve local needs, and these characteristics could also be helpful for rapid distribution during disasters or emergencies. d) Evidence from our field studies suggest that in addition to access and affordability, traditional familiarity with wood, simplicity and natural appeal of the product could promote acceptance and facilitate adoption.

We have referenced existing literature on the topic of household water treatment (HWT) adoption and added a paragraph in the main text and a new Supplementary Note 9 to highlight the importance of behavior change interventions in facilitating filter adoption and sustained usage, along with our field study findings regarding behavior change interventions that may be appropriate for xylem filters that we had previously not included in the SI. The changes are shown below (due to the addition of a supplementary note, the numbering of subsequent notes has changed and has been highlighted in the main manuscript and SI). While we agree with the reviewer that the flow rate of xylem filters is lower in comparison to commercial membrane-based filters and higher flow rate could facilitate adoption², we also note that the flow rate targets for demonstrating practical use of xylem filters are based on minimum useful flow rates reported in literature³ and have been benchmarked against ceramic filters, which are widely used for HWT in low-income communities and have flow rates of 1-3 L/h⁴. Flow rates in successive design iterations of xylem filters could be easily increased to suit user needs, e.g., by increasing the area for filtration.

Text on Page 8 of the main manuscript:

In practice, pre-filtration is not essential for operation of the filter; it is an option which in conjunction with water quality determines the flow characteristics. The decision whether to incorporate pre-treatment and the choice of pre-treatment would be governed by the tradeoff

between the added convenience of longer filter lifetime or lower filter replacement frequency and user willingness to pay the extra cost and the complication of an added replaceable component, *and the need to remove any chemical contaminants that may be present in the water* (cost estimates for xylem filters and GAC provided in Supplementary Note 7).

Text on Page 12 of main manuscript:

The need for frequent filter replacement creates the risk of lack of user compliance and disruption of sustained usage, which could be exacerbated if pre-filtration units that require replacement at a different frequency than the xylem filters are used. Technological and product design improvements (such as engineering filters with longer lifetime, designing holders for easy filter replacement) or supply-related solutions (such as filters being available at stores that provide groceries or other regularly-purchased items) could help mitigate the risk of disruption of sustained usage, whereas the decrease in flow rate of filters with time mitigates the risk of lack of user compliance in replacing the filters. These efforts would nevertheless have to be complemented with behavior change interventions. The motivators for behavior change and thus, the nature of these interventions, would depend on the local social, cultural and environmental characteristics, such as water quality (whether water has visible coloration, odor or poor taste), perceived health risk associated with water quality, education level of the population, etc.⁵⁻¹¹. The influence of these characteristics on HWT adoption has been extensively studied in literature and this knowledge, in addition to field studies with user groups, could be leveraged to facilitate sustained adoption of xylem filters (see Supplementary Note 9 for factors that affect behavior change and sustained adoption of HWT and examples of behavior change interventions for xylem filters)⁵⁻¹¹.

Text on Page 16 of SI:

Supplementary Note 9: Behavior change interventions for HWT adoption

Through field visits (see Methods section M19 for further details on sampling, data collection and analysis methods), we have identified that the following factors affect behavior change and sustained adoption of HWT methods amongst low-income communities in India (these factors are in alignment with existing literature on HWT adoption in developing countries and low-income population⁵⁻¹¹):

- *Perceived quality of water source*
- *Perceived health risk of consuming contaminated drinking water*
- *Prior history of practicing HWT*
- *Cultural practices associated with water collection, storage, and consumption*
- *Affordability*
- *Accessibility to HWT products*
- *Convenience of use of HWT products*
- *Level of social support from peers*
- *Level of positive, ongoing contact with local healthcare workers*
- *Incentive structure for HWT adoption*
- *Extent of information dissemination about the effectiveness of HWT methods and their local accessibility/availability*

Based on the aforementioned factors, which are broadly applicable to HWT methods, we have identified specific behavior change interventions that could facilitate the adoption of xylem filters:

- Community-level WASH education and awareness campaigns (would involve educating users on indicators of water quality, health effects of consuming contaminated drinking water and testing local water quality with potential users to determine whether water is safe for consumption; could be conducted in collaboration with local NGOs or healthcare workers).*
- School-level education and awareness programs (could use xylem filters as effective educational tools for designing science experiments and engineering projects that capture the interest of students and raise awareness of WASH issues at a very early age, which could be a powerful way to drive behavior change).*
- Effective marketing strategies and promotional programs to facilitate uptake of xylem filters (demonstrations, free/discounted product trials, incentive programs such as providing cell phone minutes, cash, engagement of local community leaders for product endorsement, etc.).*
- Ensuring availability of filter devices and cartridges in local shops to enable easy access.*
- Appropriate pricing of xylem filters (involves gathering data on user willingness to pay, benchmarking against locally available HWT methods, and analyzing costs for filter production and distribution).*
- Designing filtration device for maximal ease of usage and gathering user feedback to identify improvements.*
- Regular monitoring and engagement with households to encourage sustained usage (visits could be conducted in partnership with local NGO staff and health care workers).*

In order to compare the cost of xylem filters with user willingness to pay, we have added text on Page 12 of the main manuscript.

Text on Page 12 of main manuscript:

Second, our preliminary cost estimation studies suggest that, in comparison to conventional filter cartridges that cost USD 5-10 and require replacement every 3-6 months, xylem filters could cost USD 0.06–0.10 and require replacement every 1–5 days with a comparable cost per liter water filtered (see Supplementary Note 7 for cost estimation details and Supplementary Fig. 6 for cost comparison with currently available commercial filters in India). Such amortization of filter replacement costs could significantly lower the barrier to affordability for low-income households, where ‘pay-as-you-go’ or ‘buying less, but more often’ model is preferred over longer-term filter replacements¹. In Indian urban slums where Reverse Osmosis (RO) filtered water cans costing USD 0.28-0.56 per 20 L or government-operated water booths which provide RO water at USD 0.06-0.10 per 20 L are the only options, xylem filters could provide a viable HWT alternative for those who cannot afford or access these options easily. They could also facilitate sustained usage of HWT amongst those who are unable to purchase cans or fetch water from the booths on a regular basis to realize effective health outcomes.

RI.3. Describing with the WHO methods for testing the efficacy of household water filters (which are published) were not used, and if microspheres are appropriate for reporting data. In particular, viral results are presented in Table 2, but it is not clear where those results came from?

We appreciate the reviewer's comment. We sent xylem filters to a certified laboratory and have obtained rejection data for *E. coli*, MS-2 phage, and rotavirus based on WHO and NSF protocols that are consistent with microsphere rejection experiments and are now included in the manuscript.

Since the mechanism of bacterial removal in xylem filters is physical sieving¹², we originally used microspheres as proxy in the more fundamental studies to elucidate the behavior of xylem filters. Microspheres have been used as proxies to study the transport of *E. coli* in porous media due to the similarity in size and zeta potential^{13,14}. To complement these tests, we had included field data in which filtration experiments were performed with natural water sources that were contaminated with coliform bacteria.

We have added the following text in Page 25 within the Methods section M4 of the manuscript to clarify the use of microspheres as surrogates for *E. coli*:

As the mechanism of bacterial removal in xylem filters is physical sieving¹¹, we expected the contaminant size to be the key determinant of the filtration characteristics and used 1 μm latex microspheres as a model to understand the behavior of xylem filters and as surrogates for E. coli. Latex microspheres have been used as proxies to study the transport of E. coli in porous media due to the similarity in size and zeta potential^{12,13}.

We have referenced this section in the main text of the manuscript on Page 4:

*Experiments with 1 μm microspheres (used as proxy for E. coli, justification provided in Method M4) confirmed that the rejection ability of 0.25-inch thick filters made from Eastern white pine (*Pinus strobus*) was comparable to 0.50- and 1-inch thick filters (Fig. 2a).*

Further, based on the reviewer's comment, we validated the microbiological performance of the filters (without pre-filtration) with WHO-prescribed¹⁵ concentrations of *E. coli* ($\geq 10^6$ CFU/mL) and MS-2 phage ($\geq 10^5$ PFU/mL) and NSF-prescribed¹⁶ concentrations of rotavirus ($\geq 10^4$ PFU/mL) through a third party certified laboratory (QFT Laboratory LLC based in New Jersey, USA). Xylem filters showed >4-log removal of rotavirus and >3-log removal of *E. coli* and MS-2 phage. Since the virus particles are smaller than the pore size of the filters (MS-2 phage and rotavirus are 24 nm¹⁵ and 70 nm¹⁷ in diameter respectively, while the pore size is 100-500 nm¹⁸), the results suggest that the mechanism of virus removal is likely to be adsorption-driven.

These new tests and results are described in a separate section on Page 9 of the revised manuscript and graphically presented in Fig. 5. The methods for conducting the tests are described in methods section 'M20. Microbiological performance as per WHO protocol' on Page 30 and 31. A supplementary table with the performance data has been added on SI Page 19.

Text on Page 9:

Microbiological Performance

To assess the potential health impact of xylem filters and their effectiveness in reducing the risk of diarrheal diseases, we tested the filters' ability to remove E. coli, MS-2 phage, and rotavirus (the

single largest causal organism of diarrhea¹⁹) from water. Xylem filters (4 cm diameter, 0.375 inch thickness, stored for 2 years, no pre-filtration) made from ginkgo were operated under a 1.2 m gravity head with General Test Water containing either WHO-prescribed concentrations of *E. coli* ($\geq 10^6$ CFU/mL) and MS-2 phage ($\geq 10^5$ PFU/mL)²⁰ or NSF-prescribed concentrations of rotavirus ($\geq 10^4$ PFU/mL)¹⁶. *E. coli* and MS-2 phage were dosed simultaneously in the same test solution while rotavirus removal was tested separately. The bacterial and virus removal was tested at the start of filter operation and when permeance declined to 75%, 50%, and 25% of the initial value. After the first sampling point at the start of filter operation, dust was added to the test solution to accelerate clogging^{15,21} (refer to Methods section M20 for further details on test procedure). The filters showed >4-log removal of rotavirus and >3-log removal of *E. coli* and MS-2 phage (Fig. 5a, data provided in Supplementary Table 1). With such rejection performance, xylem filters would fall under the ‘comprehensive protection (high pathogen removal)’ category (★★) as per the WHO scheme for classifying water treatment technologies (Fig. 5b)¹⁵. Since the virus particles are smaller than the expected pore size of the filters (MS-2 phage and rotavirus are 24 nm¹⁵ and 70 nm¹⁷ in diameter respectively, while the pore size is 100-500 nm¹⁸), the results suggest that the mechanism of virus removal is likely to be adsorption-driven. Virions can adsorb on cellulose-based materials²², with cellulose nitrate reported to remove virions that are much smaller than the filter pore size²³. We hypothesize that the relatively slow flow rate and the large thickness of xylem filters could facilitate adsorption and removal of viruses.

Figure 5:

Fig. 5| Microbiological and field performance. **a**, Microbial removal performance of xylem filters (ginkgo, 4 cm diameter, 0.375 inch thickness, no pre-filtration) when operated under 1.2 m gravity head with General Test Water containing *E. coli* ($\geq 10^6$ CFU/mL) and MS-2 phage ($\geq 10^5$ PFU/mL)

dosed simultaneously, or rotavirus ($\geq 10^5$ PFU/mL). Rejection was measured at the start of filter operation and when permeance dropped to 75%, 50%, and 25% of initial permeance (see Method M20 for further details). Different symbols indicate different filters. **b**, WHO scheme for classification of household water treatment technologies. **c**, Field set-up for testing filter performance. **d**, Tap water sample (New Delhi, India) used for testing. **e**, Xylem filters used with GAC show reduced deposits after filtration with tap water. Scale bar, 4 cm. **f,g,h** Chir pine and ginkgo filters show peak flow rates and capacity exceeding 1 L/h and 10 L, respectively, indicated by dashed lines in **f** and **g**, and absence of coliform in the filtered water. Legend is shown at the top. In **h**, chir pine and ginkgo filters were used for tap water and groundwater studies, respectively. All data are obtained with 4 cm diameter, 0.375-inch thick filters operated under 1 m head (see Method M15 for details on field testing of filters).

References to Fig. 5 in the main manuscript have been updated based on the revised figure (changes highlighted in the main manuscript).

Text on page 30 and 31 of Methods section:

M20. Microbiological performance as per WHO protocol The microbiological performance of xylem filters was tested by a third-party laboratory (Quality Filter Testing (QFT) Laboratory LLC, 1041 Glassboro Road Suite E-4, Williamstown, New Jersey 08094, USA; ISO 17025 Laboratory certified by International Association of Plumbing and Mechanical Officials (IAPMO) to test water filters as per National Sanitation Foundation (NSF)/American National Standards Institute (ANSI standards)). Xylem filters (4 cm diameter, 0.375 inch thickness) made from ginkgo (*Ginkgo biloba*) wood using hot water treatment and ethanol-based dry preservation at Massachusetts Institute of Technology (MIT), USA and stored for 2 years were mounted in 1.5 inch diameter PVC (polyvinyl chloride) tubes (McMaster-Carr, part number 5233K77) and secured in place using hose clamps, and were shipped to the third-party laboratory. In the third-party laboratory, the tubing was connected to a water-filled tank providing a 1.2 m head. *E. coli* (ATCC 11229) and MS-2 phage (ATCC-15597-B1 inoculated with host organism *E. coli* ATCC-15597) were dosed in General Test Water at concentrations $\geq 10^6$ CFU/mL and $\geq 10^5$ PFU/mL respectively as per the WHO International Scheme to Evaluate Household Water Treatment Technologies¹⁵. For rotavirus removal, rotavirus strain SA-11 (ATCC-VR-899) was spiked in General Test Water at concentration $\geq 10^4$ PFU/mL (as specified by NSF Protocol P231 for Microbiological Water Purifiers)¹⁶. The removal of bacteria and virus were tested at the start of filter operation and when permeance declined to 75%, 50%, and 25% of the initial value. Flow rates were measured by monitoring the volume of fluid processed by the filters over time. After the first sample at the start of filter operation, dust was added to achieve accelerated clogging of the filter (dust concentration raised turbidity to 120 NTU and 130 NTU for *E. coli*/MS-2 phage and rotavirus respectively, which is above what is required for Challenge Test Water). Prior to collecting the filtrate for microbiological analysis, the bottom surface of the xylem filter, the hose clamp, and lower end of the tubing were wiped with cotton soaked in ethanol (while filtration continued). Collection of the filtered sample was started after allowing 50 mL of the sample to flow in order to allow sufficient time for any residual alcohol to be flushed or evaporated. *E. coli* and MS-2 phage were assayed using Standard Method 9222 and 9224 published by American Public Health Association (APHA) for the Examination of Water and Wastewater. The log reduction values were determined by measuring the bacteria/virus count in the feed solution (C_{feed}), and filtrate ($C_{filtrate}$) using the following equation:

$$\log \text{ removal} = -\log_{10} \frac{C_{\text{filtrate}}}{C_{\text{feed}}}$$

Supplementary Table on page 19 of SI (numbering of subsequent tables and their references in the main manuscript have been revised):

Supplementary Table 1: Microbiological performance data for xylem filters*

E. coli rejection							
Sample Point	Influent concentration (CFU/L)	Effluent concentration (CFU/L)			Log removal		
		Filter-I	Filter-II	Filter-III	Filter-I	Filter-II	Filter-III
Start	3.60×10^9	1.02×10^6	2.40×10^5	3.65×10^5	3.548	4.177	3.995
75% permeance	3.85×10^9	2.14×10^6	3.10×10^6	2.60×10^5	3.256	3.095	4.171
50% permeance	4.10×10^9	3.50×10^5	4.80×10^5	1.45×10^6	4.069	3.932	3.452
25% permeance	5.20×10^9	4.20×10^5	5.60×10^5	2.70×10^5	4.093	3.968	4.285
MS-2 phage rejection							
Sample Point	Influent concentration (PFU/L)	Effluent concentration (PFU/L)			Log removal		
		Filter-I	Filter-II	Filter-III	Filter-I	Filter-II	Filter-III
Start	4.25×10^8	2.52×10^5	3.78×10^5	1.87×10^5	3.227	3.051	3.357
75% permeance	5.30×10^8	1.88×10^5	4.56×10^4	2.45×10^5	3.451	4.066	3.336
50% permeance	6.22×10^8	2.40×10^5	2.60×10^4	3.86×10^4	3.414	4.379	4.208
25% permeance	3.52×10^8	3.65×10^4	1.96×10^4	4.56×10^4	3.985	4.255	3.888
Rotavirus rejection							
Sample Point	Influent concentration (PFU/L)	Effluent concentration (PFU/L)			Log removal		
		Filter-I	Filter-II	Filter-III	Filter-I	Filter-II	Filter-III
Start	1.03×10^7	1.00×10^2	1.50×10^2	1.70×10^2	5.011	4.835	4.781
75% permeance	1.03×10^7	2.50×10^2	3.20×10^2	3.40×10^2	4.613	4.506	4.48
50% permeance	1.02×10^7	5.00×10^2	6.70×10^2	7.50×10^2	4.308	4.181	4.132
25% permeance	1.02×10^7	7.50×10^2	8.50×10^2	9.20×10^2	4.132	4.078	4.044

*Xylem filters (4 cm diameter, 0.375 inch thickness) made from ginkgo were operated under a 1.2 m gravity head with General Test Water containing WHO-prescribed concentrations of *E. coli* ($\geq 10^6$ CFU/mL) and MS-2 phage ($\geq 10^5$ PFU/mL)²⁰ and NSF-prescribed concentrations of rotavirus ($\geq 10^4$ PFU/mL)¹⁶. *E. coli* and MS-2 phage were dosed simultaneously in the same test solution while rotavirus removal was tested separately. The bacterial and virus removal was tested at the start of filter operation and when permeance declined to 75%, 50%, and 25% of the initial value. After the first sampling point at the start of filter operation, dust was added to the test solution to accelerate

clogging^{15,21} (refer to Methods section M20 for further details on test procedure). Flow rates corresponding to different sampling points have been specified in the table below.

Sample Point	Flow rates (mL/min)					
	E. coli and MS-2 rejection experiment			Rotavirus rejection experiment		
	Filter-I	Filter-II	Filter-III	Filter-I	Filter-II	Filter-III
Start	24.50	25.00	26.00	25.00	26.00	26.00
75% permeance	18.40	18.80	19.50	18.80	19.50	19.50
50% permeance	12.30	12.50	13.00	12.50	13.00	13.00
25% permeance	6.10	6.30	6.50	6.30	6.50	6.50

In light of the current findings, Fig.7a has been updated with the results for MS-2 phage, *E. coli*, and rotavirus removal. The WHO classification scheme has been moved from Fig. 7 to Fig. 5. The updated version of Fig. 7 is shown below.

** and + indicate measurements with fresh and ethanol-dried filters resp.

E.W. stands for Eastern White

Values shown as mean±s.d.

n = 3 for 100 nm particles

n = (5,3,3) for 500 nm particles for (E.W. Pine, E.W. Cedar, Ginkgo)

n = (6,3,3) for 1 µm particles for (fresh E.W. Pine, ethanol-dried E.W. Pine, fresh Ginkgo)

Fig. 7| Potential health impact and applications of xylem filter. a, Comparison of diarrheal mortality burden due to different waterborne pathogens arranged by size to the measured rejection of particles of different size by single 0.375-inch thick xylem filters made from different tree species. (** and + indicate measurements with fresh and ethanol-dried filters respectively). Mean±s.d.; n=3 for tests with 100 nm particles; for tests with 500 nm particles, n=(5,3,3) for (eastern white pine, eastern white cedar, and ginkgo); for tests with 1 µm particles, n=(6,3) for (fresh, ethanol-dried) eastern white pine filters and n=3 for ginkgo. . **b,** WHO scheme for

classification of household water treatment technologies. Xylem filters could provide > 2-log removal of bacteria and protozoa, which cumulatively account for 66% of the diarrheal deaths, and at least provide ‘targeted protection’ under WHO’s classification scheme. Rejection performance can be improved further by stacking filters in series, increasing thickness, or using other species with tighter pores. *e, b*, Illustrations of possible configurations of xylem-based filtration devices.

The results previously presented in Table 2 were based on the rejection results of xylem filters with 100 nm microspheres that are similar in size to adenovirus. But these have now been updated based on the results from the microbiological tests as shown below:

Table 2 on Page 16 of the main manuscript:

Technology	Level of protection	Advantages	Limitations
Chlorination ²⁴	Targeted (bacteria and viruses only)	 • Portable, light-weight • Simple to use, no maintenance 	 • Less effective in inorganic-rich or turbid water • Unpleasant taste and odor • Dosage depends on water quality • Harmful organic by-products
Solar disinfection ²⁴	Targeted to comprehensive (bacteria, virus, protozoa)	 • Little/no maintenance • Minimal chance of re-contamination • Simple to use 	 • Poor efficacy with turbid water • Dependent on weather • Volume to treat depends on availability of intact container • Long treatment time
Ceramic filters ²⁴	Targeted (bacteria and protozoa only)	 • Local production • Minimal chance of re-contamination 	 • Performance heterogeneity due to manufacturing variability and susceptibility to cracks • Requires regular cleaning • Difficult to transport • Low flow rate: 1-3 L/h
Xylem filters	Targeted (with bacteria and protozoa); certain species could achieve 2-log removal of virus Comprehensive (>3-log removal of bacteria and virus)	 • Biodegradable • Light weight • Minimal chance of recontamination • Local production 	 • Frequency of replacement is relatively high (maintenance concerns) • Needs pre-filtration or large filter sizes with water having high turbidity or high organic content • Low flow rate: 1-3 L/h

If the mechanism of virus removal is adsorption-driven, the rejection of virus particles will positively correlate to their contact time with the filter, which in turn, scales inversely as the square of the filter thickness. As opposed to a filtration-based mechanism where rejection was dependent on filter thickness alone, the rejection in an adsorption-based mechanism will depend on the filter thickness as well as flow rate. This dependency is discussed in updated Table 1 (page 15 of main manuscript):

Performance metrics	Xylem filter geometry		Fabrication methods			Operating conditions/parameters	
	Sapwood area (A)	Thickness (t)	Dry preservation		Hot water treatment**	With cloth or GAC pre-filtration	Pressure used for driving flow (P)
			Alcohol treatment*	Stacking multiple filters in series*			
Flow rate (Q)	$Q \propto A$	$Q \propto 1/t$, if tracheids are well-connected, else decreases more rapidly with thickness.	Increases	$Q \propto 1/n$, where n is the number of filters	No significant effect.	Generally increases; depends on water quality and pre-filtration process.	$Q \propto P$
Volumetric capacity (V)	V ∝ A is expected for the same driving pressures	Unknown	Increases	Unknown; may be comparable to a single filter.	Increases if self-blocking is more dominant than fouling due to contaminants in water, else significant effect is not expected.	Generally increases; depends on water quality and pre-filtration process.	Unknown; depends on how pressure affects fouling.
Rejection Sieving-based (bacteria, protozoa, particulates)	No effect	Depends on particle size. Typically increases rapidly till thickness exceeds tracheid length; may not increase further if sapwood has resin canals, else theoretically expected to increase with length.	Increases	Increases; theoretically log-additive, if rejection of each filter is identical.	Depends on temperature and duration of treatment; may decrease rejection. Soaking at high temperatures or for long durations compromises rejection performance.	Depends on pre-filtration method used; GAC can augment performance by removal of chemical contaminants.	No significant effect expected.
Adsorption-based (viruses)	No significant effect expected if permeate flux is constant	Expected to increase for the same driving pressure. $\tau \propto 1/t^2$, where τ is contact time.	Unknown	Expected to increase; theoretically log-additive, if rejection of each filter is identical.	Unknown		Expected to decrease. $\tau \propto 1/P$, where τ is contact time.

Further, the text in the ‘Discussion’ section has also been updated based on recent results.

Text on Page 13:

Xylem filters can provide at least 2-log removal of bacteria and protozoan cysts (typically larger than bacteria, 4–15 μm in size²⁵) respectively, which cumulatively account for 66% of the diarrheal deaths (Fig. 7a shows the diarrheal deaths caused by different pathogen classes, the size of these pathogens^{19,26–37} and measured log₁₀ reduction of single xylem filters with microspheres of different diameters). As per the WHO categorization of water treatment technologies based on their ability to remove contaminants³⁸ (Fig. 7b), xylem filters would, at the least, satisfy the requirements for the ‘targeted protection’ category (★). Our preliminary studies suggest that filters made from gymnosperm species such as ginkgo that have tighter pores can provide 1.26±0.22 log removal of 100 nm particles, which are similar in size to adenovirus. This rejection increases to 1.91±0.15 log in the presence of slight fouling, and could be improved further by stacking filters in series. With further research and development for virus removal (particularly rotavirus, which is ~70 nm in size), xylem filters could offer ‘comprehensive protection’ (★★) against water-borne diseases.

With >3-log removal for bacteria and virus (> 4-log removal for rotavirus), xylem filters can provide ‘comprehensive protection’ against water borne pathogens as per WHO’s performance criteria for household water treatment technologies¹⁵ and have potential for reducing the health

burden of water-borne diseases (Fig. 7a). The contaminant removal ability of xylem filters could be improved further due to fouling (Supplementary Note 10), by stacking filters or using other approaches. Use of silver nanoparticles in xylem to enhance the removal of bacteria and methylene blue (a commonly used industrial dye that causes water pollution) using xylem³⁹ and chemical modification of xylem surface for copper adsorption has been reported⁴⁰. Incorporating suitable sorbents such as zeolites or ferrous oxide into pre-filtration modules could enhance removal of viruses, arsenic, or other pollutants^{41,42}. More generally, xylem filters could be used synergistically with other water treatment methods, e.g., in conjunction with chlorine to reduce organic content that can produce harmful byproducts, and to remove protozoan cysts that are relatively chlorine-resistant to chlorine. Beyond gymnosperms, plants like primitive angiosperms that have short xylem conduits and nanoscale pit membrane pores and are not prone to aspiration⁴³, or ubiquitous bamboo nodes⁴⁴, could be explored for filtration applications.

RI.4. Discussing whether 20 Liters per household per day per filter is an appropriate long-term target for water use.

We thank the reviewer for raising this point. We are not sure we fully understand the comment; our interpretation is whether a filter should be targeted to provide more than 20 L per day in the future. Our target for filter lifetime and capacity is based on literature review of guidelines for HWT product design, survey of existing HWT products, and feedback gathered from users through field studies, which suggests that a filter that provides 20 L per day would be sufficient for most households. While we believe that the 20 L per household per day would be an appropriate target in the long-term for a daily use filter, the filter capacity could be designed to reflect the user needs should it be necessary.

We have modified the text in SI on Page 8 as follows to clarify how the filter capacity target was developed, along with an expanded discussion of the filter lifetime:

Although targets for microbiological performance of water filters and minimum flow rates for practical use of water filters have been well-documented in literature^{3,38}, filter lifetimes are relatively non-standardized and vary from 3-6 months for membrane-based filter cartridges to 1-2 years for ceramic filters^{4,45-52}. Since xylem filters (in their present form) have a finite lifetime and require daily to weekly replacement that is very different from existing water treatment products, literature reports were not helpful in assessing user reception or setting filter lifetime targets for xylem filters. Consequently, we relied on data gathered from user studies (focus group discussions and individual household interviews) in India for developing a target filter lifetime. The preferred filter replacement frequency was found to be closely tied to other product attributes such as cost, ease of filter replacement, ease of availability of replacements, etc., but a minimum lifetime of one day was found to be critical for uptake. The filter capacity was thus determined based on a daily filter replacement frequency and estimates on the daily drinking water need for an average household. An average household in low-income countries of Asia and Africa comprises 4–5 members⁵³. Given a recommended daily water intake for an individual of 2–3.5 L⁵⁴, the drinking water requirement for such households is 8-17 L per day. Xylem filters operated overnight (8–10 h) could meet this requirement if they have flow rates of at least 1–2 L/h; this flow rate is also the minimum useful flow rate reported in literature³. Both flow rates and capacity could be increased by increasing the area of the xylem filter.

We would be happy to provide further information or clarification in case this answer does not address the reviewer's concern satisfactorily.

RI.5. *Discussing if local ethics approval was obtained in India before the human subjects work was conducted.*

We thank the reviewer for highlighting this point. We have modified the text on Page 30 in our 'Methods' section M19 and our reporting summary to clarify that we conformed to the local ethics guidelines:

The 'National Ethical Guidelines for Biomedical and Health Research Involving Human Participants' guidelines published by the Indian Council of Medical Research (ICMR) were followed while conducting user research in India. These guidelines suggest that social and behavioral research for health applications should be approved by the ethics committee for the researchers' institution, which in our case would be the Massachusetts Institute of Technology Committee on the Use of Human Subjects (MIT COUHES). All human-subjects research procedures were approved by MIT COUHES under protocol 1612798762.

RI.6. *Clearly linking the methods described to scientific results, and unbiased discussion.*

We have numbered our methods in the main manuscript and referenced them in figure captions for the sake of clarity.

In order to better balance the discussion, we have explicitly highlighted the limitations associated with the technology by adding text on Page 13 of the main manuscript.

Some limitations of xylem filters for household water treatment currently are: a) the higher replacement frequency of xylem in comparison to current filtration technologies could deter sustained usage, b) heterogeneity in xylem structure within and across gymnosperm species could create variability in filter performance, and c) a possibility that degradation of filters under certain conditions could cause their rejection performance to deteriorate and prevent them from getting blocked posing a health risk to users (timescales for degradation would depend on the feed water quality, type of wood used for manufacturing the filter, etc. and could be of the order of 1-2 weeks^{53,54} or less since bacteria are directly seeded on pit membranes during filtration). Mitigation of these risks will require: a) pilot studies of filter performance under different operational conditions (and measures such as increasing filter thickness or providing guidance as to when xylem filters are suitable for use if filter degradation is found to be an issue), b) standardization of sources of wood and development of methods to test and assure quality control⁵⁵, c) safety assessment of xylem filters for human use (e.g., assessing the filtrate for presence of organic compounds leached from xylem and evaluating their safety for human consumption, noting that cellulose, hemicellulose, lignin, and pectin occur naturally in many foods, and that plant material contributes to organic content in drinking water), d) designing filters holders for maximal ease of replacement along with ready local availability of filters, and e) behavior change interventions to facilitate sustained usage (see Supplementary Note 9 and 11 for more information on these topics).

We hope that this text, in conjunction with the edits made in response to comments R1.1 and R1.2 would present the technology in a more balanced light. If the reviewer feels that this concern has not been adequately addressed, we request the reviewer to kindly provide more specific suggestions as to how we could fully address this concern.

Reviewer 2

This paper reported on the evaluation of xylem filters for water treatment applications. The paper develops guidelines for use of the filters in practical applications and evaluates the filters in a treatment system that results from participatory co-design. Overall, I judge the manuscript to be well-written and the results are interesting. They take items that are very fundamental and elucidate their influence on the end practical implications. However, there are some items which need to be addressed before this work would be considered for publication in a journal like Nature Communications.

We are grateful to the reviewer for their time to read the manuscript in detail and their encouraging and insightful comments.

R2.1. Justify the usage of 1 um microspheres as a proxy for E. coli. Justify that the surface properties are the same.

We thank the reviewer for bringing up this point. We have now included studies of *E. coli*, MS-2 phage, and rotavirus rejection using third-party laboratory based on WHO and NSF protocols. The rejections for *E. coli* are consistent with those obtained with microspheres. We found that the xylem filters can achieve > 3-log removal of *E. coli* and MS-2 phage and > 4-log removal of rotavirus. We have included these results in the revised manuscript. A detailed description of the study, results and the corresponding changes in the manuscript can be found in our response to Reviewer 1 Comment R1.3 above.

We had used microspheres in the more fundamental studies to understand the behavior of xylem filters, such as the effect of filter thickness, drying, etc., and as surrogates for *E. coli*. Since the mechanism of bacterial removal in xylem filters is physical sieving¹², and microspheres have been previously used as proxies to study the transport of *E. coli* in porous media due to the similarity in size and zeta potential^{13,14}, we expect the microsphere rejection to be comparable to rejection of *E. coli* (which was indeed the case with the new studies with *E. coli*). To complement microsphere tests, we had included field data in which filtration experiments were performed with natural water sources that were contaminated with coliform bacteria. All of these results now present a consistent picture of the ability of xylem filters to remove microscale particles and *E. coli*.

We have modified the text on Page 25 in the Methods section of the manuscript to clarify the use of microspheres as surrogates for *E. coli* and referenced this section in the main text of the manuscript on Page 4. Please refer to our response to Comment R1.3 for a detailed description of the edits.

R2.2. One of the big challenge the authors state is that the filters can degrade when dry. They conduct a treatment to minimize this factor for new filters. However, in many developing world situations, it would not be expected that the filters will be used continuously. What is this implication on the permanence of the filters?

We thank the reviewer for the insightful comment. Xylem filters are not intended to be, and are not suitable as ‘permanent’ filters, but inherently as low-cost, disposable filters. The filters do get blocked when dry; the ethanol soaking dry preservation method reported in the manuscript allows the filters to be dried, stored for long periods of time, and then re-wetted for use. Once re-wetted, the filters can be used and disposed; they will block if dried again. However, the filter holder that we reported confines the xylem filter such that drying is slow. In general, the filter holder could be designed to slow down drying as needed (perhaps to a few days duration). Since the xylem filters are disposable, we do not envision use cases where the same filter would be used for a short while, dried, and reused. However, it is important for the filter to not get clogged in more common intermittent use cases, such as where water is added to the filter every night.

In order to better understand the implications of intermittent use on filter permeance, we conducted experiments where filters made from ginkgo were operated intermittently for a duration of 7 days while processing 5 L of DI water per day. The filters were mounted in the device (shown in Supplementary Fig. 5a) and operated under a gravitational pressure head of 1 m with 5 L of DI water added into the device every day. The filters stopped producing water after the 5 L water was filtered, till 5 L of water was again added on the following day. The time required to filter the 5 L of water was measured and used to calculate the average permeance to filter 5 L of water on a given day. The graph below shows the variation in permeance versus permeate filtered for continuous and intermittent operation (data for continuous operation taken from Fig. 3f; different colors correspond to different filters). The filters operated intermittently showed a qualitatively similar trend as those operated continuously with values in the same range for both permeate filtered and permeance.

The graph above has been added to Supplementary Fig. 2 in SI and referenced on Page 7 of the main manuscript.

Page 7 of manuscript:

Eastern white pine filters fabricated using hot water soaking and ethanol drying could maintain permeance >0.01 L/h.cm².kPa while filtering at least 11 L/cm² of DI water (Fig. 3f). Thus, in the absence of fouling due to constituents in the feed water, filters with 10 cm² area (3.6 cm diameter) would achieve flow rates >1 L/h and volumetric capacity of ~ 100 L under gravity-driven operation with 1 m head (*see Supplementary Fig. 2f for variation in permeance with permeate filtered for intermittent and continuous operation and Supplementary Fig. 2g for scaling of flow rate with filter area*).

The experimental protocol followed for measuring permeance and permeate filtered during intermittent operation has been described in Methods section M2 on Page 24 of the manuscript.

Page 24 of manuscript, Methods section M2:

For experiments involving intermittent filter operation, filters made from ginkgo (4 cm diameter, 0.375 inch thickness) were operated for a duration of 7 days while processing 5 L of DI water per day. The filters were mounted in the device (shown in Supplementary Fig. 5a) and operated under a gravitational pressure head of 1 m with 5 L of DI water added into the device every day. The filters stopped producing water after the 5 L water was filtered, till 5 L of water was again added on the following day. The time required to filter the 5 L of water was measured and used to calculate the average permeance to filter 5 L of water on a given day. Permeance was calculated by normalizing the flow rate by the filter cross-section area and the driving pressure. The permeate processed by the filter per unit area was calculated by normalizing the volume of the fluid processed by the filter by the filter cross-section area.

Prolonged use of filters (beyond approximately 1 week) is not desirable, since degradation of the filters could occur that may compromise performance. This degradation is different from blocking – blocking of filters is a mechanical process, whereas degradation could occur due to action of microbes that degrade wood. Although we had mentioned degradation in a few places in the original manuscript, it is now also included with additional references and explanation in the discussion section where all potential limitations of the filter are discussed. The limited capacity of the filters is protective in this regard, since filters tend to get blocked and therefore naturally limit prolonged operation. Please see response to Reviewer 1 comment R1.6 for more details.

R2.3. *Please confirm that the type of ethanol being used is safe for food use.*

Since the filtrate from xylem filters tested in lab and field studies was not intended for consumption, the ethanol used for fabricating filters for these studies was reagent grade (CAS number 64-17-5, Koptec (200-Proof) and Merck ($> 99.9\%$ purity)^{56,57}). When manufacturing filters for practical use, appropriate food safety standards should be consulted to determine the grade of alcohol (methanol, ethanol, isopropyl alcohol, etc.) that can be used⁵⁸. Further, the level of residual alcohol (especially for methanol) in dried filters should be maintained within the permissible limits for human consumption as prescribed by food safety standards^{58,59}. Food-grade alcohols (including methanol) are commonly available commercially and used for several applications such as baking, food processing, cosmetic manufacturing, production of botanical oils and extracts, etc.^{60,61}.

In addition to the text in Supplementary Note 8 on ‘Resource requirements for fabricating xylem filters’ that highlights the aforementioned points, we have added a line on Page 6 of the manuscript to stress the importance of using food-grade solvents:

The solvent used for dry-preservation must be certified as food-grade and the level of residual solvent in dried filters should be maintained within the permissible limits for human consumption as prescribed by food safety standards⁵⁹.

We also note that cost estimates for the xylem filter (Supplementary Note 7 in SI) are based on use of food-grade certified alcohol for dry preservation.

R2.4. *You note that the cost could be broken into small amounts and filters replaced every day (or few days). In this point, you compare with other filters. You should note here that the xylem filters may end up being more expensive per unit water produced, as shown in supplementary figure 6.*

We appreciate the reviewer’s comment. that the comparison of cartridge cost per unit water filtered between xylem filters and commercial filters shown in Supplementary Fig. 6 (in the original version of the manuscript) is rather conservative since the water quality used for determining capacity was not common across all filters. The capacity ratings for xylem filters were developed based on tests conducted with WHO-prescribed synthetic test waters – General Test Water (GTW), representing high quality ground water for normal operating conditions and Challenge Test Water (CTW) with aggressive water quality specifications for stressed operating conditions³⁸. In contrast, the capacity ratings for commercial filters were taken as is from manufacturer’s websites where the water quality for standard test conditions used to obtain capacity resembled⁶² or is expected to resemble GTW (capacities with CTW have been reported by independent researchers to be an order of magnitude lower than manufacturer-specified ratings for several gravity-driven filters that were included in in the original Supplementary Fig. 6⁶³).

To facilitate an accurate comparison, we have revised Supplementary Fig. 6 to separate out the performance with GTW and CTW. The cost of xylem filters has been compared to commercial filters for GTW only (Supplementary Fig. 6a). Xylem filters provide >20× reduction in recurring cost in comparison to current filters (USD 0.06-0.10 versus USD 5-10), while providing a lower or comparable cost per liter water filtered. Such amortization of filter replacement costs could significantly lower the barrier to affordability for low-income households, where ‘pay-as-you-go’ or ‘buying less, but more often’ model is preferred over longer-term filter replacements¹. In Indian urban slums where Reverse Osmosis (RO) filtered water cans costing USD 0.28-0.56 per 20 L or government-operated water booths which provide RO water at USD 0.06-0.10 per 20 L are the only alternative, xylem filters could provide a viable HWT alternative for those who cannot afford or access these options easily. They could also facilitate sustained usage of HWT amongst those who are unable to purchase cans or fetch water from the booths on a regular basis, to realize effective health outcomes (further details on how xylem filters could facilitate adoption of HWT amongst low-income households have been provided in our response to Reviewer 1 Comment R1.2).

The variation in xylem filter cost per liter water filtered with water quality and pre-filtration method is now shown in a separate panel (Supplementary Fig. 6b; with filter capacity used for the

cost calculation listed in the legend within brackets). The use of a pre-filtration module (cloth + GAC) with Challenge Test Water reduces the cartridge cost per liter water by >2× for a marginal increase in recurring cost (assuming that the cost of the pre-filtration module is amortized over cartridge replacements; details in Supplementary Note 7).

Supplementary Fig. 6 | *a*, Cost comparison of xylem filters with commercial filters belonging to four major brands (denoted by different colors) in India shows the potential to offer unprecedentedly low cartridge replacement cost for a comparable or lower cost of cartridge per liter water filtered. Product names are included. Capacity ratings for xylem filter are based on data obtained for ginkgo filters with WHO-prescribed General Test Water (GTW) (Fig. 4a, f); capacity for HUL Pureit Series is rated at TDS of 300 ± 10 ppm, pH of 7.5 ± 0.5 , TOC < ppm, turbidity < 1 NTU. Capacity ratings for other filters are based on manufacturer-specific 'standard testing conditions'; in some cases, filter lifetime can be an order of magnitude lower if the water quality resembles WHO-prescribed Challenge Test Water (CTW). *b*, Xylem filter cartridge cost per liter water filtered for CTW is greater than that for GTW; use of cloth and Granular Activated Carbon (GAC) pre-filtration can reduce this cost substantially for a marginal increase in recurring cost. Capacity ranges and average values corresponding to different water qualities and pre-filtration methods have been specified in the legend and are based on volumetric capacity measured for ginkgo filters in Fig. 4a, f (details of cost and capacity estimation have been provided in Supplementary Note 7). The horizontal lines indicate average estimated cartridge cost per liter (calculated as ratio of average filter price (INR 5.5) and average volumetric capacity specified in the legend).

Details on the cost estimates used for constructing Supplementary Fig. 6 are described on Page 13 of SI:

In Supplementary Fig. 6, the recurring (cartridge) cost of a xylem filter is compared with different commercial filters in the Indian market. The rated capacity and the costs of filtration units and replacement cartridges was obtained from product websites^{45–52}. The cartridge cost was normalized by the rated capacity to obtain the cost per liter of filtered water. The recurring cost for xylem filters was estimated using the above xylem filter cost (INR 4.5–7). Cost of cartridge per liter water filtered was obtained by dividing the recurring cost with the filter capacity for different water qualities. Capacity was estimated by using the data presented for ginkgo filters in Fig. 4a, f to obtain the permeate filtered per unit area for different water qualities and pre-filtration methods and then scaling the permeate per unit area for filters with 5 cm diameter; capacity ranges and average values for different scenarios have been specified in the legend. The average cartridge cost per liter water filtered for each scenario was estimated by dividing the average filter price (INR 5.5) with the corresponding average volumetric capacity. The cost of the GAC column was evenly amortized across the cartridge replacement costs as follows:

- *Assuming that the daily drinking water requirement for an average household is 8-17 L per day (as estimated in Supplementary Note 1), the replacement frequency of the xylem*

filter cartridge was estimated for different water qualities based on the corresponding filter capacity (as specified in the legend for Supplementary Fig. 6, based on data presented in Fig. 4a, f).

- *The total number of xylem filter cartridge replacements required over the lifetime of the GAC column (1.5-6 months) was estimated.*
- *The cost of the GAC column was evenly amortized across the total number of replacements and added to the filter cartridge cost.*

We also note that the major cost in xylem filter manufacture arises from the alcohol used for dry preservation. This cost could be substantially reduced by recycling alcohol when manufacturing filters at scale. The current calculations do not account for such reductions, as they are difficult to estimate without specifying the scale of operations.

R2.5. *You note a few days replacement time. I know you noted that this could be good from a user perspective since the cost can be spread out. However, this type of operation can often lead to equipment falling into disrepair. I was unable to find many academic studies showing these results, however regular maintenance of simple things, like water containers, is even shown to be infrequently done. Please review and discuss potential downsides of needing to replace the filters frequently.* <https://link.springer.com/article/10.1186/s12199-019-0799-3>.

We appreciate this important point. Based on the reviewer's suggestion, we have added a paragraph in the main text and a supplementary note to highlight the limitations of a higher replacement frequency and the importance of behavior change interventions in facilitating filter adoption and sustained usage. Please refer to our response to Reviewer 1 Comment R1.2 for a detailed description of the edits. We also wish to note that maintenance encompasses a broader set of activities beyond replacement, such as routine cleaning, repairs, etc. which could be quite time-consuming and require user skill/knowledge^{8,64}. With appropriate design of the filter holder, replacing xylem filters could take less than a minute and be much easier than other maintenance activities, and would not require additional space or tools. Therefore, for a well-designed device, filter replacement is unlikely to have the same negative impact on sustained usage as maintenance activities such as cleaning.

R2.6. *You note the frequency of maintenance noted for the filters, but the frequency of maintenance for the cloth or GAC filter are not noted.*

We thank the reviewer for raising this point. We had previously stated the replacement frequency of the cloth and GAC filter in Supplementary Note 7, but have now moved this information to the main manuscript. The use of GAC is not integral to the operation of a xylem-based filter. Whether it is beneficial will depend on whether the water has organic or other pollutants that need to be removed by GAC, or if the increase in volumetric capacity per xylem filter through use of GAC is desirable over more frequent filter replacement or a larger filter. The added text on Page 8 of the main manuscript has been shown in underlined text below:

In practice, pre-filtration is not essential for operation of the filter; it is an option which, in conjunction with water quality, determines the flow characteristics. The decision whether to

incorporate pre-treatment and the choice of pre-treatment would be governed by the tradeoff between the added convenience of longer filter lifetime or lower filter replacement frequency, *cost*, and the complication of an added replaceable component, plus the need to remove any chemical contaminants that may be present in the water (cost estimates for xylem filters and GAC provided in Supplementary Note 7). The replacement frequency of the cloth or the GAC module would vary depending on the type of cloth/GAC used, configuration of GAC module, and water quality. While the cloth pre-filter could be washed or replaced once it is dirty, the GAC might need replacement once every few months (1.5-6 months; see Supplementary Note 7 for estimates on GAC replacement frequency). The reduced lifetime or slower flow rates even with newly-replaced xylem filters could be used as an indicator for pre-filtration module replacement.

Further, the cost of GAC column per liter water filtered was previously estimated based on the price of commercially available GAC columns for household water filtration and manufacturer-claimed capacity ratings. Since information about water quality at which capacity ratings were determined was lacking, we have followed a more accurate approach based on adsorption characteristics of GAC to determine GAC cost and replacement frequency, which has been described in Supplementary Note 7. The changes to the text on page 13 of Supplementary Information are shown below:

~~If used, a piece of cloth would add \$1 (taken from report of surveys conducted in India⁶³) and a GAC column for pre-filtration would add \$0.1–0.9 per 1000 L of water filtered^{65–67}. While the cloth pre-filter could be washed or replaced once it is dirty, the GAC granules will necessarily need replacement once every 3–6 months.~~

~~If used, a cloth filter would add \$1 to the cost (taken from report of surveys conducted in India⁶³) and could be washed or replaced once it is dirty.~~

~~The GAC needs replacement after its adsorption capacity is expended, the frequency of which depends on the amount of GAC used, water quality, and adsorption capacity of the GAC. A GAC column for pre-filtration is expected to cost USD 0.67–5 per 1000 L of water filtered and require a replacement every 1.5-6 months based on the following estimates:~~

- ~~- Cost of coal-based GAC in a column measuring 5 cm in diameter, 15 cm length (as described in Supplementary Note 6) calculated using the following estimates:
 - ~~○ (In the US) Price and density of USD 7-10.5 per kg and 0.5 g/cm³ respectively for coal-based GAC⁶⁸: USD 2-3~~
 - ~~○ (In India) Price and density of INR 100 per kg and 0.5 g/cm³ respectively for coal-based GAC⁶⁹: INR 15~~~~
- ~~- Adsorption capacity of GAC for organic matter (estimated based on equilibrium adsorption data for dissolved organic matter (DOM) and humic acids on coal-based GAC (granule size 0.3-0.4mm) reported in literature; input concentrations of DOM/humic acid were in the range of 5.5-12 mg/L; adsorption capacity of GAC was determined based on the amount of humic acid adsorbed per unit mass of GAC at saturation, i.e., when the residual concentration in the treated solution equaled initial concentration⁷⁰): 30-100 mg (DOM or humic acid)/g GAC~~
- ~~- Total amount of organic matter that can be adsorbed by a coal-based GAC column with aforementioned dimensions : 4.5-12 g for DOM/humic acid~~

- *Volumetric capacity of GAC column (total volume of water the GAC column can process while effectively removing organic contaminants) estimated based on adsorption capacity for DOM/humic acid concentrations in water ranging from 5.5 mg/L to 12 mg/L⁷⁰: ~500 – 2000 L*
- *Cost of GAC column per 1000 L of water: USD 0.5-3 or INR 0.008-0.03*
- *Replacement frequency of GAC columns assuming each household comprises of 4 members, each consuming 3 L of drinking water per day^{53,54}: 1.5-6 months*

The cost and replacement frequency estimates of the GAC module have been referenced in Supplementary Note 6 on ‘Design of GAC column’:

Page 12 of SI:

Based on these results, we designed a coal-based GAC column (5 cm diameter, 15 cm length) consisting of 30×40 granules that could be operated at flow rates of ~2 L/h (flow rates controlled by a valve) to reduce humic acid concentration in CTW to < 1 mg/L (*details on cost and replacement frequency of GAC are provided in Supplementary Note 7*). ~~The GAC needs replacement after its adsorption capacity is expended, the frequency of which depends on the amount of GAC used, water quality, and adsorption capacity of the GAC.~~

Supplementary Fig. 6 has been revised in accordance with the updated cost estimates for the GAC column.

R2.7. *Please be sure you discuss any further issues associated with the proposed technology.*

We have explicitly highlighted the limitations associated with the technology by adding text on Page 13 and 14 of the main manuscript. Please refer to our response to Reviewer 1 Comment R1.6 for a detailed description of the edits.

R2.8. *You discuss the local manufacture of the filters, but do not tie this back to any of the issues you encountered in your prototyping. Do you feel any of these will be barriers towards local manufacturing of the filters.*

We thank the reviewer for bringing up this point. We realize that we had not made a clear distinction between local manufacture of xylem filters (wooden discs) versus that of filtration devices (device that houses the xylem filters) in the manuscript.

The term ‘local manufacture’ refers to the ability to source raw materials and perform all manufacturing-related activities for xylem filters as well as the device locally (in this case, in India). The xylem filters used for some field tests (Fig. 6f, h and Supplementary Fig. 6a, b, g) were ‘locally manufactured’ using indigenous pine trees (*Pinus roxburghii* or ‘chir’ pine) and local resources for cutting, hot water soaking, and dry preservation. The filtration devices shown in Fig. 6a, b and Supplementary Fig. 5a were manufactured in the US, but the device shown in Supplementary Fig. 5b, c was fabricated in India. In the future, even the filters that were made in the US could be ‘locally manufactured’ in India; the raw materials required for filtration device manufacture are commonly available items (plastic containers for feed and raw water, tubing, O-rings, dispensers, steel rods for creating the support structure, plastic pellets/rods for developing

filter holder) and the processes required for device fabrication (cutting, drilling, injection molding, etc.) are well-established in the industry. The filtration device shown in Supplementary Fig. 5b that was fabricated in India comprises a wall-mounted container for feed water and stainless steel receptacle for the filtered water. The xylem filter is inserted into a hole in a thick gasket, which seals the filter from the sides. The gasket is clamped between a holder plate that rests on four mounting supports within the receptacle and the lid of the receptacle. The steel and plastic containers used in the device were purchased from a local shop whereas the rubber gasket used for mounting the filter was fabricated using a custom-designed mold at a local medium-scale machine shop.

For the device shown in Fig. 6a, b, several of the challenges encountered during prototyping were related to the identification of the appropriate design and process for filter manufacture as opposed to material availability or access to resources (for example, cutting the wood at higher speeds to ensure a smooth surface finish and good seal, increasing the size of the tubing to prevent air entrapment, etc.). As we now have a good understanding of the solutions to these challenges, we do not expect them to be significant barriers for local manufacturing of the filtration devices. Further, while the proposed design illustrates how xylem filters can be incorporated within a device, several other designs could also be explored depending on local resource availability.

We have edited the manuscript to clarify what local manufacture of xylem filters means:

Page 10 of manuscript:

Xylem filters made from ginkgo trees in US and ~~locally available Pinus roxburghii (chir pine) in India~~ those manufactured in India with indigenous Pinus roxburghii (chir pine) using local resources for all fabrication steps, i.e., cutting, hot water soaking, and dry preservation were tested with water from natural springs, municipal taps, and tubewells (groundwater) (which were the primary sources of drinking water in Uttarakhand, Delhi and Bangalore respectively; see Supplementary Table 2 for water quality information).

A list of resources and equipment needed for filter and filtration device fabrication has been provided in Supplementary Note 8. Here, we have clarified that that the filtration devices shown in Fig. 6a, b and Supplementary Fig. 5a were fabricated in the US but could be locally manufactured in India in the future. We have also highlighted that the filter holder is suitable for mass manufacture using injection molding and the device design can be tuned in accordance to local resource availability.

Supplementary Note 8 in SI (added text underlined):

Page 14:

The table below enlists the resource requirement for the filter designs depicted in Fig. 6a, b and Supplementary Fig. 5a. Although the devices depicted in these figures was fabricated in the US, they are amenable to local manufacture in India. The containers, O-rings, valves, tubing, dispenser and metal rods used in the device are commonly available items and can be sourced locally. The processes necessary for device fabrication (cutting, drilling, injection molding, etc.) are also well-established in the manufacturing industry. The device design could also be tuned as

per the local availability of resources. For example, the filtration device shown in Supplementary Fig. 5b, c was fabricated in India. The steel and plastic containers used in the device were purchased from a local shop whereas the rubber gasket used for mounting the filter was fabricated using a custom-designed mold at a local medium-scale machine shop. The cost of the device (purchase price of containers, cost of mold fabrication, rubber cost) was INR 800 (USD 11), which could be reduced substantially when manufacturing at scale.

Page 15:

The particular holder, designed for 5 cm diameter, 0.375-inch thick filters, was machined from High Density Polypropylene, *but the design is amenable to mass manufacture using injection molding.*

R2.9. *In figure 1d, you give very high accuracy on the number of diarrheal deaths. I doubt you can know the breakdown with this level of accuracy.*

The number of diarrheal deaths for the year 2016 was taken as is from the Global Health Observatory data repository managed by the World Health Organization (WHO)⁷¹. The total deaths corresponding to each of the six WHO world regions (African region, region of the Americas, South-East Asia region, European region, Eastern Mediterranean region, and Western Pacific region)⁷² were calculated by adding the deaths associated with the countries in that region. We have rounded the number of diarrheal deaths to the nearest hundred for ease of reading and modified Fig. 1 as follows:

We have modified the Methods section M17 on page 29 of the manuscript to provide more details on the source of data and figure construction:

M17. Construction of Fig. 1d *The number of diarrheal deaths for the year 2016 were taken as is from the Global Health Observatory data repository managed by the World Health Organization (WHO)⁷¹. The total deaths corresponding to each of the six WHO world regions (African region, region of the Americas, South-East Asia region, European region, Eastern Mediterranean region, and Western Pacific region)⁷² were calculated by adding the deaths associated with the countries in that region. The number of diarrheal deaths were rounded to the nearest hundred for ease of reading. The numbers were color-coded on the world map depicting the WHO world regions using Adobe Photoshop. The global distribution of wild conifers reported in literature⁷³ was re-sketched over this map in Microsoft Power Point.*

R2.10. *In figure 2h, something is fishy with your rejection numbers. Shouldn't these be in the high 90s? Please review and revise accordingly.*

We thank the reviewer for pointing this out. The rejection values were specified on the log-scale but the axis was mislabeled. We have corrected this in the manuscript.

R2.11. Figure 3, there are hash lines in multiple figures and it's not clear what they represent.

We thank the reviewer for raising this point. For the sake of clarity, we have removed the hashed lines in Fig. 3a and explained the significance of the hashed lines in Fig. 3f in the caption.

Fig. 3| Self-fouling behavior. **a**, Permeance of 0.25-inch thick ethanol-dried filters made from different gymnosperm species decreases with permeate volume when filtering deionized water ($n=3$, denoted by different colors). **b**, Microfibrils are covered by deposited material in pit membranes of blocked filters (SEM image, top) and filtrate dried on a surface contains particulates (AFM image, bottom), suggesting dissolution and deposition of organic material within the filter. Scale bars, $2\ \mu\text{m}$ **c**, FTIR spectra of different samples of filtered water indicate that hemicellulose leaches out of xylem filters. Modes corresponding to FTIR peaks are specified. **d-f**, Hot water soaking improves volumetric capacity and retains rejection (0.375-inch thick filters; $n=3$, mean \pm s.d. are indicated in **e**). Different colors denote different filters in **f**). Data were obtained with 1 cm diameter Eastern white pine filters operated under 1 m gravity head. *The horizontal dashed line denotes the permeance ($0.01\ \text{L/h/cm}^2/\text{kPa}$) corresponding to the target flow rate of $1\ \text{L/h}$ with a $10\ \text{cm}^2$ filter area and 1 m gravity head, whereas the vertical dashed line corresponds to a volumetric capacity of 100 L, which is achieved by the hot water soaked and ethanol-dried filters while maintaining the target permeance.*

R2.12. *Figure 4b, why does the capacity not decrease with tannic acid. If you aren't going to explain, then why is it there? Does it add some insight?*

We thank the reviewer for bringing up this point. In Fig. 4b, the observation that the capacity does not decrease with tannic acid indicates that small, homogenous, organic compounds like tannic acid do not cause significant fouling of xylem filters. The key insight obtained from this observation is that large, heterogeneous organic contaminants like humic acids and particulate contaminants like dust have the most detrimental impact on filter lifetime and capacity and are the key targets for pre-filtration to enhance filter lifetime. We have edited the text on Page 8 of the manuscript as follows to clarify this point:

To identify the key foulants that cause deterioration, we measured filtration capacity of xylem while selectively adding different constituents at varying concentrations. Xylem filters were most

susceptible to fouling by humic acids (present in decomposed organic matter) followed by particulates (dust) (Fig. 4b and c). *By contrast, tannic acid did not impact filter capacity significantly, indicating that the filters have a low susceptibility for fouling with small, homogeneous organic molecules.*

R2.13. *Figure 4a and 4f, why is there a reason that the axes are different here? Aren't they essentially a scaled version of each other. Also, why is it 5cm one spot and 4cm another spot.*

We thank the reviewer for the note. The axes for Figure 4a and 4f are scaled versions of one another, but the data are not (measurements were made on different sets of filters for both the graphs). In Figure 4a, the normalized peak flow rates and capacity for filters with different areas demonstrate consistency (regardless of filter diameter or species) in achieving good permeance and capacity with General Test Water and low permeance and capacity with Challenge Test Water, demonstrating the need for improving filter performance in the latter case. The different symbols represent filters with different diameters. Figure 4f on the other hand, highlights the absolute peak flow rates and capacity obtained with filters for different water qualities and different methods for pre-filtration, thereby providing readers with a realistic sense of the performance that can be obtained under practical settings with given filters (ginkgo, 4 cm diameter in this case).

R2.14. *You discuss on how 0.25" is the right filter thickness, but then your prototype uses 0.375" thick filters. Please explain and justify why this thickness should be used?*

We appreciate the reviewer's comment. The close gap between a filter thickness of 0.25 inches and the typical conduit length of Eastern white pine xylem (< 0.22 inches) presented a very narrow tolerance range for cutting filters. Consequently, minor deviations in filter manufacture introduced a high sensitivity in rejection performance. In order to circumvent this challenge (particularly with larger branches, where the presence of bends makes it difficult to ensure a uniform thickness), the filter thickness was increased to 0.375 inches for all subsequent experiments. We have added text on Page 7 of the main manuscript to explain the increase in thickness:

However, we observed that the rejection performance of 0.25-inch thick filters was sensitive to the variability in filter thickness, which is expected if the filter thickness approaches the length of the xylem conduits (tracheids) in Eastern white pine⁷⁴. To circumvent this issue, the filter thickness was increased to 0.375 inches for all filters in subsequent studies.

Reviewer #3 (Remarks to the Author):

R3.1. *The authors have presented a very thorough characterization of gymnosperm xylem based filtration system and its engineering use in limited resource communities through successful field trials in India. Some of the key fluid transport issues have been addressed, particularly the "self-blocking" characteristics of these filters and an innovative solution is proposed to overcome it. It has been demonstrated that the proposed system can work well with different types of contaminated water. Also, utilization of a pre-filtration stage is interesting. The manuscript is very well-written, easy to understand for general audience and allowing an "open source" design*

architecture for the filter system would ensure significant uptake among limited resource communities globally. The manuscript is strongly recommended for publication.

We sincerely thank the reviewer for the positive assessment of the manuscript and encouraging words. We expect that the open source nature of the technology and its publication in a journal like Nature Communications that reaches a broad audience may indeed be able to facilitate its translation to resource-limited settings.

Additional changes to the manuscript

In light of the recent results on microbiological performance of filters with *E. coli*, MS-2 phage and rotavirus, the following changes have been made:

1. Previous ‘Supplementary Note 10 on rotavirus removal’ has been modified in light of the recent results to highlight the improvement in rejection of 100 nm particles due to fouling. Supplementary Note 10 on page 16 of SI:

Supplementary Note 10: Suitability of xylem filters for rotavirus removal Improvement in rejection ability due to fouling

~~Rotavirus, which causes ~29% of the diarrheal deaths amongst children under the age of five⁷⁵, as a diameter of ~70 nm¹⁷. Xylem filters made from gymnosperm species with relatively small pores may be capable of rotavirus removal. Moreover, Deposition of foulants on the pit membranes over the course of filter operation is likely to further improve rejection. We observed that *Ginkgo biloba* filters rejected 94.01±3.31% of 100 nm particles, and that the deposition of merely 0.13 mg of foulant (humic acid) per cubic centimeter of the filter volume improved the rejection to 98.73±0.41%. Virus removal may also be aided by adsorption or inactivation in pre-filters. Although further research is required to accurately evaluate the efficacy of rotavirus removal, these preliminary results show promising potential.~~

2. Abstract has been modified to include the virus removal capabilities of xylem filters: *We develop guidelines for the design and fabrication of xylem filters, demonstrate gravity-operated filters with shelf life >2 years, and show that the filters can provide >3 log removal of *E. coli*, MS-2 phage, and rotavirus from synthetic test waters and coliform bacteria from contaminated spring, tap, and ground waters.*
3. The last paragraph in the ‘Discussion’ on Page 14 has been modified to clarify that the application of xylem filters is not limited to household water treatment:
Due to the worldwide availability of gymnosperms and simplicity of the filter fabrication process, xylem filters present opportunities for local manufacture of a wide variety of water treatment and other filtration products, ranging from compact filter pouches for use in emergencies to household filtration devices (Fig. 7c). This opens up opportunities for involvement of micro-entrepreneurs at a global scale, and implementing implementation of innovative business models, where and engagement of local communities are engaged in filter distribution and manufacture to raise awareness about importance of safe drinking water and encourage adoption. In addition, the characterization, modeling, and engineering of xylem filters is not limited to household water filters and has the potential to be applied to different filtration needs, such as for disaster and emergency use (Fig. 7c),

microfiltration for assessment of water quality, and as an alternative to synthetic microfiltration membranes for some applications.

Minor edits and clarifications made to the manuscript are not listed here.

References:

1. Banerjee, A. V & Duflo, E. The Economic Lives of the Poor. *J. Econ. Perspect.* (2007) doi:10.1257/jep.21.1.141.
2. PATH. Our end-users as co-designers: Development of the Safe Water Project Reference Design and Design Guidelines. (2011).
3. Potters for Peace. Best Practice Recommendations for Local Manufacturing of Ceramic Pot Filters for Household Water Treatment. *Group 187* (2011).
4. van Halem, D., van der Laan, H., Heijman, S. G. J., van Dijk, J. C. & Amy, G. L. Assessing the sustainability of the silver-impregnated ceramic pot filter for low-cost household drinking water treatment. *Phys. Chem. Earth* **34**, 36–42 (2009).
5. Daniel, D., Marks, S. J., Pande, S. & Rietveld, L. Socio-environmental drivers of sustainable adoption of household water treatment in developing countries. *npj Clean Water* **1**, 1–6 (2018).
6. Hunter, P. R. Household water treatment in developing countries: Comparing different intervention types using meta-regression. *Environ. Sci. Technol.* **43**, 8991–8997 (2009).
7. Inauen, J., Hossain, M. M., Johnston, R. B. & Mosler, H. J. Acceptance and Use of Eight Arsenic-Safe Drinking Water Options in Bangladesh. *PLoS One* **8**, (2013).
8. Lantagne, D. S., Quick, R. & Mintz, E. D. Household water treatment and safe storage options in developing countries: a review of current implementation practices. *Woodrow Wilson Q.* 17–38 (2006) doi:10.1021/es301842u.
9. Loharikar, A. *et al.* Long-term impact of integration of household water treatment and hygiene promotion with antenatal services on maternal water treatment and hygiene practices in Malawi. *Am. J. Trop. Med. Hyg.* **88**, 267–274 (2013).
10. Ram, P. K. *et al.* Bringing safe water to remote populations: An evaluation of a portable point-of-use intervention in rural Madagascar. *Am. J. Public Health* **97**, 398–400 (2007).
11. Clasen, T. Scaling Up Household Water Treatment Among Low-Income Populations: Public Health and Environment, Water, Sanitation, Hygiene and Health, World Health Organization. *World Heal. Organ.* (2009).
12. Boutilier, M. S. H., Lee, J., Chambers, V., Venkatesh, V. & Karnik, R. Water Filtration Using Plant Xylem. *PLoS One* **9**, e89934 (2014).
13. Passmore, J. M., Rudolph, D. L., Mesquita, M. M. F., Cey, E. E. & Emelko, M. B. The utility of microspheres as surrogates for the transport of *E. coli* RS2g in partially saturated agricultural soil. *Water Res.* **44**, 1235–1245 (2010).
14. Radu, A. I., van Steen, M. S. H., Vrouwenvelder, J. S., van Loosdrecht, M. C. M. & Picioreanu, C. Spacer geometry and particle deposition in spiral wound membrane feed channels. *Water Res.* **64**, 160–176 (2014).
15. World Health Organization (WHO). WHO International Scheme to Evaluate Household Water Treatment Technologies Harmonized Testing Protocol: Technology Non-Specific. 1–5 (2014).
16. NSF International. *NSF Protocol P231 - Microbiological Water Purifiers.* (2014).
17. Kapikian, A. Z. & Shope, R. E. *Rotaviruses, Reoviruses, Coltiviruses, and Orbiviruses. Medical Microbiology* (1996).
18. Sperry, J. S. Evolution of Water Transport and Xylem Structure. *Int. J. Plant Sci.* **164**, S115–S127 (2003).

19. Troeger, C., Blacker, B.F., Khalil, I.A., Rao, P.C., Cao, S., Zimsen, S.R., Albertson, S.B., Stanaway, J.D., Deshpande, A., Abebe, Z. and Alvis-Guzman, N. Estimates of the global , regional , and national morbidity , mortality , and aetiologies of diarrhoea in 195 countries : a systematic analysis for the Global Burden of Disease Study 2016. *Lancet Infect. Dis.* 1211–1228 (2018) doi:10.1016/S1473-3099(18)30362-1.
20. World Health Organization (WHO). WHO International Scheme to Evaluate Household Water Treatment Technologies Harmonized Testing Protocol: Technology Non-specific. (2014).
21. NSF International. *NSF/ANSI 53: Drinking Water Treatment Units - Health Effects*. vol. 29 11–12 (2019).
22. Junter, G. A. & Lebrun, L. Cellulose-based virus-retentive filters: a review. *Rev. Environ. Sci. Biotechnol.* **16**, 455–489 (2017).
23. Cliver, D. O. Virus Interactions with Membrane Filters. *Biotechnol. Bioeng.* **X**, 877–889 (1968).
24. WHO. *Results of round II of the WHO international scheme to evaluate household water treatment technologies*. (2019).
25. Omarova, A., Tussupova, K., Berndtsson, R., Kalishev, M. & Sharapatova, K. Protozoan parasites in drinking water: A system approach for improved water, sanitation and hygiene in developing countries. *International Journal of Environmental Research and Public Health* (2018) doi:10.3390/ijerph15030495.
26. Viral Zone. Norovirus. https://viralzone.expasy.org/194?outline=all_by_species.
27. Modaber, I. Clostridium difficile. *Acta Med. Iran.* **18**, 111–128 (1975).
28. Casemore, D. P., Armstrong, M. & Sands, R. L. Laboratory diagnosis of cryptosporidiosis. *J. Clin. Pathol.* **38**, 1337–1341 (1985).
29. Tanyuksel, M. & Petri, W. A. Laboratory Diagnosis of Amebiasis. *Clin. Microbiol. Rev.* **16**, 713–729 (2003).
30. Drancourt, M. *38 - Acute Diarrhea. Infectious Diseases* (Elsevier Ltd, 2010). doi:10.1016/B978-0-7020-6285-8.00038-1.
31. Wikipedia. Adenoviridae.
32. Babu, M. M. & K. Sankaran. New strains of Vibrio Cholerae.
33. Britannica. Shigella dysenteriae type 1. <https://www.britannica.com/science/Shigella-dysenteriae>.
34. Wikipedia. Escherichia Coli.
35. Wikipedia. Aeromonas Hydrophilia.
36. Andino, A. & Hanning, I. Salmonella enterica: Survival, colonization, and virulence differences among serovars. *Sci. World J.* **2015**, (2015).
37. Fischer, G. H. & Paterek, E. Campylobacter. *StatPearls [Internet]. StatPearls Publ.* (2019).
38. World Health Organization (WHO). *Evaluating household water treatment options: Health-based targets and microbiological performance specifications*. https://www.who.int/water_sanitation_health/publications/2011/evaluating_water_treatment.pdf (2011).
39. Che, W. *et al.* Wood-Based Mesoporous Filter Decorated with Silver Nanoparticles for Water Purification. *ACS Sustain. Chem. Eng.* **7**, 5134–5141 (2019).
40. Vitas, S., Keplinger, T., Reichholf, N., Figi, R. & Cabane, E. Functional lignocellulosic material for the remediation of copper(II) ions from water: Towards the design of a wood

- filter. *J. Hazard. Mater.* **355**, 119–127 (2018).
41. Ryan, J. N. *et al.* Field and laboratory investigations of inactivation of viruses (PRD1 and MS2) attached to iron oxide-coated quartz sand. *Environ. Sci. Technol.* (2002) doi:10.1021/es011285y.
 42. Xu, Y. H., Nakajima, T. & Ohki, A. Adsorption and removal of arsenic(V) from drinking water by aluminum-loaded Shirasu-zeolite. *J. Hazard. Mater.* (2002) doi:10.1016/S0304-3894(02)00020-1.
 43. Barker, D. *Flowering Plant Origin, Evolution, & Phylogeny.* (1996).
 44. André, J. P. A study of the vascular organization of bamboos (Poaceae-Bambuseae) using a microcasting method. *IAWA J.* **19**, 265–278 (1998).
 45. Hindustan Unilever Ltd. Hindustan Unilever non-electric water purification products. <https://www.pureitwater.com/IN/pureit-water-purifiers/type/non-electric>.
 46. Kent. Kent gravity water purifiers. <https://www.kent.co.in/water-purifiers/gravity-uf/>.
 47. Eureka Forbes. AquaSure gravity water purifiers. <https://www.eurekaforbes.com/water-purifiers/technology/non-electric-gravity>.
 48. Tata. Tata Swach non-electric water purifiers. <https://tataswach.com/pages/non-electric-purifier>.
 49. Hindustan Unilever Ltd. Order PureIt GermKill Kit. <https://www.pureitwater.com/IN/order-gkk>.
 50. Forbes Eureka. Accessories. <https://www.eurekaforbes.com/accessories>.
 51. Swach, T. Spares. <https://tataswach.com/pages/spares>.
 52. Amazon. Kent Gold Optima Spare Kit. <https://www.amazon.in/Kent-Gold-Optima-Spare-Kit/dp/B00SMFPJG0>.
 53. United Nations. *Household size and composition around the world.* vol. 2 http://www.un.org/en/development/desa/population/publications/pdf/ageing/household_size_and_composition_around_the_world_2017_data_booklet.pdf (2017).
 54. Gandy, J. Water intake: validity of population assessment and recommendations. *Eur. J. Nutr.* **54**, 11–16 (2015).
 55. Arkhurst, B. Identification and Evaluation of Techniques for Quality Control of Low-Cost Xylem Filters. (Massachusetts Institute of Technology, 2018).
 56. KOPTEC. *200 Proof Ethanol 100 % Ethanol Technical Data Sheet* -. (2012).
 57. Sigma, M. *Material Safety Data Sheet for Ethanol absolute for analysis EMSURE® ACS,ISO,Reag. Ph Eur.* vol. 4 https://us.vwr.com/assetsvc/asset/en_US/id/16490607/contents (2012).
 58. US Food & Drug Administration (US FDA). Food Additive Status List. <https://www.fda.gov/food/food-additives-petitions/food-additive-status-list#ftnM>.
 59. International Council for Harmonisation. Guidance for Industry Q3C. *U.S. Dep. Heal. Hum. Serv. Food Drug Adm.* **9765**, 1–8 (2017).
 60. Alibaba. Food Grade Alcohol.
 61. Alibaba. Good quality food grade bulk methanol 99.5% CAS67-56-1 wholesale price.
 62. Hindustan Unilever Ltd. Hindustan Unilever Pureit - Instruction Manual. <https://www.pureitwater.com/IN/uploads/product/manual/pureit-classic-14l-manual.pdf>.
 63. Comprehensive Initiative on Technology Evaluation (Massachusetts Institute of Technology). Household Water Filter Evaluation, Suitability Report - Field Research in Ahmedabad, India. (2015).
 64. Comprehensive Initiative on Technology Evaluation at MIT. Household Water Filter

- Evaluation Sustainability Report. 1–24 (2015).
65. Distributors, H. H2O International Coconut Shell GAC Cartridge with KDF & Replaceable Sediment Pre-filter. <https://www.h2odistributors.com/kdf-gac-plus>.
 66. Crystal Clear Supply Inc. KDF/GAC Water Filter Replacement Cartridge. https://www.crystalclearsupply.com/KDF_GAC_Water_Filter_Cartridge_p/cf.htm.
 67. The Home Depot. Home Master Radial Flow GAC 20 Micron Replacement Water Filter. <https://www.homedepot.com/p/Perfect-Water-Technologies-Home-Master-Radial-Flow-GAC-20-Micron-Replacement-Water-Filter-CFrfgac20-20BB/203515345>.
 68. Serv-A-Pure. Home > Resins & Media > Filter Media & Filter Gravel > Activated Carbons. https://www.servapure.com/Water-Washed-Bituminous-Coal-12-x-40_c_5709.html.
 69. Indiamart. Granular Activated Carbon Coal-Based.
 70. Schreiber, B., Schmalz, V., Brinkmann, T. & Worch, E. The effect of water temperature on the adsorption equilibrium of dissolved organic matter and atrazine on granular activated carbon. *Environ. Sci. Technol.* **41**, 6448–6453 (2007).
 71. World Health Organization (WHO). Burden of disease from inadequate water in low- and middle-income countries. <http://apps.who.int/gho/data/view.main.INADEQUATEWATERv?lang=en>.
 72. Wikipedia. WHO regions. https://en.wikipedia.org/wiki/WHO_regions.
 73. Farjon, A. *A Handbook of the World's Conifers*. Brill Academic Publishers vol. I (2010).
 74. Bannan, M. W. Length tangential diameter and length/ width ratio of conifer tracheids. *Can. J. Bot.* **43**, 967–984 (1965).
 75. Troeger, C. *et al.* Rotavirus Vaccination and the Global Burden of Rotavirus Diarrhea among Children Younger Than 5 Years. *JAMA Pediatr.* (2018) doi:10.1001/jamapediatrics.2018.1960.

REVIEWER COMMENTS

Reviewer #1 (Remarks to the Author):

Dear Authors,

Thank you for addressing the reviewer comments and completing the additional laboratory work on the proof-of-concept of a sapwood filter. In my previous review I had asked the authors to be less biased about the success of the filter, in the review response the authors asked for specific examples of my concerns.

Here is a specific example of my concern, the claim that this proof-of-concept will have:
"transformative impact in enabling access to safe drinking water"

At this point, this work is in proof-of-concept stage, there are not even prototypes at this stage, simply a range of device configurations sketches. Additionally, the system proposed is quite complete - GAC (which needs to be replaced at breakthrough at some point) followed by sapwood (non toxic!), all with a 1-meter head, and concept sketches of designs.

The authors do not present data showing this filter would have "transformative impact in enabling access to safe drinking water". It is recommended to ensure that the conclusions are in line with the methods and results of the manuscript.

Additionally,

- Did anyone in India review and approve the study design and tools?
- Can you clarify the statement that the filters could be distributed "Even by post to remote locations" What percent of the population planned to be served has post? What would be the cost of posting?
- Can the authors clarify the statements about micro-enterprises and production, and reference experience with micro-enterprises for other filters?
- Can you clarify the statement these would also be useful "such as for disaster and emergency use" What disasters or emergencies?
What is the literature on filtration in emergencies and how does this inform this product? What is the literature on plastic water storage containers in emergencies and how does that inform the longevity of the "Plastic bag design"? Is this a robust design?

Minor edits:

- Reference 1 is outdated
- Consistency needed - > 3 or >3

- Figures are presented out of order in the text
- MechanismS of fouling
- Overuse of 'etc.' in scientific writing
- Reference for 8 Liters of water to meet daily drinking water requirement
- Use of "ethanol" and "alcohol" for treatment of wood - which is accurate? How certified food grade? Is this possible in LMIC?
-
- p-values are generally reported as <0.001 as a minimum
- "on an average"
- "onsthe"
- Why 4 significant digits in %'s - not that many samples were done
- Table 2 - why were these options selected?, in particular ceramic are not low-cost, compared to membrane and other available filters. Suggest reframing this in terms of filters.

Reviewer #2 (Remarks to the Author):

We thank the authors for their thoughtful revision of the paper. Upon review, I believe all my concerns were adequately addressed. It is particularly important to note the behaviour change aspects needed for appropriate adoption of such technologies. The paper is now very well suited for publication in Nature Communications.

Response to reviewer comments

We thank the reviewers for their careful review of the manuscript and for providing constructive feedback. In particular, we are grateful to the many suggestions of Reviewers 1, and to Reviewer 2 for recommending publication of the manuscript. We have revised the manuscript and SI as per the reviewer suggestion, which we believe adequately addresses the reviewer comments. Below, we present a point-by-point response to reviewer comments.

Original reviewer comments are *italicized*, and our response is presented in normal font with revisions to the manuscript shown in blue. Unmodified text from the main manuscript is shown as normal text. Text deletions have been shown as ~~strikethrough text~~. Text additions have been *italicized*. A separate review-only file containing highlighted changes made to the manuscript and SI is included for the reviewer's reference.

Reviewer #1

Thank you for addressing the reviewer comments and completing the additional laboratory work on the proof-of-concept of a sapwood filter.

R1. In my previous review I had asked the authors to be less biased about the success of the filter, in the review response the authors asked for specific examples of my concerns. Here is a specific example of my concern, the claim that this proof-of-concept will have: "transformative impact in enabling access to safe drinking water". At this point, this work is in proof-of-concept stage, there are not even prototypes at this stage, simply a range of device configurations sketches. Additionally, the system proposed is quite complete - GAC (which needs to be replaced at breakthrough at some point) followed by sapwood (non toxic!), all with a 1-meter head, and concept sketches of designs. The authors do not present data showing this filter would have "transformative impact in enabling access to safe drinking water". It is recommended to ensure that the conclusions are in line with the methods and results of the manuscript.

We thank the reviewer for the comment. We have modified the specified instance as per the reviewer's suggestion and have re-checked the manuscript for unsubstantiated claims. We would also like to clarify that the device presented in Fig. 6 is a functional device prototype incorporating xylem filters that was tested in the lab and the field, and that this specific sentence was an outlook and not a conclusion. We do see the reviewer's point that it can come across as a conclusion, and given that multiple barriers (including behavioral, political, commercial) that would need to be overcome for large-scale impact to occur, have removed the phrase 'transformative impact'.

Page 3 of main manuscript

The ability to create filters from different gymnosperms, widespread availability of gymnosperm xylem¹ (Fig. 1d), low cost, natural appeal, ease of transport and distribution, and the traditional comfort associated with wood, could help xylem filters lower the barriers of access, affordability, and social acceptance, and ~~create a transformative impact in enabling~~ *thereby facilitate* access to safe drinking water.

Page 11 of main manuscript:

First, the abundant availability of gymnosperms, local access to other raw materials required for filter fabrication (primarily alcohol for dry preservation), simplicity of the manufacturing process, and open access to this technology (filter fabrication has not been patented to facilitate adoption) ~~provides~~ *could provide* an opportunity for local manufacture of these filters, making them more accessible to local communities (see Supplementary Note 8 for a list of equipment that can be used in low-resource settings).

Page 14 of main manuscript

Due to the worldwide availability of gymnosperms and simplicity of the filter fabrication process, xylem filters ~~present opportunities~~ *offer potential* for local manufacture of a wide variety of water treatment and other filtration products, ranging from compact filter pouches for use in emergencies to household filtration devices (Fig. 7b). This opens up ~~opportunities~~ *the possibility* for involvement of micro-entrepreneurs at a global scale², implementation of innovative business models, and engagement of local communities in filter distribution^{3,4} and manufacture to raise awareness about importance of safe drinking water and encourage adoption.

R2. *Did anyone in India review and approve the study design and tools?*

The 'National Ethical Guidelines for Biomedical and Health Research Involving Human Participants' guidelines published by the Indian Council of Medical Research (ICMR) were followed while conducting user research in India. These guidelines suggest that social and behavioral research for health applications should be approved by the ethics committee for the researchers' institution, which in our case would be the Massachusetts Institute of Technology Committee on the Use of Human Subjects (MIT COUHES). All human-subjects research procedures were approved by MIT COUHES under protocol 1612798762 (also described in Method M19 on Page 30 of main manuscript).

3. *Can you clarify the statement that the filters could be distributed "Even by post to remote locations" What percent of the population planned to be served has post? What would be the cost of posting?*

We thank the reviewer for the comment. In India, the postal network serves ~1.3 billion people, which is practically the entire population of the country⁵. 90% of the post offices in India are in rural locations⁵. Beyond mail delivery, the postal network in India offers a wide range of supply chain and logistics support services for businesses, which involves managing the entire value chain from collection to storage to transmission to distribution across the country⁵. Although it is likely that a dedicated supply chain and distribution network would be designed for transporting xylem filters from the manufacturing facility to the users if the technology is commercialized, the postal network can be leveraged to transport filters to areas where the distribution network has not penetrated. Based on our field visits in India, the postal charges for the postal charges for sending filters as a package of 200 amounts to 0.03–0.07¢ per filter (these estimates will vary with geography and the number of filters being shipped).

Beyond India, the postal network could potentially be a channel for aiding filter distribution in other countries too, depending upon the need for doing so, strength of the postal network, and the services offered by it.

4. *Can the authors clarify the statements about micro-enterprises and production, and reference experience with micro-enterprises for other filters?*

We believe the reviewer is referring to the following statement in the manuscript where microenterprises have been mentioned: 'Due to the worldwide availability of gymnosperms and simplicity of the filter fabrication process, xylem filters present opportunities for local manufacture of a wide variety of water treatment and other filtration products, ranging from compact filter pouches for use in emergencies to household filtration devices (Fig. 7b). This opens up opportunities for involvement of micro-entrepreneurs at a global scale, implementation of innovative business models, and engagement of local communities in filter distribution and manufacture to raise awareness about importance of safe drinking water and encourage adoption.'

What we mean by the statement is that the key raw materials required for xylem filter manufacture include gymnosperm sapwood and food-grade alcohol, which are available in several locations across the globe. In contrast to conventional membrane-based filter cartridges, the manufacturing process for filter fabrication is relatively simple and has not been patented for ease of dissemination of the technology. Accessibility to raw materials, simplicity of the manufacturing process, and the open source nature of the technology can facilitate its uptake by entrepreneurs and NGOs interested in taking it to the users and create an opportunity for business models where local communities can be engaged in filter manufacture and distribution. For example, communities could be involved to source the right kind of wood, or to process discarded branches, and bring them to a manufacturing facility, to distribute manufactured filters within rural/slum communities (the strategy of training local community members in retail and marketing for product sales and last-mile delivery is commonly employed by several fast moving consumer goods companies^{3,4}), collect used filters to make charcoal, etc., and in some cases to fabricate filters in manufacturing facilities. The involvement of local communities in filter production and distribution could also help raise awareness about the safe importance of drinking water and encourage adoption of water treatment methods.

We thank the reviewer for the suggestion to add references for microenterprises in water filtration. An example of an NGO that has implemented a micro-enterprise based model for water filters is Potters for Peace. Potters for Peace is a non-profit organization that has facilitated the dissemination of ceramic filters across the globe by training potters in several low-income communities to make these filters in their local facility in a sustainable manner². The organization has established best practices for ceramic filter manufacture, offers training programs for potters and also provides ground assistance with factory set-up worldwide. Such a model could serve as an inspiration and an example template for xylem filter dissemination. We have included the reference in the manuscript and also added a Supplementary Note to further clarify how local communities and micro-enterprises could be involved in filter manufacture and distribution.

Page 14 of manuscript:

This opens up the possibility for involvement of micro-entrepreneurs at a global scale², implementation of innovative business models, and engagement of local communities in filter distribution^{3,4} and manufacture to raise awareness about importance of safe drinking water and encourage adoption (*see Supplementary Note 13 for further details on potential avenues for engaging local communities and micro-enterprises*).

Page 19 of SI

Supplementary Note 13: Potential avenues for engagement of micro-enterprises and local communities in xylem filter manufacture and distribution

The key raw materials required for xylem filter manufacture include gymnosperm sapwood and food-grade alcohol, which are available in several locations across the globe. In contrast to conventional membrane-based filter cartridges, the manufacturing process for filter fabrication is relatively simple and has not been patented for ease of dissemination of the technology. Accessibility to raw materials, simplicity of the manufacturing process, and the open source nature of the technology can facilitate its uptake by entrepreneurs and NGOs interested in taking it to the users and create an opportunity for business models where local communities can be engaged in filter manufacture and distribution. For example, communities could be involved to source the right kind of wood, or to process discarded branches, and bring them to a manufacturing facility, to distribute manufactured filters within rural/slum communities (the strategy of training local community members in retail and marketing for product sales and last-mile delivery is commonly employed by several fast moving consumer goods companies^{3,4}), collect used filters to make charcoal, etc., and in some cases to fabricate filters in manufacturing facilities. The involvement of local communities in filter production and distribution could also help raise awareness about the safe importance of drinking water and encourage adoption of water treatment methods.

An example of an NGO that has implemented a micro-enterprise based model for water filters is Potters for Peace. Potters for Peace is a non-profit organization that has facilitated the dissemination of ceramic filters across the globe by training potters in several low-income communities to make these filters in their local facility in a sustainable manner². The organization has established best practices for ceramic filter manufacture, offers training programs for potters and also provides ground assistance with factory set-up worldwide. Such a model could serve as an inspiration and an example template for xylem filter dissemination.

5. Can you clarify the statement these would also be useful "such as for disaster and emergency use" What disasters or emergencies? What is the literature on filtration in emergencies and how does this inform this product? What is the literature on plastic water storage containers in emergencies and how does that inform the longevity of the "Plastic bag design"? Is this a robust design?

We thank the reviewer for the note. We were referring to disasters and emergency situations such as floods, disease outbreaks, contamination of public water supply network, etc., where access to safe drinking water is a major cause of concern. Some common methods employed to provide safe drinking water to the affected population under such situations include distribution of bottled/package water procured from government agencies or commercial vendors, delivering water through water tankers, using neighboring water systems and using water treatment systems

at point-of-entry or point-of-use⁶. Point-of-use, household water treatment (HWT) methods can be useful during the acute phase of an emergency when responders cannot reach the affected population and in the recovery phase when longer term solutions are still under development⁷. Effective use of HWT treatment methods, including water filtration, has been shown to effectively reduce the incidence of water-borne diseases during emergencies⁷. Examples of studies where filtration methods were evaluated during emergencies include the following: use of ceramic filters after the 2004 tsunami in Sri Lanka⁸ and 2003 floods in Dominican Republic⁹, distribution of ceramic/biosand after the 2010 earthquake in Haiti^{10,11}, and use of membrane/ceramic filters during an emergency in Pakistan in 2007¹². The key learnings from these studies, in conjunction with guidelines enlisted in the Sphere handbook (the primary reference tool for NGOs, UN agencies, and governments to respond to emergencies and disasters) that can help guide filter design and its implementation for emergency use include the following: a) the device should be able to meet minimum drinking water requirement for survival, which is 2.5-3 liters per person per day¹³ and meet the minimum drinking water quality requirements during emergencies (< 10 CFU/100 mL, turbidity < 5 NTU)¹³, b) the target price for emergency filters can be benchmarked against reported costs of filtration devices distributed during emergencies; during the 2003 floods in Dominican Republic, ceramic filters had an upfront cost of \$15 (though they were distributed free of charge) with \$4.50 recurring cost every 6 months for candle replacement⁹, c) major factors affecting HWT usage rates amongst the affected population included quality of source water, prior experience with using HWT methods, need for training associated with device use, availability of replacement parts, level of programmatic support, ease of portability of device, and the living environment (usage rates varied between people living in permanent shelters and those moving between temporary shelters)^{8-10,12,14}. Based on findings presented in the manuscript, xylem filters could be designed to meet the performance and cost targets specified above and due to their light weight, could be easy to transport and distribute. Other factors which could affect product adoption such as user training, filter lifetime, supply and distribution strategy, and user-perceived need for product would need to be further investigated.

Plastic containers have often been distributed for safe water storage and bucket chlorination during disasters and emergencies¹⁵⁻¹⁸. Several membrane filters designed for emergency use are available commercially^{12,19,20}, and some of these are membranes are housed in a plastic bag/container^{12,20}. The plastic bag design depicted in Fig. 7b is a conceptual sketch. If xylem filters are designed for emergency use, the plastic bags could be made out of Poly-vinyl Chloride (PVC) or Poly-propylene, which are also used for fabricating intravascular (IV) fluid bags. The robustness and longevity of the plastic bags would depend on the use case; filters could be designed to last for a single use or for a few days, weeks or months with provisions for cartridge replacement.

We have added a Supplementary Note to summarize the literature on household water treatment during emergencies and highlight the key design considerations for emergency use filters. The note has been referenced in the main manuscript.

Page 14 of main manuscript:

Due to the worldwide availability of gymnosperms and simplicity of the filter fabrication process, xylem filters present opportunities for local manufacture of a wide variety of water treatment and other filtration products, ranging from compact filter pouches for use in emergencies to household

filtration devices (Fig. 7b, see *Supplementary Note 12 for further details on HWT for emergency use*).

Page 18 of SI:

Supplementary Note 12: HWT for emergency use

During disasters and emergency situations such as floods, disease outbreaks, contamination of public water supply network, etc., access to safe drinking water is a major cause of concern. Some common methods employed to provide safe drinking water to the affected population under such situations include distribution of bottled/package water procured from government agencies or commercial vendors, delivering water through water tankers, using neighboring water systems and using water treatment systems at point-of-entry or point-of-use⁶. Point-of-use, household water treatment (HWT) methods can be useful during the acute phase of an emergency when responders cannot reach the affected population and in the recovery phase when longer term solutions are still under development⁷. Effective use of HWT treatment methods, including water filtration, has been shown to effectively reduce the incidence of water-borne diseases during emergencies⁷. Examples of studies where filtration methods were evaluated during emergencies include the following: use of ceramic filters after the 2004 tsunami in Sri Lanka⁸ and 2003 floods in Dominican Republic⁹, distribution of ceramic/biosand after the 2010 earthquake in Haiti^{10,11}, and use of membrane/ceramic filters during an emergency in Pakistan in 2007¹². The key learnings from these studies, in conjunction with guidelines enlisted in the Sphere handbook (the primary reference tool for NGOs, UN agencies, and governments to respond to emergencies and disasters) that can help guide filter design and its implementation for emergency use include the following: a) the device should be able to meet minimum drinking water requirement for survival, which is 2.5-3 liters per person per day¹³ and meet the minimum drinking water quality requirements during emergencies (< 10 CFU/100 mL, turbidity < 5 NTU)¹³, b) the target price for emergency filters can be benchmarked against reported costs of filtration devices distributed during emergencies; during the 2003 floods in Dominican Republic, ceramic filters had an upfront cost of \$15 (though they were distributed free of charge) with \$4.50 recurring cost every 6 months for candle replacement⁹, c) major factors affecting HWT usage rates amongst the affected population included quality of source water, prior experience with using HWT methods, need for training associated with device use, availability of replacement parts, level of programmatic support, ease of portability of device, and the living environment (usage rates varied between people living in permanent shelters and those moving between temporary shelters)^{8-10,12,14}. Based on findings presented in the manuscript, xylem filters could be designed to meet the performance and cost targets specified above and due to their light weight, could be easy to transport and distribute. Other factors which could affect product adoption such as user training, filter lifetime, supply and distribution strategy, and user-perceived need for product would need to be further investigated.

The plastic bag design depicted in Fig. 7b is a conceptual sketch. Plastic containers have often been distributed for safe water storage and bucket chlorination during disasters and emergencies¹⁵⁻¹⁸. Several membrane filters designed for emergency use are available commercially^{12,19,20}, and some of these are membranes are housed in a plastic bag/container^{12,20}. The robustness and longevity of the plastic bags would depend on the use case; filters could be designed to last for a single use or for a few days, weeks or months with provisions for cartridge replacement.

6. *Reference 1 is outdated*

We thank the reviewer for the comment. We have updated the reference.

Page 1 of manuscript:

Diarrheal diseases caused by microbial contamination of water and poor sanitation *are a global problem. In 2019, diarrheal diseases accounted for 1.5 million deaths per year, primarily in resource-limited settings amongst children under the age of five*²¹.

7. *Consistency needed - > 3 or >3*

We thank the reviewer for the comment. We have removed the space between the ‘greater than’ sign and the number across the entire manuscript for consistency. Changes have not been specified here or highlighted in the manuscript.

8. *Figures are presented out of order in the text*

We thank the reviewer for the note. We believe the reviewer is referring to the text on Page 5 of the manuscript where Supplementary Fig. 1c-f were referred to before Supplementary Fig. 1b. The ordering has been corrected and highlighted in the revised manuscript.

9. *MechanismS of fouling*

We thank the reviewer for the comment. We have corrected the typo.

Page 2 of the main manuscript:

However, several other material characteristics of xylem that are critical for practical water filtration applications, such as its structural stability over the course of its shelf- and operational life, susceptibility to different foulants present in water, *and ~~mechanism~~ mechanisms of fouling*, remain to be explored.

10. *Overuse of 'etc.' in scientific writing*

We thank the reviewer for the note. We have reduced the use of ‘etc.’ in the manuscript.

Page 2 of main manuscript:

However, several other material characteristics of xylem that are critical for practical water filtration applications, such as its structural stability over the course of its shelf- and operational life, susceptibility to different foulants present in water, *and mechanisms of fouling, ~~etc.~~* remain to be explored. While the hydraulic properties of xylem have been well-characterized in the context of sap transport in plants^{22–24}, xylem’s functional attributes as a water filter, such as flow rate, filtration capacity, *and variation in flow rate over time, ~~etc.~~*, particularly with contaminated water as the fluid medium and in the absence of active transport mechanisms that regulate flow in plants, are currently not well understood.

Page 10 of main manuscript:

A wide range of device configurations could be designed to suit different use cases, water quality, resources available, *and* user preferences, *etc.*

Page 13 of main manuscript:

Timescales for degradation would depend on the feed water quality, type of wood used for manufacturing the filter, *and local environmental factors etc.* and could be of the order of 1-2 weeks^{53,54} or less since bacteria are directly seeded on pit membranes during filtration.

11. *Reference for 8 Liters of water to meet daily drinking water requirement*

We thank the reviewer for the comment. The rationale for using 8 liters as the minimum daily requirement has been provided in Supplementary Note 1. We have referenced this note in the main manuscript.

Page 3 of main manuscript:

Literature reports and our field trips to India revealed that, to be useful in households in resource-limited settings, xylem filters should a) process at least 8 L of water to meet the daily drinking water requirement (*see Supplementary Note 1*), b) have flow rates of at least 1 L/h, c) effectively remove contaminants²⁵, d) function reliably with contaminated water, e) operate under gravity with heads less than 1 m to minimize operation costs and space requirements, and f) be easy to access and use²⁵ (Supplementary Note 1).

12. *Use of "ethanol" and "alcohol" for treatment of wood - which is accurate? How certified food grade? Is this possible in LMIC?*

We thank the reviewer for the note. While our laboratory and field experiments involved the use of ethanol for dry preservation, any other food-grade alcohol could also be used instead of ethanol (data presented in Supplementary Fig. 1f). Since the filtrate from xylem filters tested in lab and field studies was not intended for consumption, the ethanol used for fabricating filters for these studies was reagent grade (CAS number 64-17-5, Koptec (200-Proof) and Merck (> 99.9% purity)^{26,27}). When manufacturing filters for practical use, appropriate food safety standards should be consulted to determine the grade of alcohol (methanol, ethanol, isopropyl alcohol, etc.) that can be used²⁸. Further, the level of residual alcohol (especially if methanol is used) in dried filters should be maintained within the permissible limits for human consumption as prescribed by food safety standards (also specified on Page 6 of main manuscript)^{28,29}. Food-grade alcohols (including methanol) are commonly available commercially across the world (including low and middle-income countries) and used for several applications such as baking, food processing, cosmetic manufacturing, production of botanical oils and extracts, etc.^{30,31}.

In the main manuscript, we have specified the type of alcohol (ethanol or isopropyl alcohol) while presenting experimental results. As we expect other alcohols to have a similar effect as ethanol/isopropyl alcohol on pit aspiration during drying, we have used the phrase ‘alcohol treatment’ to describe a more generalized technique for dry preservation at certain instances.

The wording in the Methods section could have been more accurate and precise with regards to the type of alcohol that was used. We have revised this section as follows:

Page 26 of main manuscript

M5. Treatment of xylem filters Xylem filters (freshly cut or hot-water soaked) of 1 cm and 4-5 cm diameter were treated by flushing and soaking, respectively (filter to ~~alcohol~~ *ethanol/isopropyl alcohol* volume ratio of 1:6–8). For flushing, 10 psi gas pressure was used to drive the desired volume of ~~alcohol~~ *ethanol/isopropyl alcohol* loaded in plastic tubing in which xylem filters were mounted, similar to permeance measurements described above. For soaking, filters were placed edge-on in appropriately sized glass vessels and completely immersed in ethanol for 24–48 h. Most of the filters (~90%) were soaked for more than 24 hours, in which case the ~~alcohol~~ *ethanol* in the vessel was typically replaced with fresh ~~alcohol~~ *ethanol* after 24 h. Filters manufactured in the field were treated with ~~alcohol~~ *ethanol* only by soaking.

M6. Drying of xylem filters In laboratory experiments, all filters (soaked in water or in ~~alcohol~~ *ethanol/isopropyl alcohol*) were dried by placing edge-on on aluminum foil in an oven (VWR 1410 vacuum oven) at atmospheric pressure and 45°C.

Page 28 of main manuscript

M15. Testing xylem filters in field studies The feed was introduced after 5–10 min to allow sufficient time for *isopropyl* alcohol evaporation. For microbiological testing, the feed and filtrate were collected in glass or plastic bottles that were previously disinfected in boiling hot water for 15–20 min. The filtrate for microbiological testing was collected during the filtration process as follows. Prior to collecting the filtrate for microbiological analysis, the bottom surface of the xylem filter, the hose clamp, and lower end of the tubing were wiped with cotton soaked in isopropyl alcohol (while filtration continued). Collection of the filtered sample was started after 10 min to allow sufficient time for any residual *isopropyl* alcohol to be flushed or evaporated.

Page 31 of main manuscript

M20. Microbiological performance as per WHO protocol Collection of the filtered sample was started after 10 min to allow sufficient time for any residual ~~alcohol~~ *ethanol* to be flushed or evaporated.

13. *p-values are generally reported as <0.001 as a minimum*

We thank for the note. We have changed the reported of p-values as per the reviewer's suggestion.

Page 5 of main manuscript:

The rejection performance of ethanol-dried filters with 1 µm microspheres was significantly better than water-dried filters ($p = 2.7 \times 10^{-5}$, 3.0×10^{-9} , and 1.4×10^{-6} , $p < 0.001$ for 0.25-, 0.50-, and 1-inch filters respectively) and comparable to fresh filters ($p = 0.02$, 0.59, and 0.08 for 0.25-, 0.50-, and 1-inch filters respectively) (Fig. 2f).

14. "on an average"

We thank the reviewer for pointing out this error. We have corrected it in the revised manuscript.

Page 8 of main manuscript:

When used in conjunction with cloth pre-filtration, the GAC column improved the performance of xylem filter with CTW significantly (Fig. 4f); on an average, capacity and flow rates increased by a factor of ~3x and 5x respectively.

15. "onsthe"

We thank the reviewer for catching the typo. We have corrected it in the revised manuscript.

Page 11 of main manuscript:

Some challenges encountered during device design included obtaining a leak-proof seal between the holder and xylem filter due to irregularities onsthe on the wood surface, and preventing air entrapment in the tubing and filter holder that could obstruct flow.

16. *Why 4 significant digits in %'s - not that many samples were done*

We thank the reviewer for the comment. The standard deviation of filter rejection performance (measured in %) has been reported to two significant digits and the mean values have been reported to the same number of decimal points as the standard deviation, which is common practice.

17. *Table 2 - why were these options selected?, in particular ceramic are not low-cost, compared to membrane and other available filters. Suggest reframing this in terms of filters.*

We thank the reviewer for the note. The options were selected because chlorination, solar disinfection, ceramic filtration are some of the most widely promoted water treatment technologies in low- and middle-income countries that have been proven to reduce the incidence of diarrheal diseases^{32,33}. Since the xylem filters are primarily targeted towards such communities, we selected these three options for comparison. Other options that are also popular amongst low-income groups include biosand filtration and flocculation/disinfection. These were not initially enlisted in the table, but flocculation/disinfection has now been included. Biosand filtration was not included because, unlike aforementioned HWT methods, official World Health Organization data on their effectiveness against water-borne pathogens was unavailable.

Table 2 has been updated as follows:

Technology	Level of protection	Advantages	Limitations
Chlorination ³⁴	Targeted (bacteria and viruses only)	 • Portable, light-weight • Simple to use, no maintenance 	 • Less effective in inorganic-rich or turbid water • Unpleasant taste and odor • Dosage depends on water quality • Harmful organic by-products
Solar disinfection ³⁴	Targeted to comprehensive (bacteria, virus, protozoa)	 • Little/no maintenance • Minimal chance of re-contamination • Simple to use 	 • Poor efficacy with turbid water • Dependent on weather • Volume to treat depends on availability of intact container • Long treatment time

Ceramic filters ³⁴	Targeted (bacteria and protozoa only)	 Local production Minimal chance of re-contamination 	 Performance heterogeneity due to manufacturing variability and susceptibility to cracks Requires regular cleaning Difficult to transport Low flow rate: 1-3 L/h
Flocculation-disinfection ^{34,35}	Comprehensive (bacteria, virus and protozoa)	 Residual protection against re-contamination Easy to transport (typically available as sachets) Reduction of some heavy metals and particle-associated pesticides 	 Need for multiple steps and additional user support Can have a negative effect on odor and taste Requires reliable supply chain
Xylem filters	Comprehensive (>3-log removal of bacteria and virus)	 Biodegradable Light weight Minimal chance of recontamination Local production 	 Frequency of replacement is relatively high (maintenance concerns) Needs pre-filtration or large filter sizes with water having high turbidity or organic content Low flow rate: 1-3 L/h

References:

1. Farjon, A. *A Handbook of the World's Conifers*. Brill Academic Publishers vol. I (2010).
2. Potters for Peace. Ceramic Water Filter Project.
3. Dolan, C., Johnstone-Louis, M. & Scott, L. Shampoo, saris and SIM cards: Seeking entrepreneurial futures at the bottom of the pyramid. *Gend. Dev.* (2012) doi:10.1080/13552074.2012.663619.
4. Neuwirth, B. Marketing Channel Strategies in Rural Emerging Markets: Unlocking Business Potential. *Kellogg Sch. Manag.* 40 (2012).
5. Department of Posts (Ministry of Communication India). *Annual Report 2019-20*. (2019).
6. Bross, L. *et al.* Insecure security: Emergency water supply and minimum standards in Countries with a high supply reliability. *Water (Switzerland)* 11, (2019).
7. Lantagne, D. & Clasen, T. Point-of-use water treatment in emergency response. *Waterlines* 31, 30–52 (2012).
8. Casanova, L. M., Walters, A., Naghawatte, A. & Sobsey, M. D. A post-implementation evaluation of ceramic water filters distributed to tsunami-affected communities in Sri Lanka. *J. Water Health* (2012) doi:10.2166/wh.2012.181.
9. Clasen, T. Household-Based Ceramic Water Filters for the Treatment of Drinking Water in Disaster Response: An Assessment of a Pilot Programme in the Dominican Republic. *Water Pract. Technol.* 1, (2006).
10. Lantagne, D. S. & Clasen, T. F. Use of household water treatment and safe storage methods in acute emergency response: Case study results from Nepal, Indonesia, Kenya, and Haiti. *Environ. Sci. Technol.* 46, 11352–11360 (2012).
11. Lantagne, D. & Clasen, T. Effective use of household water treatment and safe storage in response to the 2010 haiti earthquake. *Am. J. Trop. Med. Hyg.* 89, 426–433 (2013).
12. Ensink, J. H. J., Bastable, A. & Cairncross, S. Assessment of a membrane drinking water filter in an emergency setting. *J. Water Health* 13, 362–370 (2015).
13. Sphere Association. *The Sphere Handbook: Humanitarian Charter and Minimum Standards in Humanitarian Response*. CHS Alliance, Sphere Association and Groupe URD vol. 1 (2018).
14. Staveley, L. & Lantagne, D. *Oxfam Household Water Treatment Technical Brief*. (2008).
15. Sikder, M. *et al.* Effectiveness of water chlorination programs along the emergency-transition-post-emergency continuum: Evaluations of bucket, in-line, and piped water

- chlorination programs in Cox's Bazar. *Water Res.* (2020)
doi:10.1016/j.watres.2020.115854.
16. Ferron, S. OXFAM EVALUATION REPORTS ENABLING ACCESS TO NON-FOOD ITEMS IN AN EMERGENCY RESPONSE A review of Oxfam programmes. (2017).
 17. Ali, S. I., Ali, S. S. & Fesselet, J. F. Evidence-based chlorination targets for household water safety in humanitarian settings: Recommendations from a multi-site study in refugee camps in South Sudan, Jordan, and Rwanda. *Water Res.* **189**, 116642 (2021).
 18. Center for Disease Control and Prevention (CDC). The Oxfam Bucket. [https://www.cdc.gov/safewater/oxfam-bucket.html#:~:text=The Oxfam Bucket \(Oxfam\),for use in program implementation](https://www.cdc.gov/safewater/oxfam-bucket.html#:~:text=The Oxfam Bucket (Oxfam),for use in program implementation).
 19. GrifAid. GrifAid Water Filters.
 20. Lifesaver. Lifesaver Cube.
 21. World Health Organization, W. The top 10 causes of death. <http://www.who.int/news-room/fact-sheets/detail/the-top-10-causes-of-death>.
 22. Pittermann, J., Sperry, J. S., Hacke, U. G., Wheeler, J. K. & Sikkema, E. H. Inter-tracheid pitting and the hydraulic efficiency of conifer wood: The role of tracheid allometry and cavitation protection. *Am. J. Bot.* **93**, 1265–1273 (2006).
 23. Hacke, U. G., Sperry, J. S. & Pittermann, J. Analysis of circular bordered pit function II. Gymnosperm tracheids with torus-margo pit membranes. *Am. J. Bot.* **91**, 386–400 (2004).
 24. Sperry, J. S., Hacke, U. G. & Pittermann, J. Size and function in conifer tracheids and angiosperm vessels. *Am. J. Bot.* **93**, 1490–1500 (2006).
 25. Peter-Varbanets, M., Zurbrügg, C., Swartz, C. & Pronk, W. Decentralized systems for potable water and the potential of membrane technology. *Water Research* (2009)
doi:10.1016/j.watres.2008.10.030.
 26. KOPTEC. *200 Proof Ethanol 100 % Ethanol Technical Data Sheet -*. (2012).
 27. Sigma, M. *Material Safety Data Sheet for Ethanol absolute for analysis EMSURE® ACS,ISO,Reag. Ph Eur.* vol. 4
https://us.vwr.com/assetsvc/asset/en_US/id/16490607/contents (2012).
 28. US Food & Drug Administration (US FDA). Food Additive Status List.
<https://www.fda.gov/food/food-additives-petitions/food-additive-status-list#ftnM>.
 29. International Council for Harmonisation. Guidance for Industry Q3C. *U.S. Dep. Heal. Hum. Serv. Food Drug Adm.* **9765**, 1–8 (2017).
 30. Alibaba. Food Grade Alcohol.
 31. Alibaba. Good quality food grade bulk methanol 99.5% CAS67-56-1 wholesale price.
 32. Clasen, T. T. F. *et al.* Interventions to improve water quality for preventing diarrhoea (Review). *Cochrane Libr.* **2015**, CD004794 (2015).
 33. Sobsey, M. D., Stauber, C. E., Casanova, L. M., Brown, J. M. & Elliott, M. A. Point-of-use household drinking water filtration: a practical, effective solution for providing sustained access to safe drinking water in the developing world. *Environ. Sci. Technol.* **42**, 4261–4267 (2008).
 34. World Health Organization (WHO). *Results of round II of the WHO international scheme to evaluate household water treatment technologies.*
http://www.who.int/household_water/scheme/household-water-treatment-report-round-1/en/#.VtmmIDJjI5c.mendeley (2019).
 35. World Health Organization (WHO). *Results of Round I of the WHO International Scheme to Evaluate Household Water Treatment Technologies.* World Health Organization

[http://eutils.ncbi.nlm.nih.gov/entrez/eutils/elink.fcgi?dbfrom=pubmed&id=15620560&retmode=ref&cmd=prlinks%5Cnpapers2://publication/doi/10.1016/S1473-3099\(04\)01253-8%5Cnhttp://www.who.int/household_water/scheme/household-water-treatment-report-round-1/en/#](http://eutils.ncbi.nlm.nih.gov/entrez/eutils/elink.fcgi?dbfrom=pubmed&id=15620560&retmode=ref&cmd=prlinks%5Cnpapers2://publication/doi/10.1016/S1473-3099(04)01253-8%5Cnhttp://www.who.int/household_water/scheme/household-water-treatment-report-round-1/en/#) (2016).